# Learning with Differentially Private Sliced Wasserstein Gradients

**David Rodríguez-Vítores**                                          *david.rodriguez.vitores@uva.es*
*Universidad de Valladolid*

**Clément Lalanne**                                          *clement.lalanne@math.univ-toulouse.fr*
*Institute of Mathematics of Toulouse.*

**Jean-Michel Loubes**                                          *jean-michel.a.loubes@inria.fr*
*ANITI & Regalia INRIA*

**Reviewed on OpenReview:** *https://openreview.net/forum?id=pqo5VNImZE*

## Abstract

In this work, we introduce a novel framework for privately optimizing objectives that depend on sliced Wasserstein distances between data-dependent empirical measures. Our main theoretical contribution is a non-trivial analysis of the sensitivity of the Wasserstein gradients to individual data points, derived from an explicit formulation of the gradient in a fully discrete setting. This enables strong privacy guarantees with minimal utility loss. We demonstrate that standard privacy accounting methods naturally extend to Wasserstein-based objectives, allowing for large-scale private training. This supports a wide range of private machine learning applications involving distribution matching under privacy constraints on the source, the target, or both. These include: (i) an in-processing method for fairness mitigation using a private Wasserstein penalty, and (ii) what we believe is the first approach for training private sliced Wasserstein autoencoders. We validate our framework through experiments showing its ability to effectively balance privacy and utility, offering a theoretically grounded approach to privacy-preserving machine learning with sliced Wasserstein losses.

## 1 Introduction

Privacy has emerged as a critical consideration in machine learning, driven by widespread concerns about data misuse and regulatory frameworks such as GDPR and the European AI Act. Traditional machine learning approaches often require access to sensitive information, and it is well-established that releasing statistics based on such data, without appropriate safeguards, can result in serious privacy breaches (Narayanan & Shmatikov, 2006; Backstrom et al., 2007; Fredrikson et al., 2015; Dinur & Nissim, 2003; Homer et al., 2008; Loukides et al., 2010; Narayanan & Shmatikov, 2008; Sweeney, 2000; Wagner & Eckhoff, 2018; Sweeney, 2002).

Differential privacy (Dwork et al., 2006b), has become the gold standard for addressing these concerns, providing rigorous guarantees on individual data protection during model training. Differential privacy incorporates randomness into the computation process, ensuring that the estimator relies not only on the dataset, but also on an additional source of randomness. This mechanism obscures the influence of individual data points, safeguarding user privacy. The formal definition of differential privacy and the basic results and tools used throughout the paper are given in Section 2. To the non-expert reader, it is only useful to know that privacy is tuned by two scalar numbers $\varepsilon, \delta \geq 0$, and that the smaller they are, the stronger the privacy guarantees are. Privacy is usually obtained via noise addition mechanisms, of which the amount of noise usually scales with the *sensitivity* of a given query Dwork et al. (2006b). Prominent organizations like the

US Census Bureau (Abowd, 2018), Google (Erlingsson et al., 2014), Apple (Thakurta et al., 2017), and Microsoft (Ding et al., 2017) have adopted this approach. Notably, an extended body of literature studies the interplay between privacy and learning / statistics (Wasserman & Zhou, 2010; Barber & Duchi, 2014; Diakonikolas et al., 2015; Karwa & Vadhan, 2018; Bun et al., 2019; Kamath et al., 2019; Biswas et al., 2020; Kamath et al., 2020; Acharya et al., 2021; Lalanne, 2023; Aden-Ali et al., 2021; Cai et al., 2021; Brown et al., 2021; Cai et al., 2021; Kamath et al., 2022a; Lalanne et al., 2023a;b; Lalanne & Gadat, 2024; Singhal, 2024; Kamath et al., 2025; 2022b). However, achieving privacy guarantees without significantly hampering the efficiency of the models remains challenging, particularly when dealing with complex distributional objectives such as those based on Wasserstein distances.

Besides, optimal transport, and in particular the Wasserstein distance, has become increasingly important in machine learning due to its ability to capture meaningful geometric relationships between probability distributions, enabling a wide range of applications from generative modeling to domain adaptation and fairness. Given two measures of probability $P$ and $Q$ in $\mathbb{R}^d$ equipped with its Borel $\sigma$-algebra, the Wasserstein distance $W_p(P,Q)$ is defined as the $p^{th}$ root of the cost associated to the optimal transport plan, i.e., $W_p(P,Q) = \left(\inf_{\pi \in \Pi(P,Q)} \int_{\mathbb{R}^d} \|x-y\|_2^p d\pi(x,y)\right)^{1/p}$, where $\Pi(P,Q)$ represents the set of measures of probability in the product space with marginals $P$ and $Q$. Except mentioned otherwise, this article will look at the specific case of $W_2$. Its straightforward geometric interpretation makes it an effective tool for comparing distributions even when the supports do not align, offering a significant advantage over other widely used metrics and divergences. Thus, it has been successfully applied in a number of areas, including generative models (Arjovsky et al., 2017), representation learning (Tolstikhin et al., 2018), domain adaptation (Courty et al., 2017) and fairness (Gordaliza et al., 2019; Risser et al., 2022; De Lara et al., 2024; Jiang et al., 2020; Chzhen et al., 2020; Gaucher et al., 2023).

To tackle the curse of dimensionality in its computations, two main alternatives have been explored, namely, entropic regularization, as proposed by Cuturi (2013), and leveraging the Wasserstein distance between one-dimensional projections.

Indeed, denoting by, for any measure of probability $P$ on $\mathbb{R}$, $F_P$ its cumulative distribution function (CDF) and $F_P^{-1}$ its quantile function, which is defined as the generalized inverse of $F_P$, then the Wasserstein distance satisfies $W_2(P,Q) = \|F_P^{-1} - F_Q^{-1}\|_{L^2}$ for any measures of probability $P, Q$ on $\mathbb{R}$. This perspective has inspired a variety of distance surrogates that reduce to one-dimensional projections. In this work, we center our attention on the sliced Wasserstein distance (Rabin et al., 2011; Bonneel et al., 2015), defined as $SW_2(P,Q) = \left(\int_{\mathbb{S}^{d-1}} W_2^2(\mathrm{Pr}_\vartheta \#P, \mathrm{Pr}_\vartheta \#Q) d\rho(\vartheta)\right)^{1/2}$, where $\#$ is the *push forward* operation of a measure by a measurable mapping, $\mathrm{Pr}_\vartheta$ the projection along the direction of $\vartheta$, and $\rho$ denotes the uniform measure on the unit sphere $\mathbb{S}^{d-1}$. Note that the integral on the sphere may be approximated by Monte Carlo methods. A substantial body of research has demonstrated the effectiveness of the sliced Wasserstein distance as a discrepancy measure for generative modeling (Deshpande et al., 2018; Wu et al., 2019), representation learning (Kolouri et al., 2019), domain adaptation (Lee et al., 2019) and fairness (Risser et al., 2022).

## 1.1 Contributions

In this work, we address the challenge of optimizing objectives which depend on sliced Wasserstein distances between empirical measures. Assume that we have samples $\mathbf{X} = (x_1, \ldots, x_n) \in \mathcal{X}^n$, $\mathbf{Z} = (z_1, \ldots, z_m) \in \mathcal{Z}^m$, and denote by $P_\mathbf{X}$ and $P_\mathbf{Z}$ the empirical distributions associated with $\mathbf{X}$ and $\mathbf{Z}$. Assuming that we have a vector of trainable weights $\theta$ that parametrizes two general functions $g_\theta : \mathcal{X} \to \mathbb{R}^d$ and $h_\theta : \mathcal{Z} \to \mathbb{R}^d$, we want to optimize for the function

$$\theta \mapsto W_2^2(g_\theta \# P_\mathbf{X}, h_\theta \# P_\mathbf{Z}), \text{ if } d = 1, \quad \text{ or } \quad \theta \mapsto SW_2^2(g_\theta \# P_\mathbf{X}, h_\theta \# P_\mathbf{Z}), \text{ if } d > 1, \tag{1}$$

under differential privacy guarantees. Our main theoretical advance lies in carefully analyzing the *sensitivity* of Wasserstein gradients (the gradients of the previous function) to individual data points. This general contribution can be split as follows.

**1) Sensitivity analysis with $1/n$ scaling.** Although $W_2^2$ lacks the usual finite-sum structure (e.g., $\frac{1}{n}\sum_i \ell(g_\theta(x_i))$), we prove that its gradient admits a decomposition favorable for privacy analysis and compatible with standard autodiff frameworks (Abadi et al., 2015; Paszke et al., 2019; Bradbury et al., 2018).

In Section 4, we show that the gradient's $\ell_2$-sensitivity decays as $O(1/n)$, allowing the use of classical DP mechanisms with vanishing noise cost.

**2) A practical deep-learning recipe.** While DP analyses typically assume bounded data-dependent quantities, in practice clipping is required (Abadi et al., 2016), often introducing bias (Kamath et al., 2025). We show in Section 5 that clipping and privacy accounting (Abadi et al., 2016; Dong et al., 2022) extend naturally to Wasserstein gradients. This yields a scalable, DP-compatible training framework that integrates seamlessly with standard optimizers.

**3) Novel applications to distributional learning under DP.** To the best of our knowledge, our framework is the first to enable end-to-end learning with sliced Wasserstein losses under differential privacy for both source and target samples. We illustrate its breadth through two applications:

- **Private sliced Wasserstein autoencoders (Section 6)**: the first DP procedure for training SWAEs, enabling privacy-preserving representation learning with generative capabilities;
- **Fairness via private in-processing (Section 7)**: a new approach to bias mitigation based on private Wasserstein penalties between conditional distributions. This provides, to the best of our knowledge, the first methods for fair and private *multivariate regression* and *representation learning*.

## 1.2 Related Work

**Differential Privacy and Optimal Transport.** Our work directly compares to the work of Rakotomamonjy & Ralaivola (2021), which extends the ideas of Harder et al. (2021), originally applied to the Maximum Mean Discrepancy (MMD), to the sliced Wasserstein loss. Their work establishes privacy guarantees for the value of the sliced Wasserstein distance, based on the use of the post-processing lemma. However, their privacy guarantees are insufficient for training models privately, except in simple scenarios such as the generative model proposed by Harder et al. (2021). Indeed, their technique requires private data to be static (as opposed to trainable), which prevents its use when the Wasserstein loss appears downstream in the learning process. In contrast, our work is significantly broader in scope, and adapts to a wider range of problems, as discussed in Remark 4.5. A more detailed comparison with Rakotomamonjy & Ralaivola (2021) is provided later in the article, both in terms of applicability of the methods and numerical results when both are applicable. Liu et al. (2025) follows the same line of work as Rakotomamonjy & Ralaivola (2021), extending their methodology to an alternative definition of the sliced Wasserstein distance. In a different vein, other existing works develop task-specific private methodologies leveraging optimal transport. The sliced Wasserstein distance has been applied in data generation by Sebag et al. (2025) from a different approach based on gradient flows. Lê Tien et al. (2019) tackled differentially private domain adaptation with optimal transport by perturbing the optimal coupling between noisy data. Recently, Xian et al. (2024) proposed a post-processing method based on the Wasserstein barycenter of private histogram estimators of conditional densities to obtain a fair and private regressor. The private estimation of optimal transport maps was recently studied by Lalanne et al. (2025). Beyond these approaches, optimal transport has also been explored in novel privacy paradigms unrelated to our work (Pierquin et al., 2024; Kawamoto & Murakami, 2019; Yang et al., 2024a).

**Fairness in Machine Learning.** Fairness in machine learning has emerged as a critical area of research, driven by the growing recognition of its societal impact and the ethical implications of algorithmic decision-making. Additionally, regulatory frameworks such as the General Data Protection Regulation (GDPR) and the recent European AI Act[1] mandate stringent measures to identify and mitigate bias in AI systems, emphasizing the need for fair and private methodologies in machine learning. Unfairness arises when certain variables, often referred to as sensitive variables, systematically bias the behavior of an algorithm against specific groups of individuals, leading to disparate outcomes. This field of research has received growing attention over the last few years as pointed out in the following papers and references therein (Chouldechova & Roth, 2020; Dwork et al., 2012; Oneto & Chiappa, 2020; Wang et al., 2022; Barocas et al., 2023; Besse et al., 2022). The Wasserstein distance offers a compelling framework for addressing fairness, as it provides a principled way to quantify discrepancies between the distributions of different subgroups. Moreover, as stated

---

[1]https://artificialintelligenceact.eu/

first by Feldman et al. (2015), then by Le Gouic et al. (2020) or Chzhen et al. (2020), Wasserstein distance between the conditional distributions of the algorithm for each group, is the natural measure to quantify the cost of ensuring fairness of the algorithm, defined as algorithms exhibiting the same behavior for each group. Hence optimal transport based methods are commonly used to assess and mitigate distributional biases, paving the way for more equitable algorithmic decision-making. We refer, for instance, to the previously mentioned references (Chiappa et al., 2020; Gordaliza et al., 2019) and references therein.

**Differential Privacy and Fair Learning.** The interplay between fairness and differential privacy has received significant attention in recent years. A comprehensive review of this topic in decision and learning problems is provided by Fioretto et al. (2022). Within the learning framework, research has progressed in various directions. From a theoretical standpoint, despite the early work of Cummings et al. (2019) demonstrating inherent incompatibilities between exact fairness and differential privacy, Mangold et al. (2023) recently presented promising theoretical results indicating that fairness is not severely compromised by privacy in classification tasks. Another research direction has focused on studying the disparate impacts on model accuracy introduced by private training of algorithms. This phenomenon was first observed by Bagdasaryan et al. (2019) and has been extensively studied in subsequent works (Farrand et al., 2020; Tran et al., 2021; Xu et al., 2021; Esipova et al., 2023). A third line of research aims to develop models that are both private and fair. Private and fair classification models have been proposed using in-processing and post-processing techniques across various scenarios (Xu et al., 2019; Jagielski et al., 2019; Ding et al., 2020; Lowy et al., 2023; Yaghini et al., 2023; Ghoukasian & Asoodeh, 2024). A recent comparison of these works can be found in Ghoukasian & Asoodeh (2024). In the topic of fair and private regression, the only available work is the aforementioned post-processing method of Xian et al. (2024).

## 2 Differential Privacy

We briefly recall the notions of differential privacy that underpin our sensitivity analysis in Section 4. Differential privacy (Dwork et al., 2006b;a) starts with a dataset space $\mathcal{D}$ and a neighboring relation $\sim$ on $\mathcal{D}$. For $\mathbf{D}, \tilde{\mathbf{D}} \in \mathcal{D}$, we write $\mathbf{D} \sim \tilde{\mathbf{D}}$ when they differ by the data of a single individual. A randomized mechanism $\mathcal{M} : \mathcal{D} \to \mathcal{O}$ is then required to make $\mathcal{M}(\mathbf{D})$ statistically hard to distinguish from $\mathcal{M}(\tilde{\mathbf{D}})$.

The neighboring relation $\sim$ is *application specific* and is usually either the *addition / deletion* relation ($\mathbf{D}$ and $\tilde{\mathbf{D}}$ are neighbors iff one can be obtained from the other by adding or removing the data of one individual from the dataset) or the *replacement* relation ($\mathbf{D}$ and $\tilde{\mathbf{D}}$ are neighbors iff one can be obtained from the other by changing the data of one individual from either dataset). In general, it is useful to the reader to understand $\mathbf{D} \sim \tilde{\mathbf{D}}$ as : "The difference between $\mathbf{D}$ and $\tilde{\mathbf{D}}$ only comes from one individual's data".

In our paper, due to the splitting of the data into separate categories in the Wasserstein distance, and because of potential asymmetry that may arise in their treatments, we will occasionally employ modified definitions of neighboring relations, which can be encompassed within the following family, indexed by the number of classes $k \geq 1$. Note that the case $k = 1$ reduces to the usual *replacement* relation.

**Definition 2.1.** (k-end neighboring relation) Let $\mathcal{D} = \mathcal{D}_1^{n_1} \times \ldots \times \mathcal{D}_k^{n_k}$ be the set of partitioned datasets with sizes $n_1, \ldots, n_k \geq 1$. Given two datasets $\mathbf{D} = (\mathbf{D^1}, \ldots, \mathbf{D^k})$, $\tilde{\mathbf{D}} = (\tilde{\mathbf{D}^1}, \ldots, \tilde{\mathbf{D}^k}) \in \mathcal{D}$, we say that $\mathbf{D} \sim_k \tilde{\mathbf{D}}$ if there exists an index $j \in [k] := \{1, \ldots, k\}$ such that $\mathbf{D^i} = (d_1^i, \ldots, d_{n_i}^i)$ and $\tilde{\mathbf{D}^i} = (\tilde{d}_1^i, \ldots, \tilde{d}_{n_i}^i)$ coincide up to a permutation of the elements if $i \neq j$, and up to a permutation and a replacement of one of the $d_l^i$'s by any element in $\mathcal{D}_i$ if $i = j$.

The historic definition of differential privacy (Dwork et al., 2006b;a; Dwork, 2006) with $(\varepsilon, \delta)$ reads:

**Definition 2.2** ($(\varepsilon, \delta)$-DP (Dwork et al., 2006a)). A randomized mechanism $M : \mathcal{D} \to \mathcal{O}$ is $(\varepsilon, \delta)$-differentially private ($(\varepsilon, \delta)$-DP) if $\forall \mathbf{D} \sim \tilde{\mathbf{D}}$, and $\forall$ measurable $S \subset \mathcal{O}$,

$$\mathbb{P}\left(M(\mathbf{D}) \in S\right) \leq e^{\varepsilon} \mathbb{P}\left(M(\tilde{\mathbf{D}}) \in S\right) + \delta \, ,$$

where the randomness is taken on $M$ only.

A ubiquitous building block for building private mechanisms is the so-called *Gaussian mechanism* which consists in adding independent Gaussian noise to the output of a deterministic mapping. Quantifying the

privacy of this (now randomized) mechanism then boils down to controlling the variations of the deterministic mapping on neighboring datasets, as formalized in the following lemma.

**Lemma 2.3** (Privacy of the Gaussian mechanism (Corollary of Theorem 2.7, Corollary 3.3 and Corollary 2.13 in Dong et al. (2022))). *Given a deterministic function $h$ mapping a dataset to a quantity in $\mathbb{R}^d$, one can define the $\ell_2$-sensitivity of $h$ as*

$$\Delta_2(h) := \sup_{\mathbf{D} \sim \tilde{\mathbf{D}}} \left\| h(\mathbf{D}) - h(\tilde{\mathbf{D}}) \right\|_2 .$$

*When this quantity is finite, for any $\sigma > 0$, the Gaussian mechanism defined as $\mathbf{D} \mapsto h(\mathbf{D}) + \sigma \mathcal{N}(0, I_d)$, is $(\varepsilon, \delta(\varepsilon))$-DP for any $\varepsilon \geq 0$ where, by noting $\mu = \frac{\Delta_2(h)}{\sigma}$, $\delta(\varepsilon) = \Phi\left(-\frac{\varepsilon}{\mu} + \frac{\mu}{2}\right) - e^\varepsilon \Phi\left(-\frac{\varepsilon}{\mu} - \frac{\mu}{2}\right)$, where $\Phi$ denotes the standard normal CDF.*

In particular, Section 4 proves that the $\ell_2$-sensitivity of the Wasserstein gradient decreases as $O(1/n)$, where $n$ is the sample size, motivating the methods presented in this article.

Finally, this article will at times use the notion of *Gaussian Differential Privacy* (GDP). Informally, given $\mu > 0$, a mechanism $M$ is said to be $\mu$-GDP if it makes $M(\mathbf{D})$ at least as hard to discriminate from $M(\tilde{\mathbf{D}})$ as $\mathcal{N}(0, 1)$ is to discriminate from $\mathcal{N}(\mu, 1)$, for any $\mathbf{D} \sim \tilde{\mathbf{D}}$. We refer to Dong et al. (2022) for a formal definition and the links to $(\varepsilon, \delta)$-DP.

# 3 Wasserstein Gradients

To analyze privacy, we require explicit formulas for the gradients of the Wasserstein distance between empirical distributions. Let $\mathbf{U} = (U_1, \ldots, U_n) \in \mathbb{R}^n$ and $\mathbf{V} = (V_1, \ldots, V_m) \in \mathbb{R}^m$, and define the one-dimensional empirical measures $P_\mathbf{U} = \frac{1}{n} \sum_{i=1}^n \delta_{U_i}$ and $P_\mathbf{V} = \frac{1}{m} \sum_{j=1}^m \delta_{V_j}$. We denote by $U_{(1)} \leq \cdots \leq U_{(n)}$ and $V_{(1)} \leq \cdots \leq V_{(m)}$ the corresponding order statistics. Then, the squared Wasserstein distance admits the following weighted quadratic form.

**Proposition 3.1.** *With $R_{i,j} = \lambda\left(\left(\frac{i-1}{n}, \frac{i}{n}\right] \cap \left(\frac{j-1}{m}, \frac{j}{m}\right]\right)$ where $\lambda$ denotes the Lebesgue measure, and rank permutations $\sigma, \tau$ such that $U_i = U_{(\sigma(i))}$ for each $i \in [n]$ and $V_j = V_{(\tau(j))}$ for each $j \in [m]$, one has*

$$W_2^2(P_\mathbf{U}, P_\mathbf{V}) = \sum_{i=1}^n \sum_{j=1}^m R_{\sigma(i), \tau(j)} (U_i - V_j)^2.$$

*Proof.* See Appendix G.1. $\qquad\square$

Proposition 3.1 expresses the Wasserstein distance as a weighted sum of squared differences, with weights $R_{\sigma(i), \tau(j)}$, that depend only on the rank permutations. Thus, the partial derivative with respect to $U_i$, is well defined, as long as its rank $\sigma(i)$ remains unchanged in a neighborhood of $U_i$.

**Proposition 3.2.** *With all the previous definitions, $W_2^2(P_\mathbf{U}, P_\mathbf{V})$ is differentiable as a function of $\mathbf{U}$ in the set of points verifying $U_{(1)} < \ldots < U_{(n)}$, and $\nabla_U W_2^2(P_\mathbf{U}, P_\mathbf{V}) = \left(2 \sum_{j=1}^m R_{\sigma(i), \tau(j)} (U_i - V_j)\right)_{i \in [n]}$. Similarly, it is differentiable as a function of $\mathbf{V}$ in the set of points verifying $V_{(1)} < \ldots < V_{(m)}$, and $\nabla_V W_2^2(P_\mathbf{U}, P_\mathbf{V}) = \left(2 \sum_{i=1}^n R_{\sigma(i), \tau(j)} (V_j - U_i)\right)_{j \in [m]}$.*

*Proof.* Simple derivation in Proposition 3.1. $\qquad\square$

From a practical viewpoint, non-differentiability when points coincide is benign: the rank permutation is not unique, but any consistent choice yields valid updates, in line with prior practice (Deshpande et al., 2018; Kolouri et al., 2019). We therefore follow earlier work in referring to these updates as "gradients"; importantly, our privacy guarantees that are presented in the next section remain valid even at such points.

# 4   A Private Surrogate for Wasserstein Gradients

We now present our main theoretical result (see Theorem 4.1): a tight upper bound on the sensitivity of Wasserstein gradients, showing that it decays as $O(1/n)$. This scaling directly enables the use of the Gaussian mechanism (Lemma 2.3) to obtain differential privacy at vanishing cost.

For clarity, we present the one-dimensional case $g_\theta : \mathcal{X} \to \mathbb{R}$ and $h_\theta : \mathcal{Z} \to \mathbb{R}$, the multidimensional extension being given in Remark 4.3. Throughout this section, we make no assumptions on $\mathcal{X}$ and $\mathcal{Z}$. We only assume that $\Theta \subseteq \mathbb{R}^r$, where $r$ is the number of parameters, and that, for every $x \in \mathcal{X}$ and $z \in \mathcal{Z}$, the maps $\theta \mapsto g_\theta(x)$ and $\theta \mapsto h_\theta(z)$ are differentiable on $\Theta$. Then, combining Proposition 3.2 with the chain rule yields

$$\nabla_\theta W_2^2(g_\theta \# P_{\mathbf{X}}, h_\theta \# P_{\mathbf{Z}}) = 2 \sum_{i=1}^{n} \sum_{j=1}^{m} R_{\sigma(i),\tau(j)}(g_\theta(x_i) - h_\theta(z_j)) \nabla_\theta g_\theta(x_i)$$

$$+ 2 \sum_{i=1}^{n} \sum_{j=1}^{m} R_{\sigma(i),\tau(j)}(h_\theta(z_j) - g_\theta(x_i)) \nabla_\theta h_\theta(z_j) .$$

This decomposition enables the following sensitivity analysis:

**Theorem 4.1.** *With all the previous notation, assume that there exist constants $M, L_1, L_2 \geq 0$ such that for each $\theta \in \Theta$, $x \in \mathcal{X}$ and $z \in \mathcal{Z}$,*

$$(i) \ |g_\theta(x)| \leq M, \ |h_\theta(z)| \leq M , \qquad (ii) \ \|\nabla_\theta g_\theta(x)\|_2 \leq L_1, \ \|\nabla_\theta h_\theta(z)\|_2 \leq L_2 .$$

*Under the neighboring relation $\sim_1$ in $\mathcal{D} = \mathcal{X}^n$, if we define $\Phi(\mathbf{X}) = \nabla_\theta W_2^2(g_\theta \# P_{\mathbf{X}}, h_\theta \# P_{\mathbf{Z}})$, then*

$$\Delta_2(\Phi) \leq 4M \frac{3L_1 + L_2}{n} . \tag{2}$$

*Consequently, under the neighboring relation $\sim_2$ in $\mathcal{D} = \mathcal{X}^n \times \mathcal{Z}^m$, if we define $\Psi(\mathbf{X}, \mathbf{Z}) = \nabla_\theta W_2^2(g_\theta \# P_{\mathbf{X}}, h_\theta \# P_{\mathbf{Z}})$, then*

$$\Delta_2(\Psi) \leq 4M \max\left\{ \frac{3L_1 + L_2}{n}, \frac{L_1 + 3L_2}{m} \right\} . \tag{3}$$

*Proof.* See Appendix G.2. □

**Intuition.**   The proof works because, although replacing one observation may modify many ranks, this modification has a very rigid structure once the samples are reordered. Thanks to the explicit formula of Section 3, the Wasserstein gradient can be written as a weighted sum whose coefficients depend only on the rank permutations. After changing one point, all the other observations are unchanged as values; only their positions in the ordering may shift relative to the modified point. This means that the difference between the two gradients can be decomposed into a small number of terms: one term coming from the modified sample itself, and permutation terms coming from the shift of the rank weights. The crucial observation is that, after reindexing by order statistics, these permutation terms all have the same sign, so they do not accumulate chaotically; instead, they collapse into a mass of $O(1/n)$.

For any $\mu > 0$, define $\sigma_\mu = \Delta/\mu$, where $\Delta$ denotes the right-hand side of (2), if only $\mathbf{X}$ is private, or (3), if both $\mathbf{X}$ and $\mathbf{Z}$ are private. Under the assumptions of Theorem 4.1, if we define, with a slight abuse,

$$\hat{\nabla}_\theta W_2^2 := \nabla_\theta W_2^2 + \sigma_\mu \mathcal{N}(0, I) , \tag{4}$$

then Theorem 4.1 together with Lemma 2.3 ensure that for every $\varepsilon \geq 0$, the mechanism is $(\varepsilon, \delta(\varepsilon))$-DP, where $\delta(\varepsilon)$ is given in Lemma 2.3. Furthermore, the mechanism admits the following high-probability utility bound:

**Corollary 4.2.** *With the above notation, if $r$ denotes the dimension of the parameter space, for any $\delta \in (0, 1)$, with probability at least $1 - \delta$,*

$$\|\hat{\nabla}_\theta W_2^2 - \nabla_\theta W_2^2\|_2^2 \leq \sigma_\mu^2 \left( r + 2\sqrt{r \log(\tfrac{1}{\delta})} + 2 \log(\tfrac{1}{\delta}) \right).$$

*Proof.* Follows from the exponential inequality in Lemma 1 of Laurent & Massart (2000). □

*Remark* 4.3 (**Extension to arbitrary dimension**). Theorem A.1 in Appendix A extends Theorem 4.1 to the multidimensional setting $g_\theta(\cdot), h_\theta(\cdot) \in \mathbb{R}^d$, using the sliced Wasserstein distance $SW_2^2$ or its Monte Carlo approximation. The conclusions of Theorem A.1 remain identical to Theorem 4.1, but now require uniform control over $\vartheta \in \mathbb{S}^{d-1}$ of the composite functions $\mathrm{Pr}_\vartheta \circ g_\theta, \mathrm{Pr}_\vartheta \circ h_\theta$. To achieve this, Assumption (i) is replaced by $\|g_\theta(x)\|_2 \leq M$, $\|h_\theta(x)\|_2 \leq M$ and Assumption (ii) by $\|\mathcal{J}_\theta g_\theta(x)\|_2 \leq L_1$, $\|\mathcal{J}_\theta h_\theta(z)\|_2 \leq L_2$, where $\|\cdot\|_2$ denotes here the spectral norm of a matrix. Note that, if $d = 1$, $\mathcal{J}_\theta g_\theta = \nabla_\theta g_\theta$ and the spectral norm coincides with the euclidean norm.

*Remark* 4.4 (**On the assumptions**). Conditions (i)–(ii) are standard boundedness and Lipschitz assumptions. Assumption (ii) is verified as soon as $g_\theta$ and $h_\theta$ are Lipschitz with respect to the parameter $\theta$. When they do not hold exactly, clipping techniques (Abadi et al., 2016) ensure they can be enforced in practice (see Section 5 for a detailed discussion on a technique that we introduce, called *inner-clipping*, that plays the same role as regular clipping in vanilla DPSGD but that is compatible with SW losses).

*Remark* 4.5 (**Comparison to the baseline** (Rakotomamonjy & Ralaivola, 2021)). Assuming $h_\theta = I_d$, by the chain rule, and with a slight abuse of notation,

$$\nabla_\theta SW_2^2(g_\theta(\mathbf{X}), \mathbf{Z}) = \nabla_{g_\theta(\mathbf{X})} SW_2^2(g_\theta(\mathbf{X}), \mathbf{Z}) \times \mathcal{J}_\theta g_\theta(\mathbf{X}) .$$

Rakotomamonjy & Ralaivola (2021) privatize only the first term, so their guarantees apply only when $g_\theta$ does not act on private data. Our bound covers both terms, extending their approach to genuine end-to-end private training. A detailed comparison is provided in Appendix D, both from a conceptual perspective—highlighting the broader applicability of our technique—and from a numerical perspective, comparing results in a setting where both methods are applicable.

*Remark* 4.6 (Other orders). The proof of Theorem 4.1 crucially exploits the quadratic structure of $W_2^2$. The same arguments do not extend, in general, to Wasserstein losses of other orders. As shown by the counterexample in Appendix F, for $W_p$ with general $p \geq 1$, even under the same boundedness assumptions, the sensitivity of the gradient need not decrease with the sample size. In other words, the favorable privacy scaling established here is not a generic property of Wasserstein objectives, but a consequence of the quadratic transport cost. This limitation is somewhat unfortunate from an applications viewpoint, since several important models rely on non-quadratic transport losses, most notably Wasserstein GANs (Arjovsky et al., 2017), which are based on the 1-Wasserstein distance.

## 5 A Framework for Deep Learning

This section explains how to quantify the privacy guarantees after $T$ steps of any first-order optimization algorithm (e.g., SGD) minimizing a sliced Wasserstein loss when using (4) as a substitute for the usual gradients, particularly in a deep learning framework where it is not possible to guarantee a priori that gradients and activations are bounded, and where one typically needs to run many iterations in a batched setting.

### 5.1 Inner-Clipping of the Gradients

Directly applying Theorem 4.1 to general deep learning models is typically infeasible, as the required boundedness conditions are not satisfied. As a solution, we propose to introduce three hyperparameters $M \geq 0$, $L_1 \geq 0$, and $L_2 \geq 0$, and to use as a proxy for $\nabla_\theta W_2^2$ the following quantity:

$$\nabla_\theta^{M,L_1,L_2} W_2^2 = 2 \sum_{i=1}^n \sum_{j=1}^m R_{\sigma(i),\tau(j)}(U_i - V_j) \mathrm{Proj}_{L_1}(\nabla_\theta g_\theta(x_i)) + 2 \sum_{i=1}^n \sum_{j=1}^m R_{\sigma(i),\tau(j)}(V_j - U_i) \mathrm{Proj}_{L_2}(\nabla_\theta h_\theta(z_j))$$

(5)

where for all $i$ and $j$, we have $U_i = \mathrm{Proj}_M(g_\theta(x_i))$, $V_j = \mathrm{Proj}_M(h_\theta(z_j))$, and $\sigma, \tau$ are defined as in Section 3. This technique is known as *clipping* and was historically introduced as a preprocessing of the gradients for problems with a finite-sum structure (Abadi et al., 2016). For Wasserstein gradients, note that we also

need to clip the *activations*. Now, Theorem 4.1 applies and one may add noise to this quantity to make it private with the Gaussian mechanism.

*Remark* 5.1 (**Computational considerations**)*.* This approach requires access to per-sample gradients (or per-sample Jacobians in the case of $SW_2^2$, see Remark A.2) at the layer where the constraint is applied, which introduces overhead, but remains compatible with modern autodiff libraries. A broader discussion on this topic is provided in Appendix E including running-times comparisons.

## 5.2 Amplification by Subsampling

In deep-learning, sub-sampling is often a necessity because of the size of the datasets. With differential privacy, it allows to leverage a property called *privacy amplification by subsampling*. Since such property varies depending on the neighboring relation, we formalize it in the following lemma with the conventions of this article.

**Lemma 5.2** (Privacy amplification by subsampling)**.** *Let $n_1' \leq n_1, \ldots, n_k' \leq n_k$. If a mechanism $M_{batch}$ is $(\varepsilon, \delta)$-DP on $\mathcal{D}_1^{n_1'} \times \ldots \times \mathcal{D}_k^{n_k'}$, the mechanism $M$ that (i) selects $n_i'$ among the $n_i$ points in each category without replacement, and then (ii) applies $M_{batch}$ to the sampled dataset, is $(\varepsilon', \delta')$-DP on $\mathcal{D}_1^{n_1} \times \ldots \times \mathcal{D}_k^{n_k}$ where $\varepsilon' = \ln\left(1 + p\left(e^\varepsilon - 1\right)\right), \ \delta' = p\delta$ and $p = \max\left(\frac{n_1'}{n_1}, \ldots, \frac{n_k'}{n_k}\right)$.*

*Proof.* See Appendix G.3. $\square$

## 5.3 Privacy Accounting

In the influential article of Abadi et al. (2016), the authors introduce the *moment accountant* method, a framework for quantifying the privacy of a composition of subsampled Gaussian mechanisms. We now detail why similar methods (Dong et al., 2022) are applicable to Wasserstein gradients.

Since our neighboring relation is based on the replacement relation and since we use subsampling with fixed batch size and without replacement, the classical moment accountant method (Abadi et al., 2016) does not apply. We thus turn to the accounting techniques of Dong et al. (2022) that build on the theory of $f$-differential privacy and that are more suited to this scenario. Using the notation of Dong et al. (2022), let $\Delta$ be an upper bound on the sensitivity of $\nabla_\theta^{M,L_1,L_2} W_2^2$ (which is controlled by Theorem 4.1). Then, $\nabla_\theta^{M,L_1,L_2} W_2^2 + \sigma N(0, I)$ is $\frac{\Delta}{\sigma}$-GDP (for *Gaussian Differential Privacy*) ignoring the subsampling step. In order to account for subsampling, one would like to apply Theorem 4.2 in Dong et al. (2022). This is not possible since this article uses a different neighboring relation and a different form of subsampling. However, we can notice that we can substitute Lemma 4.4 in the proof of Theorem 4.2 in Dong et al. (2022) by our Lemma 5.2 and the rest of the proof follows. We thus get that the overall procedure described by Algorithm 1 is $C_p(G_{(\sigma/\Delta)^{-1}})^{\otimes T}$-DP where $p = \max(n_1'/n_1, \ldots, n_k'/n_k)$ with the formalism of $f$-differential privacy (Dong et al., 2022).

---

**Algorithm 1** Sequential Computation of Subsampled Wasserstein Noisy Gradients

---
**for** $t = 1$ to $T$ **do**
    Select $n_i'$ among the $n_i$ points in each category without replacement.
    Compute $\hat{\nabla}_\theta W_2^2 := \nabla_\theta^{M,L_1,L_2} W_2^2 + \sigma \mathcal{N}(0, I)$ on the subsampled dataset.
    Publish $\hat{\nabla}_\theta W_2^2$.
    Wait for the optimizer to update $\theta$.
**end for**

---

Finally, our approach aligns with the privacy accounting framework in the asymptotic regime described in Section 5.2 of (Dong et al., 2022). In our experiments, we implement this privacy accountant using the (Yousefpour et al., 2021) library. Note that this accountant can be replaced by any accountant, tailored for fixed size batch sampling without replacement and with the replacement neighboring relation.

| Quantity | Assumption enforced | Practical implementation |
|---|---|---|
| $M$ | Bounded outputs of $g_\theta$ and $h_\theta$ | Activation clipping |
| $L_1$ | Bounded derivative of $g_\theta$ w.r.t. $\theta$ | Per-sample gradient/Jacobian clipping on the $g_\theta$ branch |
| $L_2$ | Bounded derivative of $h_\theta$ w.r.t. $\theta$ | Per-sample gradient/Jacobian clipping on the $h_\theta$ branch |

Table 1: From theoretical clipping parameters to implementation in Section 5.1.

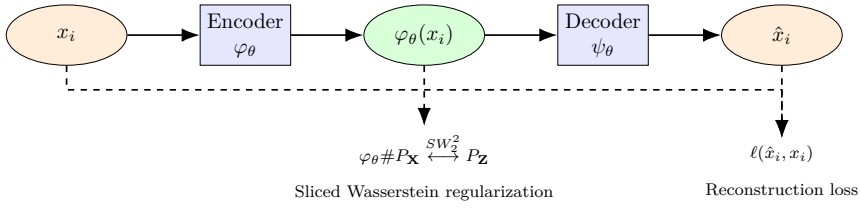

Total loss: $\mathscr{L}_\alpha = (1-\alpha) \cdot \frac{1}{n}\sum_i \ell(\hat{x}_i, x_i) + \alpha \cdot SW_2^2(\varphi_\theta \# P_{\mathbf{X}}, P_{\mathbf{Z}})$

Figure 1: Sliced Wasserstein autoencoder with privacy-aware training. The encoder maps input $x_i$ to a latent code $\varphi_\theta(x_i)$, encouraged to match the prior $P_{\mathbf{Z}}$. The decoder reconstructs $\hat{x}_i$. The loss combines reconstruction and sliced Wasserstein distance in latent space.

# 6 Private Sliced Wasserstein Autoencoders

Building on our analysis, we present what is, to the best of our knowledge, the first differentially private training procedure for the sliced Wasserstein autoencoders of Kolouri et al. (2019). [2]

## 6.1 General Method

Given a private dataset $\mathbf{X} \in \mathcal{X}^n$, the goal is to learn an encoder $\varphi_{\theta_a}$ and a decoder $\psi_{\theta_b}$ by minimizing a reconstruction loss $\ell$, regularized with a sliced Wasserstein penalty in the latent space. This encourages the encoded representations to match a non-private random sample $\mathbf{Z}$ from a prescribed distribution $Q$ on $\mathbb{R}^{d_0}$. Denoting $\theta = (\theta_a, \theta_b)$, $\varphi_\theta = \varphi_{\theta_a}$ and $\psi_\theta = \psi_{\theta_b}$, we minimize

$$\mathscr{L}_\alpha(\theta) = (1-\alpha)\frac{1}{n}\sum_{i=1}^n \ell(\psi_\theta(\varphi_\theta(x_i)), x_i) + \alpha\, SW_2^2\big(\varphi_\theta \# P_{\mathbf{X}},\, P_{\mathbf{Z}}\big)\,, \tag{6}$$

where $\alpha \in [0,1]$ is the regularization parameter (see Figure 1). While regular DPSGD allows privately optimizing the first term of this sum, our method allows optimizing the second (and hence the sum).

## 6.2 Private Training Loop

We now control the sensitivity of the gradients. For the first term in Equation (6), the finite-sum structure carries to the gradient: if $\|\nabla_\theta \ell(\psi_\theta(\varphi_\theta(x_i)), x_i)\|_2 \le C$ for all $x_i \in \mathcal{X}$, the sensitivity is bounded by $2C/n$ under $\sim_1$. For the second term, if $\varphi_\theta$ satisfies Assumptions (i)–(ii) of Theorem A.1,

$$\Delta_2\big(\nabla_\theta \mathscr{L}_\alpha(\theta)\big) \;\le\; (1-\alpha)\frac{2C}{n} \;+\; \alpha \cdot \frac{12\,M\,L}{n}\,.$$

This benefits from large $n$, enabling private gradients with minimal noise. As in Section 5, the training uses:

- **Clipping:** If assumptions fail, we clip per-example reconstruction gradients by $C > 0$, and apply inner clipping to the sliced Wasserstein part using $M > 0$ for the *encoder output* and $L > 0$ for the spectral norm of the encoder Jacobian;
- **Subsampling:** We estimate gradients over batches of size $n'$, replacing $n$ by $n'$ in the above sensitivity.

---

[2]The code required to reproduce the experiments in this paper is available at `https://github.com/rvitores/LearningDPslicedWasserstein`.

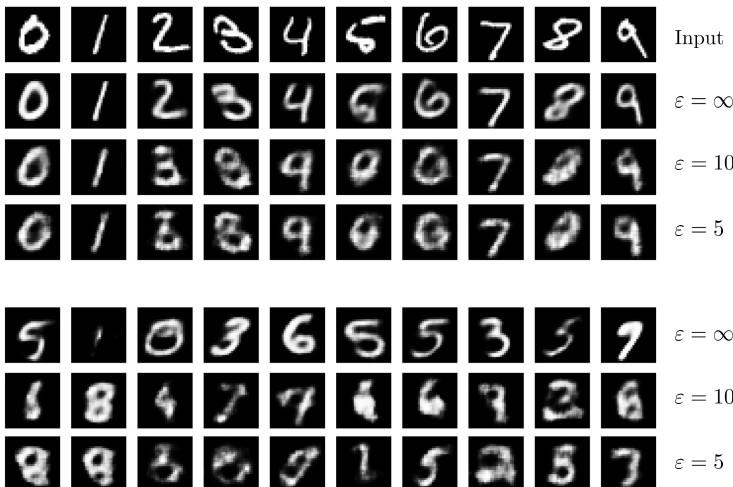

Figure 2: Benchmarking the capabilities of a $(\varepsilon, \delta)$-DP sliced Wasserstein autoencoder on MNIST with $\delta = 10^{-5}$ and varying $\varepsilon$. **Top:** Reconstructed digits from the test dataset. The first row shows the earliest sample of each digit (0–9) in the test set, followed by reconstructions. **Bottom:** Generated samples from the same model by decoding noise from the latent space.

### 6.3 Generative Modeling as a Byproduct — Comparison to the Baseline

A byproduct of SWAEs is their ability to generate data resembling the training distribution: matching $\varphi_\theta \# P_\mathbf{X}$ to a prior and decoding fresh prior samples yields synthetic data. Since training is differentially private end-to-end, this yields a privacy-preserving data generator.

### 6.4 Data and Results

We use $\mathbf{X} \in (\mathbb{R}^{28 \times 28})^{60000}$ from MNIST or Fashion-MNIST, and $\mathbf{Z}$ as i.i.d. samples from the uniform distribution on the unit ball in $\mathbb{R}^{d_0}$ with $d_0 = 6$. Encoder/decoder architectures are in Table 4 (Appendix B). We train with binary cross-entropy and $\alpha = 0.1$, using private gradients on random batches of size $n' = 600$, and ADAM updates (Kingma & Ba, 2015). Clipping values: $M = 1.5$ for the encoder output, $L = \sqrt{6}$ for the spectral norm of the last-layer encoder Jacobian (naive approach of Remark A.2), and $C = 1$ for per-example reconstruction gradients. We run 5000 iterations, adding noise each step to reach $(\varepsilon, \delta)$-DP over all iterations. The $SW_2$ Monte Carlo approximation uses 100 random projection directions. Reconstructions and embeddings for MNIST are shown in Figures 2 and 3. The latent distributional constraints (including support shape) are enforced while preserving meaningful reconstructions. This structured latent representation also enables meaningful interpolations between samples, as illustrated in Figure 7 (Appendix B).

For generative modeling, examples for MNIST appear in Figure 2; analogous Fashion-MNIST results are given in Appendix B. Within Wasserstein-based approaches, our results are competitive; cf. Sebag et al. (2025) comparing private sliced Wasserstein flows to Rakotomamonjy & Ralaivola (2021). Broader comparisons to task-specific methods are presented in Tables 2 and 3 in Appendix B. While our approach does not match specialized state-of-the-art methods for DP data generation (e.g., DP diffusion), this is not its primary goal: the contribution here is to provide, for the first time, a differentially private training procedure for SWAEs. Despite this shift in focus, our results remain competitive with recent alternatives and demonstrate strong downstream utility, highlighting the value of SWAE-style representation learning under privacy constraints. In practice, we observe that generation yields adequate samples for $\varepsilon$ bigger than 10, while reconstruction can be pushed to $\varepsilon = 5$.

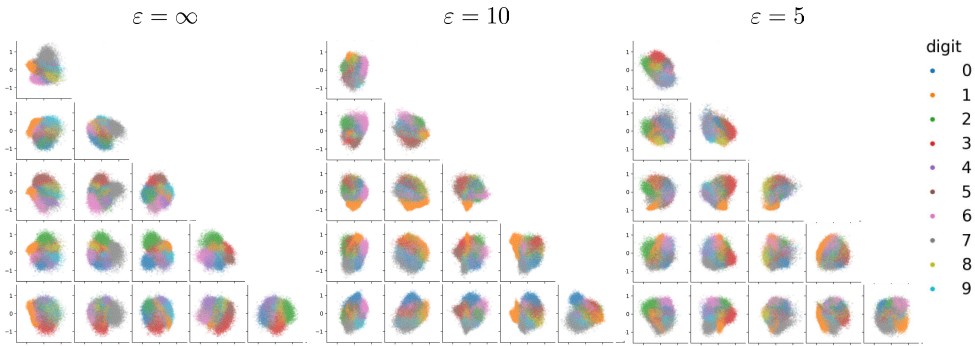

Figure 3: Encoded latent space MNIST samples for the autoencoder, trained under $(\varepsilon, \delta)$-DP, with $\delta = 10^{-5}$ and varying values of $\varepsilon$.

## 7  Fairness via Private In-Processing

This section shows how our framework enables training models that are simultaneously fair and differentially private.

### 7.1  General Method

Given a dataset $\mathbf{D}$ with samples $(x_i, a_i, y_i)$ or $(x_i, a_i)$, where $x_i$ are non-sensitive attributes, $a_i \in \{0, 1\}$ is a binary sensitive attribute, and $y_i$ is a response variable (in supervised settings), standard learning algorithms minimize the empirical risk

$$\min_\theta \mathscr{L}(\theta) := \frac{1}{n} \sum_{i=1}^n \ell(g_\theta(x_i)),$$

with $\ell(g_\theta(x_i)) = \ell(g_\theta(x_i), y_i)$ in supervised tasks and $\ell(g_\theta(x_i), x_i)$ in unsupervised ones. In our framework, fairness notions are enforced by adding suitable Wasserstein penalties to $\mathscr{L}(\theta)$, while keeping sensitivity bounded for DP guarantees.

Among fairness notions, *statistical parity* requires decisions to be independent of $A$: $\mathcal{L}(g_\theta(X) \mid A = 0) = \mathcal{L}(g_\theta(X) \mid A = 1)$. Writing $\mathbf{X}_j = \{x_i : a_i = j\}$ with $n_j = |\mathbf{X}_j|$, we minimize

$$\mathscr{L}_\alpha^{SP}(\theta) = (1 - \alpha)\mathscr{L}(\theta) + \alpha \, SW_2^2(\varphi_\theta \# P_{\mathbf{X}_0}, \, \varphi_\theta \# P_{\mathbf{X}_1}),$$

where $\alpha \in [0, 1]$ controls the fairness–accuracy tradeoff, and $\varphi_\theta = g_\theta$ (output-level) or an intermediate representation. Other notions are discussed in Appendix C.

### 7.2  Training Details

If $\varphi_\theta$ verifies $\|\varphi_\theta(x)\|_2 \leq M$, $\|\mathcal{J}_\theta \varphi_\theta(x)\|_2 \leq L$ and $\|\nabla_\theta \ell(g_\theta(x))\|_2 \leq C$, Corollary C.1 ensures that the sensitivity of $\nabla_\theta \mathscr{L}_\alpha^{SP}(\theta)$ is bounded by

$$(1 - \alpha)\frac{2C}{n} + \alpha \frac{16ML}{\min\{n_0, n_1\}},$$

under the two-end neighboring relation $\sim_2$. In practice, as in Section 6, training combines clipping, subsampling and privacy accounting within the DP-SGD framework described in Section 5. Full details are given in Appendix C.

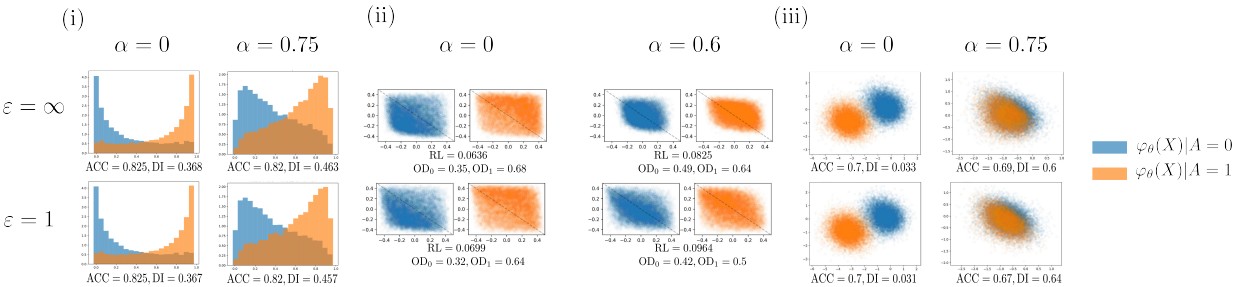

Figure 4: Distributions of $\varphi_\theta(X)$ conditioned on $A = 0$ and $A = 1$, for models trained to minimize $\mathcal{L}_\alpha^{SP}$ under $(\varepsilon, \delta)$-DP, with $\delta = 0.1/n$ and varying values of the regularization parameter $\alpha$ and the privacy budget $\varepsilon$. The representation $\varphi_\theta$ corresponds to: (i) predicted class probabilities in a classification task, (ii) predicted values in a bidimensional regression task, and (iii) bidimensional latent representations from an autoencoder. Below each figure, we display the comparison metrics explained in Section 7.3

## 7.3 Data and Results

To illustrate the versatility of our methodology for private fairness mitigation, we rely on a synthetic dataset designed to reproduce bias patterns commonly observed in supervised learning datasets, such as those in Becker & Kohavi (1996) or Hofmann (1994). The data consist of i.i.d. samples $D = \{(x_i, a_i, y_i^C, y_i)\}_{i=1}^n$, where $X$ denotes the features, $A$ the sensitive attribute, $Y^C \in \mathbb{R}^2$ a continuous response, and $Y$ a binary discretization of $Y^C$. The features $X$ are split into a core component, informative about $Y^C$, and a spurious component, informative about $A$. Because $Y$ and $A$ are positively correlated, the spurious component may bias the algorithm toward unfair decisions. The detailed construction of the dataset is given in Appendix C.

We evaluate our approach across three scenarios:

- **Private and fair binary classification:** a well-studied benchmark where the goal is to predict $Y$ from the features $X$. We train a logistic regressor with an additional Wasserstein penalty applied to the layer of predicted probabilities.

- **Private and fair multidimensional regression:** a novel extension beyond current one-dimensional approaches such as Xian et al. (2024). The goal is to predict the continuous response $Y^C$ from the features $X$. We train a two-layer neural network with squared loss and a sliced Wasserstein penalty applied to the output layer.

- **Private and fair representation learning:** a new setting where fairness is enforced directly at the representation level. We train an autoencoder with squared reconstruction loss and a sliced Wasserstein penalty applied to the latent representation.

The complete experimental setup and detailed results for each scenario are given in Appendix C. A summary of the results is given in Figure 4. For each scenario, Figure 4 shows the distribution of the fairness-penalized layer for different values of the privacy budget $\epsilon$ and the regularization parameter $\alpha$, together with some evaluation metrics. In the classification setting, ACC denotes classification accuracy and DI denotes disparate impact, a statistical parity measure defined for a binary classifier $G$ as

$$DI(G) = \frac{\mathbb{P}(G(X) = 1 | A = 0)}{\mathbb{P}(G(X) = 1 | A = 1)} \ . \tag{7}$$

The same metrics are reported in the representation-learning experiment, after training a logistic regressor on the learned representation. In the regression setting, we report the regression loss (RL) together with the number of points over the diagonal in each conditional distribution ($\text{OD}_i$, $i = 0, 1$), to aid the visual interpretation of the plot. Precise definitions of these and other evaluation metrics, together with detailed results, are given in Appendix C. Two conclusions emerge: (i) increasing $\alpha$ effectively mitigates bias through Wasserstein penalization, and (ii) adding privacy has minimal effect on fairness outcomes.

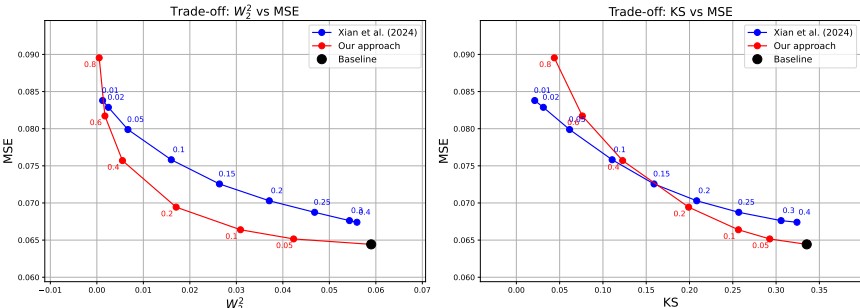

Figure 5: Tradeoff between fairness and accuracy for different $(2, 1/n)$-DP methods on the one-dimensional private regression problem of Remark C.4, averaged over 5 random seeds. Fairness is measured in terms of both $W_2^2$ (squared Wasserstein distance) and KS distance, while accuracy is measured by mean squared error (MSE). Blue numbers represent the radius $\beta$ of the KS ball used in the approach of Xian et al. (2024), and red numbers the value of the regularization parameter $\alpha$ in our approach. Our method achieves competitive performance under both fairness metrics compared to the post-processing approach of Xian et al. (2024).

### 7.4 Fair and private 1-D regression: comparison to the baseline

Finally, we compare with the one-dimensional regression method of Xian et al. (2024) which controls fairness via Kolmogorov–Smirnov (KS) distance, using a transformation of the same biased data. Their approach is specific to the 1-D case, whereas ours extends naturally. On the same task, our method achieves a better fairness–utility tradeoff under $W_2^2$ and remains competitive under the KS metric (Figure 5, see Remark C.4 for further details).

## 8 Conclusion

We introduced a novel sensitivity analysis for sliced Wasserstein gradients, enabling differentially private learning in distributional matching problems. Building on this foundation, we proposed two new applications: (i) the first private training procedure for sliced Wasserstein autoencoders, and (ii) a general in-processing method for private and fair learning. On benchmarks where Wasserstein-based baselines exist (Section 6 and Appendix D), our approach consistently matches or surpasses these prior methods, while uniquely enabling applications that were previously out of reach. These results underline the potential of our framework to advance private (and fair) machine learning.

We note that sliced Wasserstein autoencoders, while not state-of-the-art for data generation compared to specialized models such as DP diffusion, were never primarily designed for this purpose. Their value lies in being lightweight, versatile, and now, for the first time, trainable under differential privacy. This positions our framework not as a competitor to tailored generative models, but as a complementary tool for representation learning, fairness, and private distributional matching.

A practical limitation is the computational overhead of Jacobian computations (Remark 5.1). Such costs are common in differential privacy (De et al., 2022), but improving the efficiency of Jacobian evaluation remains an important avenue for scaling our method to larger models.

Our analysis also highlights several promising research directions. First, the sensitivity bounds hinge on the quadratic structure of $W_2^2$: as shown in Appendix F, they do not extend to general $W_p$ with $p \geq 1$. Extending the framework to $W_p^p$ for $p > 1$ seems both feasible and worthwhile. Second, incorporating sliced Wasserstein barycenters under DP would broaden applicability to aggregation and clustering tasks. Finally, *entropic regularization* could reduce complexity and improve stability.

## Acknowledgements

This work was partially funded by the Agence Nationale de la Recherche under grant ANR-23-CE23-0029 Regul-IA. The research leading to these results received funding from MCIN/AEI/10.13039/501100011033/FEDER under Grant Agreement Number PID2021-128314NBI00. The authors also acknowledge the support of the AI Cluster ANITI (ANR-19-PI3A-0004).

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

## Appendix

## A    Extension to the Sliced Wasserstein Distance ($d \geq 2$)

As pointed out in Remark 4.3, the results of this paper can be extended to higher dimensions by considering the sliced Wasserstein distance. Assume now that $g_\theta : \mathcal{X} \to \mathbb{R}^d$ and $h_\theta : \mathcal{Z} \to \mathbb{R}^d$. As in Section 4, we make no assumptions on $\mathcal{X}$ and $\mathcal{Z}$. We only assume that $\Theta \subseteq \mathbb{R}^r$, where $r$ is the number of parameters, and that, for every $x \in \mathcal{X}$ and $z \in \mathcal{Z}$, the maps $\theta \mapsto g_\theta(x)$ and $\theta \mapsto h_\theta(z)$ are differentiable on $\Theta$. Following the notation of Section 4, we are interested now in bounding the sensitivity of the gradient of the (squared) sliced Wasserstein distance between the distributions $g_\theta \# P_{\mathbf{X}}$ and $h_\theta \# P_{\mathbf{Z}}$ in $\mathbb{R}^d$, defined as

$$SW_2^2(g_\theta \# P_{\mathbf{X}}, h_\theta \# P_{\mathbf{Z}}) = \int_{\mathbb{S}^{d-1}} W_2^2\Big(\mathrm{Pr}_\vartheta \#(g_\theta \# P_{\mathbf{X}}), \mathrm{Pr}_\vartheta \#(h_\theta \# P_{\mathbf{Z}})\Big) d\rho(\vartheta) \ ,$$

where $\rho$ represents the uniform measure on $\mathbb{S}^{d-1}$, the unit sphere of $\mathbb{R}^d$. From a practical standpoint, we are mainly interested in the study of the gradient of its Monte Carlo approximation given by $k$ i.i.d. random directions $\vartheta_1, \ldots, \vartheta_k \in \mathbb{S}^{d-1}$,

$$SW_{2,k}^2(g_\theta \# P_{\mathbf{X}}, h_\theta \# P_{\mathbf{Z}}) = \frac{1}{k} \sum_{l=1}^{k} W_2^2\Big(\mathrm{Pr}_{\vartheta_l} \#(g_\theta \# P_{\mathbf{X}}), \mathrm{Pr}_{\vartheta_l} \#(h_\theta \# P_{\mathbf{Z}})\Big) \ .$$

As in the proof of Theorem 4.1, it suffices to bound the sensitivity of the gradient with respect to the substitution neighboring relation $\mathbf{X} \sim_1 \tilde{\mathbf{X}}$. If we define $\Phi(\mathbf{X}) = \nabla_\theta SW_2^2(g_\theta \# P_{\mathbf{X}}, h_\theta \# P_{\mathbf{Z}})$ and $\Phi_\vartheta(\mathbf{X}) = \nabla_\theta W_2^2(\mathrm{Pr}_\vartheta \#(g_\theta \# P_{\mathbf{X}}), \mathrm{Pr}_\vartheta \#(h_\theta \# P_{\mathbf{Z}}))$, by the chain rule and the same reasoning as in the proof of Theorem 1 in Bonneel et al. (2015), we know that under suitable smoothness assumptions, in the set of non-repeated points $\Gamma = \{\theta : g_\theta(x_i) \neq g_\theta(x_j), h_\theta(z_i) \neq h_\theta(z_j) \text{ for } i \neq j\}$,

$$\Phi(\mathbf{X}) = \int_{\mathbb{S}^{d-1}} \Phi_\vartheta(\mathbf{X}) d\rho(\vartheta) \ .$$

As in Section 3, we can define the *gradient* $\Phi(\mathbf{X})$ by this expression, even outside the set of differentiability points $\Gamma$, and provide privacy guarantees for every point. Similarly, if we consider the Monte Carlo approximation of the gradient $\Phi(\mathbf{X}) = \nabla_\theta SW_{2,k}^2(g_\theta \# P_{\mathbf{X}}, h_\theta \# P_{\mathbf{Z}})$, it follows that $\Phi(\mathbf{X}) = \frac{1}{k} \sum_{l=1}^{k} \Phi_{\vartheta_l}(\mathbf{X})$. In any case, we can conclude that

$$\Delta_2(\Phi) = \sup_{\mathbf{X} \sim \tilde{\mathbf{X}}} \|\Phi(\mathbf{X}) - \Phi(\tilde{\mathbf{X}})\|_2 \leq \sup_{\vartheta \in \mathbb{S}^{d-1}} \Delta_2(\Phi_\vartheta) \ .$$

The sensitivity of $\Phi_\vartheta$ can be controlled with the one-dimensional results in Section 4. Note that if we define $g_\theta^\vartheta(x) = (\mathrm{Pr}_\vartheta \circ g_\theta)(x) = \vartheta^T g_\theta(x)$ and $h_\theta^\vartheta(z) = (\mathrm{Pr}_\vartheta \circ h_\theta)(z) = \vartheta^T h_\theta(z)$, then $\Phi_\vartheta(\mathbf{X}) = \nabla_\theta W_2^2(g_\theta^\vartheta \# P_{\mathbf{X}}, h_\theta^\vartheta \# P_{\mathbf{Z}})$, and we can conclude

$$\Delta_2(\Phi_\vartheta) \leq 4M \frac{3L_1 + L_2}{n}$$

provided that:

(i)  $|g_\theta^\vartheta(x)| = |\vartheta^T g_\theta(x)| \leq M, |h_\theta^\vartheta(x)| = |\vartheta^T h_\theta(z)| \leq M$ .

(ii) $\|\nabla_\theta g_\theta^\vartheta(x)\|_2 = \|\vartheta^T \mathcal{J}_\theta g_\theta(x)\|_2 \leq L_1, \|\nabla_\theta h_\theta^\vartheta(z)\|_2 = \|\vartheta^T \mathcal{J}_\theta h_\theta(z)\|_2 \leq L_2.$

In particular, both inequalities hold uniformly in $\vartheta \in \mathbb{S}^{d-1}$ if we impose the following, more natural conditions:

(i)  $\|g_\theta(x)\|_2 \leq M, \|h_\theta(z)\|_2 \leq M$

(ii) $\|\mathcal{J}_\theta g_\theta(x)\|_2 = \sup\limits_{\|\eta\|_2=1} \|\mathcal{J}_\theta g_\theta(x)\eta\|_2 \leq L_1$, $\|\mathcal{J}_\theta h_\theta(z)\|_2 = \sup\limits_{\|\eta\|_2=1} \|J_\theta h_\theta(z)\eta\|_2 \leq L_2$.

By the properties of the spectral norm, $\|\mathcal{J}_\theta g_\theta(x)\|_2 = \|\mathcal{J}_\theta g_\theta(x)^T\|_2$. Therefore, for every $\vartheta \in \mathbb{S}^{d-1}$ and $x$,

$$\|\nabla_\theta g_\theta^\vartheta(x)\|_2 = \|\vartheta^T \mathcal{J}_\theta g_\theta(x)\|_2 = \|\mathcal{J}_\theta g_\theta(x)^T \vartheta\|_2 \leq L_1 \ ,$$

and similarly for $h_\theta$. As in the one dimensional setting, Assumption (ii) is verified if $g_\theta$ and $h_\theta$ are $L_1$-Lipschitz and $L_2$-Lipschitz with respect to $\theta$. To see this, note that if $\|\eta\|_2 = 1$, by the Lipschitz condition,

$$\|\mathcal{J}_\theta g_\theta(x)\eta\|_2 = \left\| \lim_{t \to 0} \frac{g_{\theta+t\eta}(x) - g_\theta(x)}{t} \right\| \leq L_1 \|\eta\|_2 = L_1 \ .$$

Therefore, Theorem 4.1 can be extended to the multidimensional setting with the sliced Wasserstein distance as follows:

**Theorem A.1.** *With all the previous notation, assume that there exist constants $M, L_1, L_2 \geq 0$ such that for each $\theta \in \Theta$, $x \in \mathcal{X}$ and $z \in \mathcal{Z}$,*

*(i) $\|g_\theta(x)\|_2 \leq M$, $\|h_\theta(z)\|_2 \leq M$*

*(ii) $\|\mathcal{J}_\theta g_\theta(x)\|_2 = \sup\limits_{\|\eta\|_2=1} \|\mathcal{J}_\theta g_\theta(x)\eta\|_2 \leq L_1$, $\|\mathcal{J}_\theta h_\theta(z)\|_2 = \sup\limits_{\|\eta\|_2=1} \|J_\theta h_\theta(z)\eta\|_2 \leq L_2$.*

*Then, under neighboring relation $\sim_1$ in $\mathcal{D} = \mathcal{X}^n$, if we define $\Phi(\mathbf{X})$ as $\nabla_\theta SW_2^2(g_\theta\#P_\mathbf{X}, h_\theta\#P_\mathbf{Z})$ or its Monte Carlo approximation $\nabla_\theta SW_{2,k}^2(g_\theta\#P_\mathbf{X}, h_\theta\#P_\mathbf{Z})$ then*

$$\Delta_2(\Phi) \leq 4M \frac{3L_1 + L_2}{n} \ .$$

*Consequently, under neighboring relation $\sim_2$ in $\mathcal{D} = \mathcal{X}^n \times \mathcal{Z}^m$, if we define $\Psi(\mathbf{X}, \mathbf{Z})$ as $\nabla_\theta SW_2^2(g_\theta\#P_\mathbf{X}, h_\theta\#P_\mathbf{Z})$ or its Monte Carlo approximation $\nabla_\theta SW_{2,k}^2(g_\theta\#P_\mathbf{X}, h_\theta\#P_\mathbf{Z})$, then*

$$\Delta_2(\Psi) \leq 4M \max\left\{ \frac{3L_1 + L_2}{n}, \frac{L_1 + 3L_2}{m} \right\} \ .$$

*Remark* A.2. From a computational point of view, if we want to define a clipped approximation $\mathcal{J}_\theta^{L_1} g_\theta(x)$ of $\mathcal{J}_\theta g_\theta(x)$ that verifies Assumption (ii) in Theorem A.1, this might be done by clipping the eigenvalues of the singular value decomposition of $\mathcal{J}_\theta g_\theta(x)$. This should be done at each step, for each $x_i$ in the batch. To simplify the computation and enable easy parallelization, we have adopted a suboptimal, naive alternative approach. Given $g_\theta = (g_\theta^1, \ldots, g_\theta^d)$, we define

$$\mathcal{J}_\theta^{L_1} g_\theta(x) = \begin{pmatrix} \text{clip}_{\frac{L_1}{\sqrt{d}}}(\nabla_\theta g_\theta^1(x)) \\ \vdots \\ \text{clip}_{\frac{L_1}{\sqrt{d}}}(\nabla_\theta g_\theta^d(x)) \end{pmatrix} \ .$$

Then, it follows that for every $\eta$ verifying $\|\eta\|_2 = 1$,

$$\|\mathcal{J}_\theta^{L_1} g_\theta(x)\eta\|_2 = \left( \sum_{i=1}^d \langle \text{clip}_{\frac{L_1}{\sqrt{d}}}(\nabla_\theta g_\theta^i(x)), \eta \rangle^2 \right)^{1/2} \leq \left( \sum_{i=1}^d \frac{L_1^2}{d} \right)^{1/2} = L_1$$

which implies the desired bound on the spectral norm, $\|\mathcal{J}_\theta^{L_1} g_\theta(x)\|_2 \leq L_1$.

# B Additional Details on the Private Sliced Wasserstein Autoencoder

Table 2: Comparison of different methods on MNIST and Fashion-MNIST under $(\varepsilon, \delta)$-DP, with $\boldsymbol{\varepsilon = 10}$, $\delta = 10^{-5}$. Results for our approach are averaged over 5 random seeds. Scores for all methods except ours are taken from Dockhorn et al. (2023) or Yang et al. (2024b). - means that the metric was not reported. FID denotes the Fréchet Inception Distance (Heusel et al., 2017), computed using statistics from features of a pretrained network as in Dockhorn et al. (2023). Accuracy (Acc) is computed by training an MLP classifier (width = 100, as in Yang et al. (2024b)) on synthetic images, and testing it on the test set. Since our generator is unconditional, labels are assigned using a KNN classifier in the embedding space (used for noise injection). Values in parentheses correspond to the composition of our autoencoder and the trained classifier, leveraging our autoencoder structure to improve utility.

| METHOD | MNIST (FID/Acc) | F-MNIST (FID/Acc) |
|---|---|---|
| DP-SWAE (Ours) | 123.6 / 0.755 (0.848) | 145.8 / 0.692 (0.733) |
| DP-NTK (Yang et al., 2024b) | - / 0.880 | - / 0.784 |
| DPDM (Dockhorn et al., 2023) | 5.0 / 0.948 | 18.6 / 0.830 |
| PEARL (Liew et al., 2022) | 116.0 / 0.783 | 102.0 / 0.732 |
| DP-Sinkhorn (Cao et al., 2021) | 48.4 / 0.827 | 128.3 / 0.746 |
| DP-CGAN (Torzadehmahani et al., 2019) | 179.2 / 0.600 | 243.8 / 0.500 |
| DP-HP (Vinaroz et al., 2022) | - / 0.804 | - / 0.717 |
| DP-MERF (Harder et al., 2021) | 116.3 / 0.783 | 132.6 / 0.745 |
| GS-WGAN (Chen et al., 2020) | 61.3 / 0.790 | 131.3 / 0.650 |

Table 3: Comparison of different methods on MNIST and Fashion-MNIST under $(\varepsilon, \delta)$-DP, with $\boldsymbol{\varepsilon = 1}$, $\delta = 10^{-5}$, using the same experimental setting as in Table 2.

| METHOD | MNIST (FID/Acc) | F-MNIST (FID/Acc) |
|---|---|---|
| DP-SWAE (Ours) | 146.2 / 0.556 (0.772) | 160.9 / 0.567 (0.709) |
| DP-NTK (Yang et al., 2024b) | - / 0.862 | - / 0.764 |
| DPDM (Dockhorn et al., 2023) | 23.4 / 0.916 | 18.6 / 0.769 |
| PEARL (Liew et al., 2022) | 121.0 / 0.796 | 109.0 / 0.740 |
| DP-HP (Vinaroz et al., 2022) | - / 0.801 | - / 0.721 |
| DP-MERF (Harder et al., 2021) | - / 0.807 | - / 0.738 |

Table 4: Architecture of the convolutional autoencoder.

| ENCODER | DECODER |
|---|---|
| Input: $1 \times 28 \times 28$ image | Input: latent vector ($\mathbb{R}^{\text{latent\_dim}}$) |
| Conv2D (8 filters, $3 \times 3$), LeakyReLU (0.2) | FC: latent\_dim $\to$ 64, ReLU |
| AvgPool ($2 \times 2$) $\to 8 \times 14 \times 14$ | FC: $64 \to 128$, ReLU |
| Conv2D (16 filters, $3 \times 3$), LeakyReLU (0.2) | FC: $128 \to 784$, ReLU |
| AvgPool ($2 \times 2$) $\to 16 \times 7 \times 7$ | Reshape to $16 \times 7 \times 7$ |
| Conv2D (16 filters, $3 \times 3$), LeakyReLU (0.2) | Conv2D (16 filters, $3 \times 3$), LeakyReLU (0.2) |
| Flatten $\to \mathbb{R}^{784}$ | Upsample $\to 16 \times 14 \times 14$ |
| FC: $784 \to 128$, ReLU | Conv2D (8 filters, $3 \times 3$), LeakyReLU (0.2) |
| FC: $128 \to 64$, ReLU | Upsample $\to 8 \times 28 \times 28$ |
| FC: $64 \to$ latent\_dim | Conv2D (1 filter, $3 \times 3$), Sigmoid |

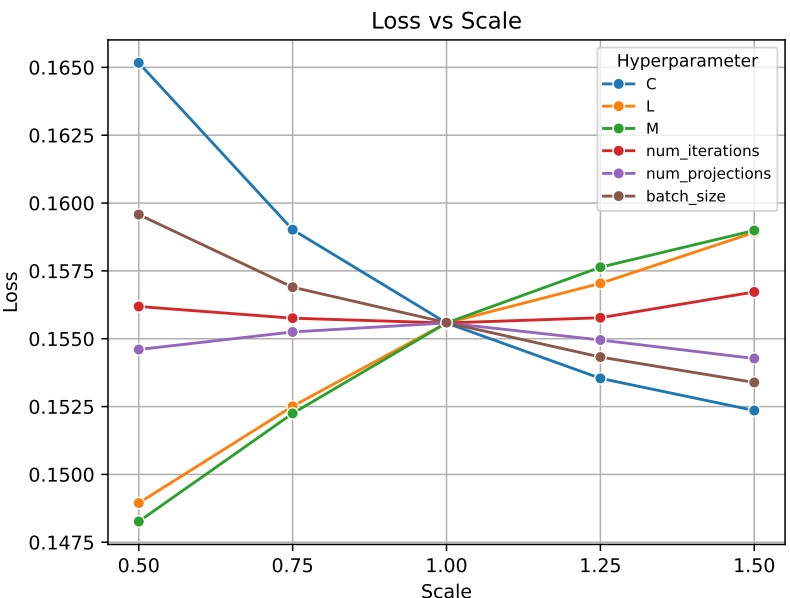

Figure 6: Ablation study of the original hyperparameters used to train the DP-SWAE model of Section 6: $C = 1$, $M = 1.5$, $L = \sqrt{6}$, number of iterations $= 5000$, number of projections $= 100$, and batch size $= 600$, under $(\varepsilon, \delta)$-DP with $\varepsilon = 5$ and $\delta = 10^{-5}$. In each case, one hyperparameter is scaled while the remaining hyperparameters are kept fixed, and the noise level is adjusted accordingly to preserve $(\varepsilon, \delta)$-DP after the specified number of iterations. The figure reports the resulting training loss for each hyperparameter across different scaling levels.

**Comments on the ablation study.** From Figure 6, we draw three main observations. First, the optimization procedure appears robust to changes in both the number of projections and the number of iterations. Second, the method benefits from larger batch sizes, which is consistent with the behavior already observed for standard DPSGD in deep learning De et al. (2022). Third, the results suggest that better choices of $L$, $M$, and $C$ could have been made for this particular problem. We note, however, that $C$ was fixed to 1, following the default setting used in private optimization libraries Yousefpour et al. (2021), while $L$ and $M$ were selected a priori based on the geometry of the problem. This design choice is especially well motivated in the differentially private setting, where hyperparameter tuning is itself difficult and typically requires private validation procedures that consume additional privacy budget.

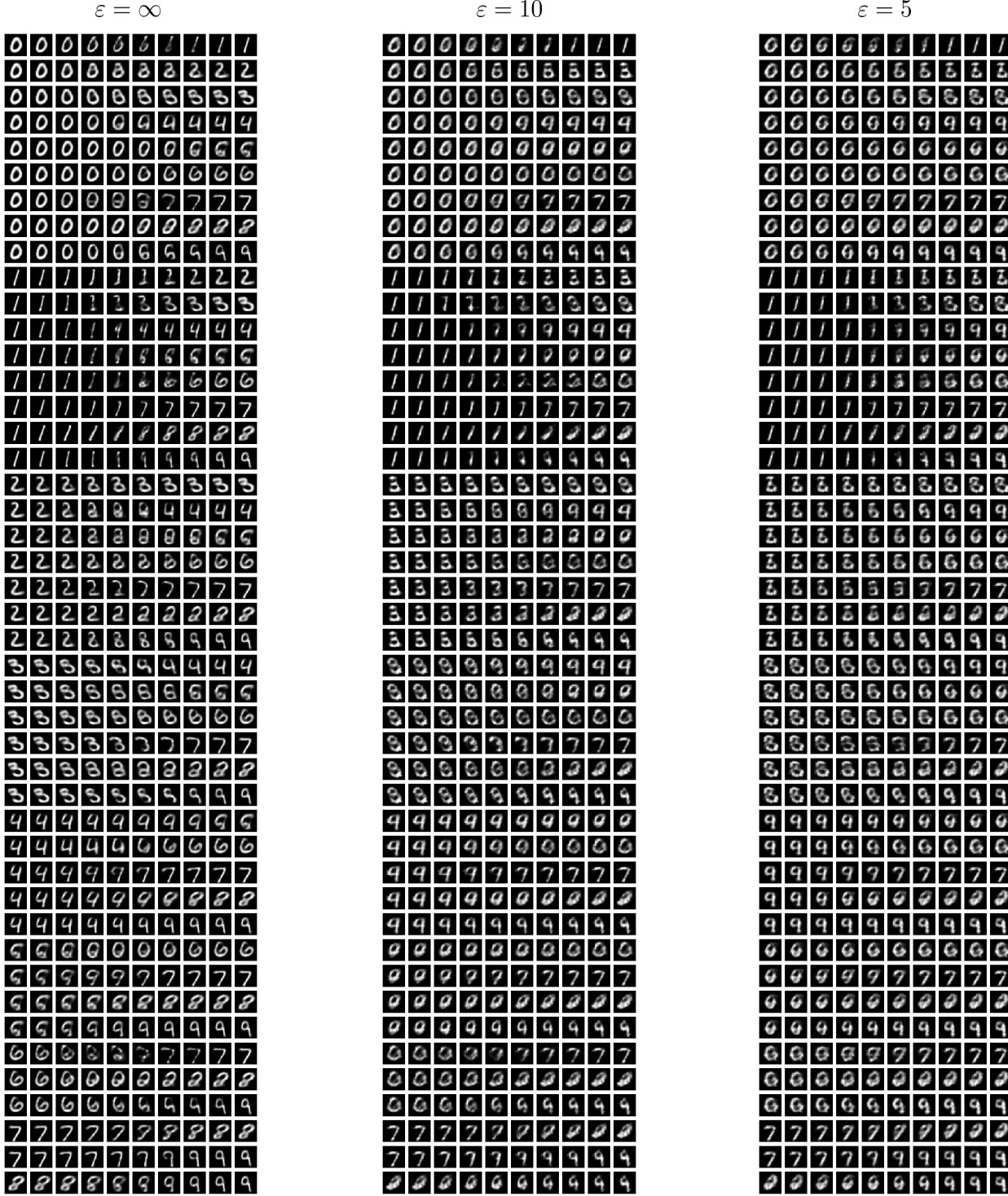

Figure 7: MNIST interpolations between the earliest samples $\{z_i\}_{i=0}^{9}$ of each digit (0–9) in the test set, produced by the encoder $\varphi_\theta$ and decoder $\psi_\theta$ trained under $(\varepsilon, \delta)$-DP, with $\delta = 10^{-5}$ and varying values of $\varepsilon$. For each pair $0 \leq i < j \leq 9$, we compute the linear interpolation in the latent space $(1 - \eta_l)\varphi_\theta(z_i) + \eta_l \varphi_\theta(z_j)$, for $\eta_l = \frac{l}{9}$, $0 \leq l \leq 9$, and map it back to the image space using the decoder $\psi_\theta$.

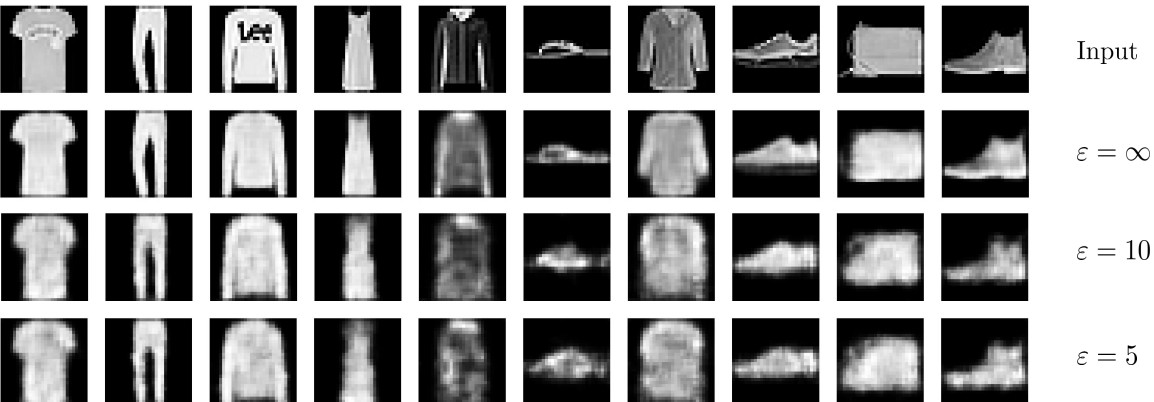

Figure 8: Reconstructed Fashion-MNIST images from the test dataset. The first row shows the earliest sample of each label (0–9) in the test set. The subsequent rows display the corresponding reconstructions produced by the trained autoencoder under $(\varepsilon, \delta)$-DP, with $\delta = 10^{-5}$ and varying values of $\varepsilon$.

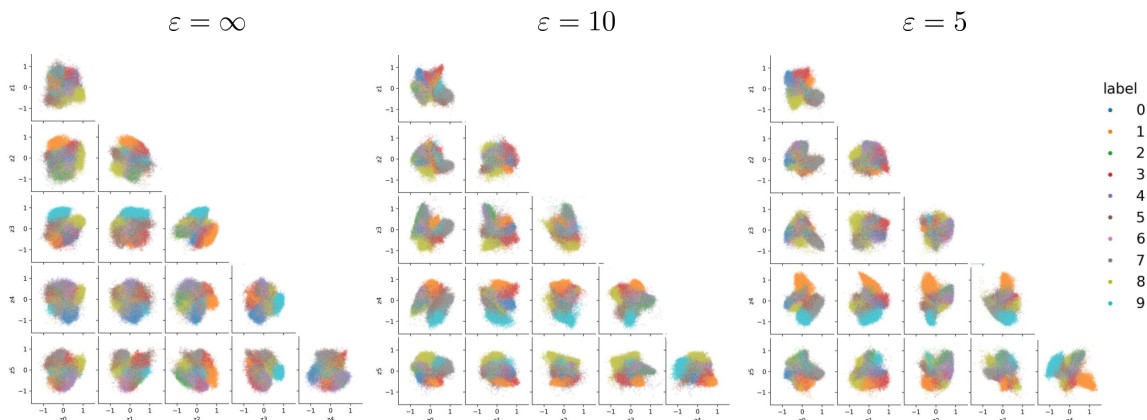

Figure 9: Encoded latent space Fashion-MNIST samples for the autoencoder, trained under $(\varepsilon, \delta)$-DP, with $\delta = 10^{-5}$ and varying values of $\varepsilon$.

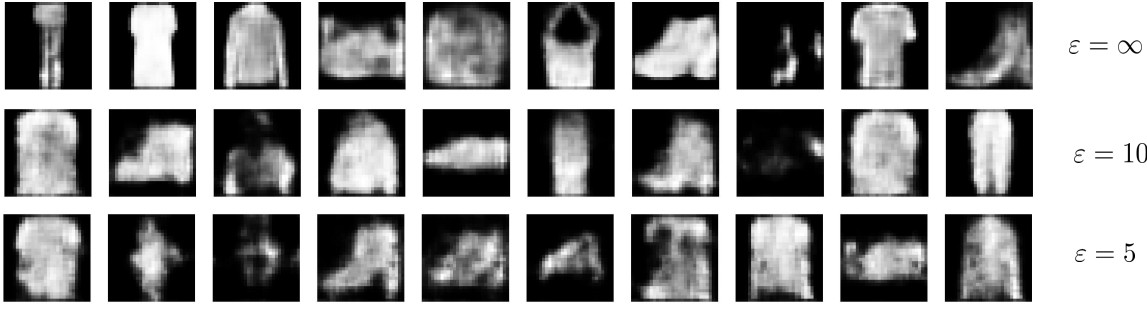

Figure 10: Generated samples from the Fashion-MNIST autoencoder trained under $(\varepsilon, \delta)$-DP, with $\delta = 10^{-5}$ and varying values of $\varepsilon$.

$\varepsilon = \infty$  $\varepsilon = 10$  $\varepsilon = 5$

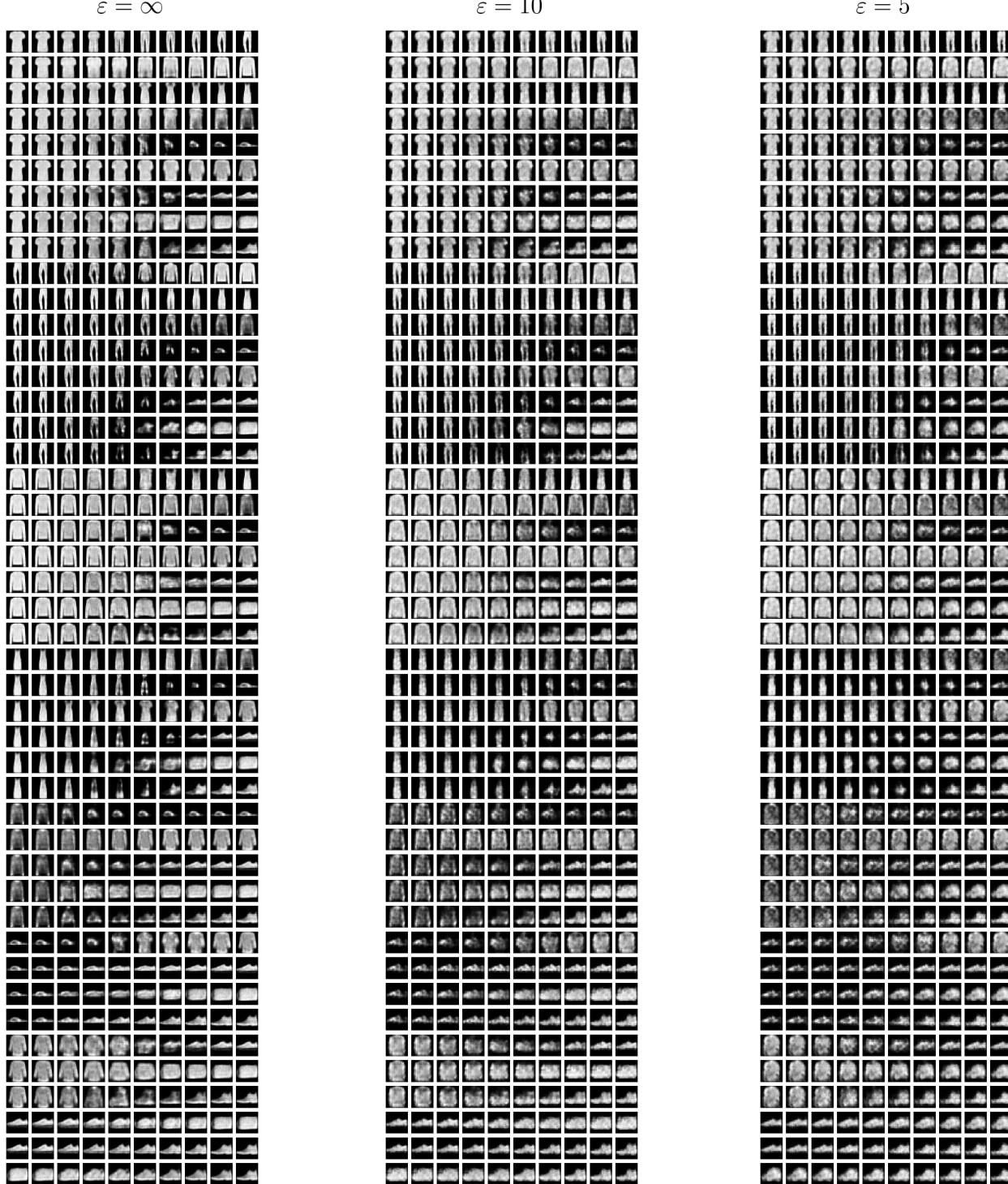

Figure 11: MNIST interpolations between the earliest samples $\{z_i\}_{i=0}^{9}$ of each label (0–9) in the test set, produced by the encoder $\varphi_\theta$ and decoder $\psi_\theta$ trained under $(\varepsilon, \delta)$-DP, with $\delta = 10^{-5}$ and varying values of $\varepsilon$. For each pair $0 \leq i < j \leq 9$, we compute the linear interpolation in the latent space $(1 - \eta_l)\varphi_\theta(z_i) + \eta_l\varphi_\theta(z_j)$, for $\eta_l = \frac{l}{9}$, $0 \leq l \leq 9$, and map it back to the image space using the decoder $\psi_\theta$.

## C  Fairness via Private In-Processing

### C.1  General method

Let us begin by recalling the general framework introduced in Section 7. Let $\mathbf{D}$ denote a dataset with $n$ samples $(x_i, a_i, y_i)$ or $(x_i, a_i)$, where $x_i$ are the non-sensitive attributes, $a_i \in \{0, 1\}$ is the sensitive attribute and $y_i$ the response variable, only available in supervised problem. We consider the general empirical risk minimization problem:

$$\min_\theta \mathscr{L}(\theta) := \min_\theta \frac{1}{n} \sum_{i=1}^n \ell(g_\theta(x_i)) \, , \tag{8}$$

where $\ell(g_\theta(x_i))$ is a shorthand for $\ell(g_\theta(x_i), y_i)$ in supervised problems and $\ell(g_\theta(x_i), x_i)$ in unsupervised problems. The goal is to privately optimize a regularized version of this problem, incorporating a fairness penalty based on the sliced Wasserstein distance. As explained in Section 6, the finite-sum structure of the loss function translates to the gradient, and as long as $\|\nabla_\theta \ell(g_\theta(x))\|_2 \leq C$ for all $x \in \mathcal{X}$, the sensitivity of $\nabla_\theta \mathscr{L}(\theta)$ is bounded by $2C/n$, not only for the substitution relation $\sim_1$, but for any k-end neighboring relation $\sim_k$. To choose an adequate fairness penalty, we have considered in this work two different fairness notions: *Statistical parity* and *Equality of Odds*. The first was already discussed in Section 7, we include it again for completeness. We present this section for the general case of the sliced Wasserstein distance, note that the case $d = 1$ coincides with the one-dimensional Wasserstein distance. As in Section 7, let $\varphi_\theta$ denote the representation to which the fairness penalty is applied. That is, $\varphi_\theta = g_\theta$ if the penalty is applied at the output, or $g_\theta = \psi_\theta \circ \varphi_\theta$ if it is applied to an intermediate representation (e.g., an intermediate layer of a neural network). In either case, for any pair of distributions $P, Q$, the equality $\mathcal{L}(\varphi_\theta \# P) = \mathcal{L}(\varphi_\theta \# Q)$ implies $\mathcal{L}(g_\theta \# P) = \mathcal{L}(g_\theta \# Q)$, which justifies penalizing any intermediate representation.

**Statistical Parity (SP):** Statistical parity corresponds to the situation where the algorithmic decision does not depend on the sensitive variable. Statistical parity is thus satisfied if $\mathcal{L}(g_\theta(X)|A = 0) = \mathcal{L}(g_\theta(X)|A = 1)$. Given our data, if we define $\mathbf{X}_j = (x_i : a_i = j)$, $n_j = |\mathbf{X}_j|$ for $j = 0, 1$, statistical parity can be favored by minimizing

$$\mathscr{L}_\alpha^{SP}(\theta) = (1 - \alpha)\mathscr{L}(\theta) + \alpha SW_2^2\Big(\varphi_\theta \# P_{\mathbf{X}_0}, \varphi_\theta \# P_{\mathbf{X}_1}\Big) \tag{9}$$

where $\alpha \in [0, 1]$ measures the weight of each part in the optimization. In order to establish privacy guarantees, we need to assume that $n_0$ and $n_1$ are fixed.

**Equality of Odds (EO):** Beyond guaranteeing the same decision for all, which is not suitable in some cases where the sensitive variable impacts the decision, bias mitigation may require that the model performs with the same accuracy for all groups, often referred to as equality of odds. We focus only in the supervised case, where $y_i$ is available and takes values in $\{0, \ldots, R - 1\}$. In this case, equality of odds is verified if $\mathcal{L}(g_\theta(X)|A = 0, Y = k) = \mathcal{L}(g_\theta(X)|A = 1, Y = k)$ for all $k \in \{0, \ldots, R - 1\}$. With the same ideas as before, if we define $\mathbf{X}_{j,k} = (x_i : a_i = j, y_i = k)$, $n_{j,k} = |\mathbf{X}_{j,k}|$ for $j \in \{0, 1\}, k \in \{0, \ldots, R - 1\}$, equality of odds bias mitigation can be enforced by training with loss

$$\mathscr{L}_\alpha^{EO}(\theta) = (1 - \alpha)\mathscr{L}(\theta) + \frac{\alpha}{R} \sum_{k=1}^R SW_2^2\Big(\varphi_\theta \# P_{\mathbf{X}_{0,k}}, \varphi_\theta \# P_{\mathbf{X}_{1,k}}\Big) \tag{10}$$

To obtain privacy guarantees, now we need to impose that the values $n_{j,k}$ are fixed.

**Corollary C.1.** *If $\varphi_\theta$ verifies $\|\varphi_\theta(x)\|_2 \leq M$, $\|\mathcal{J}_\theta \varphi_\theta(x)\|_2 \leq L$ and $\|\nabla_\theta \ell(g_\theta(x))\|_2 \leq C$, then*

*• For SP, under $\sim_2$, the sensitivity of $\nabla_\theta \mathscr{L}_\alpha^{SP}(\theta)$ or its MC approximation is bounded by*

$$(1 - \alpha)\frac{2C}{n} + \alpha \frac{16ML}{\min\{n_0, n_1\}} \, . \tag{11}$$

- *For EO, under $\sim_{2R}$, the sensitivity of $\nabla_\theta \mathscr{L}_\alpha^{EO}(\theta)$ or its MC approximation is bounded by*

$$(1-\alpha)\frac{2C}{n} + \frac{\alpha}{R}\frac{16ML}{\min_{j,k}\{n_{j,k}\}} \ . \tag{12}$$

*Proof.* See Appendix G.3. $\qquad\qquad\square$

*Remark* C.2. Our privacy guarantees in the fairness framework are built upon the knowledge of class sizes. The importance of controlling these sizes has been previously recognized. For example, Lowy et al. (2023) imposes a restriction on the minimum proportion of elements in each class, while Ghoukasian & Asoodeh (2024) and Xian et al. (2024) derive privacy guarantees that degrade with smaller class sizes. Conceptually, our framework for establishing privacy guarantees is very sound. Even though an attacker might learn some information about the number of individuals in each class used during training, they cannot distinguish between the outputs of two datasets $\mathbf{D}$ and $\tilde{\mathbf{D}}$ differing only in one individual from the same class.

## C.2 Applications: Private bias mitigation in diverse learning tasks

In order to demonstrate the versatility of our methodology for imposing fairness in different scenarios, we use an illustrative synthetic model to simulate bias in algorithmic decision-making. Note that we do not include extensive comparisons with other application-specific methodologies, as our approach is highly general and does not include any of the statistical, convergence, or fairness guarantees described by other methods (see Xu et al. (2019); Jagielski et al. (2019); Ding et al. (2020); Lowy et al. (2023); Yaghini et al. (2023); Ghoukasian & Asoodeh (2024) for the fair and private classification problem, or Xian et al. (2024) for fair and private one-dimensional regression). Yet we provide, to the best of our knowledge, the first method to handle novel problems such as multidimensional fair and private regression, or fair and private representation learning.

We consider $\mathbf{D} = \{(x_i, a_i, y_i^C, y_i)\}_{i=1}^n$ i.i.d. samples with the same distribution as $(X, A, Y^C, Y)$, where $X$ denotes the features, $A$ is the sensitive variable, $Y^C = (Y_1^C, Y_2^C)$ is a continuous response variable and $Y$ is a discrete version of $Y^C$, related by

(i) $Y^C \sim U([0,1] \times [0,1])$

(ii) $Y = I\left(Y_2^C > 1 - Y_1^C\right)$

(iii) $A = BY + (1-B)(1-Y)$, where $B \sim \text{Bernoulli}(p)$ independent of $Y$.

(iv) $X_{core} = \underbrace{[Y^C, \ldots, Y^C]}_{d_{core}/2 \ times} + N\left(0, \sigma_{core}^2 I_{d_{core}}\right), \quad X_{sp} = \underbrace{[A, \ldots, A]}_{d_{sp} \ times} + N\left(0, \sigma_{sp}^2 \ I_{d_{sp}}\right).$

(v) $X = [X_{core}, X_{sp}]$

Therefore, this generated data consists in a response variable $Y_C$, which is correlated with the sensitive attribute $A$. The features $X$ are divided into two parts: a first part $X_{core}$ which is a noisy transformation of $Y_C$, and a second spurious part $X_{sp}$ which is a noisy version of $A$. If $p$ is close to 1, most of the cases verify $A = Y$ and therefore, the decision of the algorithm relies highly on the sensitive variable $A$. Bias in the algorithmic decision is created when the sensitive variable $A$ is not aligned with the decision. When $Y \neq A$, the learning task is more complicated since while $X_{core}$ is correlated with $Y$, the spurious part pushes towards the bad decision. This setting reproduces the characteristics of some of the main biases present in many data sets, for instance, (Becker & Kohavi, 1996) or (Hofmann, 1994) in supervised learning. We explore this problem across different scenarios, demonstrating how penalized models using our Wasserstein–based losses can help mitigate unfairness, according to various fairness notions, while maintaining differential privacy guarantees. All experiments are conducted on the same synthetic dataset, generated with the previous mechanism for a fixed seed and values $n = 30000$, $p = 0.7$, $d_{core} = d_{sp} = 8$, $\sigma_{core}^2 = 1/5$, $\sigma_{sp}^2 = 2/5$.

All models are trained with DP-SGD as explained in Section 5, with clipping constant $C > 0$ for the individual gradients in (8), as usual in DP-SGD, and inner clipping constants $M, L > 0$ for the Wasserstein

gradient approximation (5) or its sliced version. In the latter case, all the experiments use the naive clipping procedure explained in Remark A.2. Corollary C.1 and the procedure described in Section 5, enable us to compute the privacy budget obtained after $T$ iterations of DP-SGD. In particular, in all the experiments, we fix the number of iterations $T$ and the value of $\delta$, and compute the required noise to obtain $(\varepsilon, \delta)$-DP after $T$ iterations, for different values of the privacy budget $\varepsilon$ and the weight $\alpha \in [0, 1]$ in the penalized loss functions (9) and (10).

Following the above notation, $\mathbf{X}_j = (x_i : a_i = j)$, $n_j = |\mathbf{X}_j|$ for $j = 0, 1$, and $\mathbf{X}_{j,k} = (x_i : a_i = j, y_i = k)$, $n_{j,k} = |\mathbf{X}_{j,k}|$ for $j, k \in \{0, 1\}$. Given our data generation procedure, we know that $\mathbb{E}(n_j) = n/2$, $\mathbb{E}(n_{j,j}) = pn/2$ and $\mathbb{E}(n_{j,1-j}) = (1-p)n/2$ for $j \in \{0, 1\}$. In all the subsequent experiments, the batch sizes considered are $n'_j \approx n_j/10$ when minimizing (9), and $n'_{j,k} \approx n_{j,k}/10$ when minimizing (10), where the approximation is related to internal parallelization of the gradient computations in the code detailed in Section E. A summary of the sample sizes in the data and the batch sizes considered is presented in Table 5.

Table 5: Sample sizes / batch sizes for different groups and subgroups. The center part displays the pairs $n_{j,k}/n'_{j,k}$, while the right side displays the pairs $n_j/n'_j$.

|  | $\mathbf{Y = 0}$ | $\mathbf{Y = 1}$ | $\mathbf{TOTAL}$ |
|---|---|---|---|
| $\mathbf{A = 0}$ | 10608 / 1050 | 4564 / 450 | 15172 / 1500 |
| $\mathbf{A = 1}$ | 4446 / 400 | 10382 / 1000 | 14828 / 1450 |

### C.3 Classification

First, we consider the problem of predicting the label $Y$ as a function of $X$. Our decision rule is based on logistic regression, where the function $g_\theta$ maps each $x_i$ to the predicted probability $g_\theta(x_i) \in (0, 1)$. The classification rule is given by $G_\theta(x) = I(g_\theta(x) > 1/2)$. $g_\theta$ is defined as a neural network with one layer and a sigmoid activation function, and the classification loss is defined as the binary cross-entropy. We impose fairness using the two notions presented above. Considering $\varphi_\theta = g_\theta$, we have trained our model with simple iterations of DP-SGD to minimize (9) to enforce SP, and (10) to enforce EOO. In both cases, we have trained the model for different values of the weight $\alpha$ and the privacy budget $\varepsilon$, when we fix $\delta = 0.1/n$, number of iterations $T = 500$, clipping constants $C = 5$, $M = L = 1$ and learning rate $= 0.05$. For each fairness notion, we present a brief overview of the methodology, followed by a detailed exposition of the results. We include statistical measures to assess the trade-off between fairness, privacy, and utility for each model.

**Statistical Parity:** Statistical parity in classification it is usually measured by the Disparate Impact $DI(G_\theta)$, defined as in (7). However, enforcing statistical parity by enforcing independence between $G_\theta(X)$ and $A$ often produces unstable solutions as discussed by Krco et al. (2025) or Barrainkua et al. (2024), hence many authors propose to mitigate not only the mean but the whole distribution of the predicted probabilities $g_\theta(X) \in (0, 1)$ as in Risser et al. (2022), Le Gouic et al. (2020) or Chzhen et al. (2020), the same approach we have adopted in this work.

Figure 12 shows the results of training the model minimizing (9) for different values of the weight $\alpha$ and the privacy budget $\varepsilon$. Two main conclusions can be drawn. First, it confirms that the Wasserstein penalization approach mitigates the unfair biases present in the data set. We can see that, for increasing values of $\alpha$, the histograms of the scores conditioned on the value of the sensitive variable get closer, leading to a progressive reduction of biases, as seen in the decreasing values of the disparate impact, albeit at the expense of accuracy, as expected. The second important conclusion is that adding privacy does not significantly alter the results of the optimization. For different privacy budgets $\varepsilon$, both the histogram and the computed measures do not change much across the rows. Moreover, Figure 13 shows the training loss curve of the optimization for each value of $\alpha$ considered when $\varepsilon$ varies. Low values of $\varepsilon$ result in noisier versions of the loss curve, but they remain very close to the non-private counterpart.

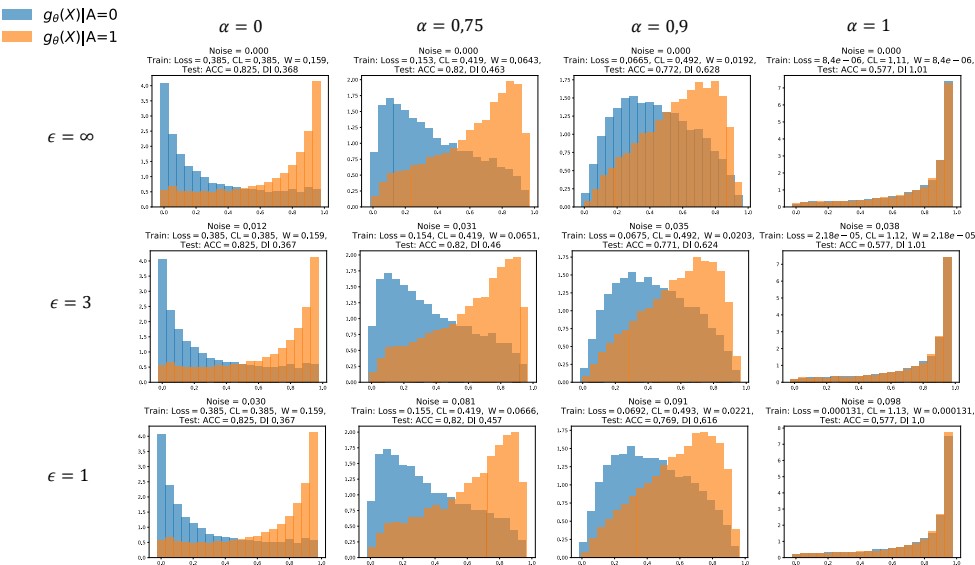

Figure 12: Histogram of the predicted probabilities $g_\theta(X)$ conditioned on the values of $A$ in the private classification model with SP regularization, for the different values of $\alpha$ and $\varepsilon$, and fixed $\delta = 0.1/n$. Above each graph we indicate the noise added at each step of DP-SGD to obtain the desired privacy level, the value of the loss (9) in the training procedure, together with the individual value of the classification loss (CL) and the distributional Wasserstein loss (W). Last line includes accuracy (ACC) and disparate impact (DI) of the classification rule $G_\theta$ computed with independent test data.

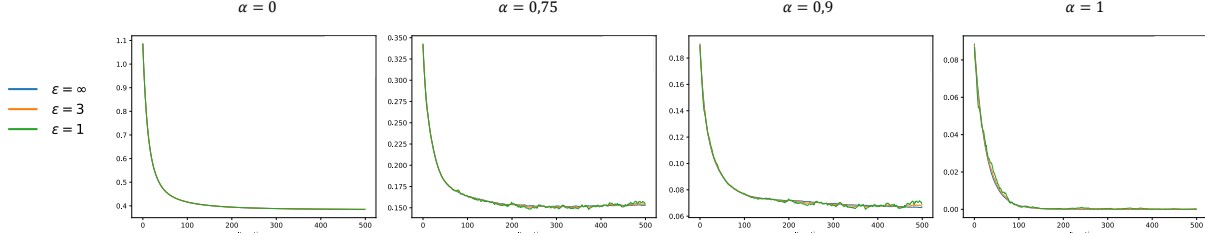

Figure 13: Training loss curve for the experiment of Figure 12. Each graph represents the training loss (9) for a fixed value of $\alpha$ along the iterations of DP-SGD, for the different values of $\varepsilon$.

**Equality of Odds:** To measure the unfairness of our a classification rule $G_\theta$, we consider the two following indexes:

$$EO_1(G_\theta) = \frac{\mathbb{P}(G_\theta(X) = 1|A = 0, Y = 1)}{\mathbb{P}(G_\theta(X) = 1|A = 1, Y = 1)} \ . \tag{13}$$

$$EO_0(G_\theta) = \frac{\mathbb{P}(G_\theta(X) = 1|A = 0, Y = 0)}{\mathbb{P}(G_\theta(X) = 1|A = 1, Y = 0)} \ . \tag{14}$$

Figure 14 shows the results of optimizing (10) for different values of $\alpha$ and $\varepsilon$. The histogram of the distribution of the predicted probabilities, $g_\theta(X)|A = 1, Y = j$ versus $g_\theta(X)|A = 0, Y = j$, illustrates the model's capability to minimize discrepancies between the distributions as $\alpha$ increases, as shown by the values of $EO_0, EO_1$. From Figure 14, we can also observe that private training has minimal impact on the model's fit

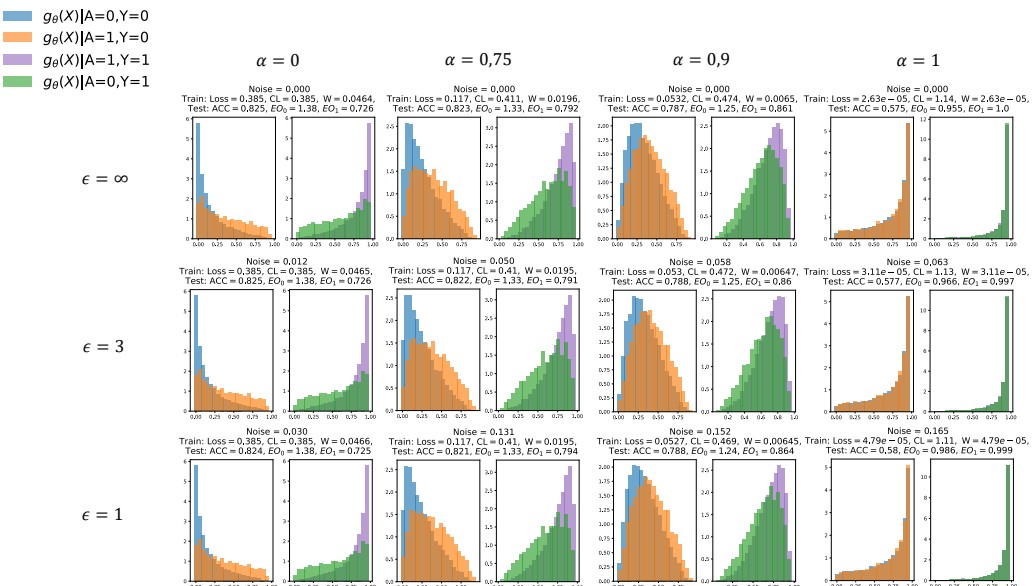

Figure 14: Histogram of the predicted probabilities $g_\theta(X)$ conditioned on the values of $A$ and $Y$ in the private classification model with EOO regularization, for the different values of $\alpha$ and $\varepsilon$, and fixed $\delta = 0.1/n$. Above each graph we indicate the noise added at each step of DP-SGD to obtain the desired privacy level, the value of the loss (10) in the training procedure, together with the individual value of the classification loss (L) and the Wasserstein loss (W). Last line includes accuracy, $EO_0$ and $EO_1$ indexes computed with independent test data.

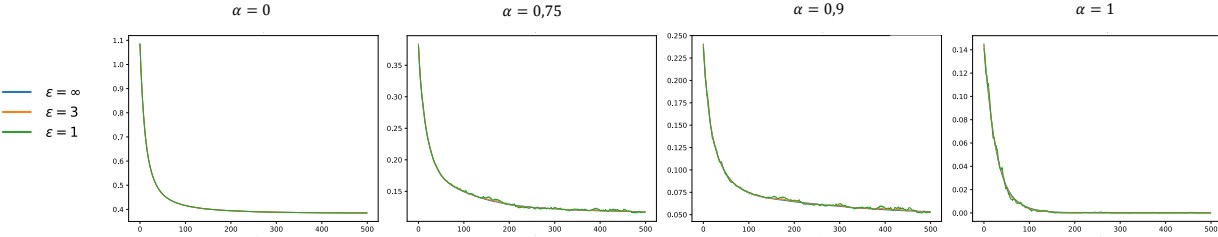

Figure 15: Training loss curve for the experiment of Figure 14. Each graph represents the training loss (10) for a fixed value of $\alpha$ along the iterations of DP-SGD, for the different values of $\varepsilon$.

across all values of $\alpha$. Similarly, it does not significantly affect the learning loss curve during optimization, as shown in Figure 15.

*Remark* C.3 (**Fair and private classification: comparison with the baseline**)*.* Following a reviewer's suggestion, we evaluate the privacy–utility tradeoff of our method on the Adult dataset and compare it with the results reported in Figure 2 of Lowy et al. (2023), including the baseline of Tran et al. (2021). For this comparison, we directly adopt their reported privacy–utility curves and reuse their preprocessing and evaluation setup, based on the notion of demographic parity, defined in the binary case as

$$\left| \mathbb{P}(\hat{Y} = 1 \mid A = 1) - \mathbb{P}(\hat{Y} = 1 \mid A = 0) \right|.$$

As in their work, we train a logistic regression model under $(\varepsilon, \delta)$-DP guarantees, for $\varepsilon \in \{0.5, 1, 3, 9\}$ and $\delta = 1/n^2$, and evaluate our method for varying $\alpha \in [0, 1]$. Results are shown in Figure 16. The comparison shows that (i) in regimes with large demographic parity violations, our method achieves better utility,

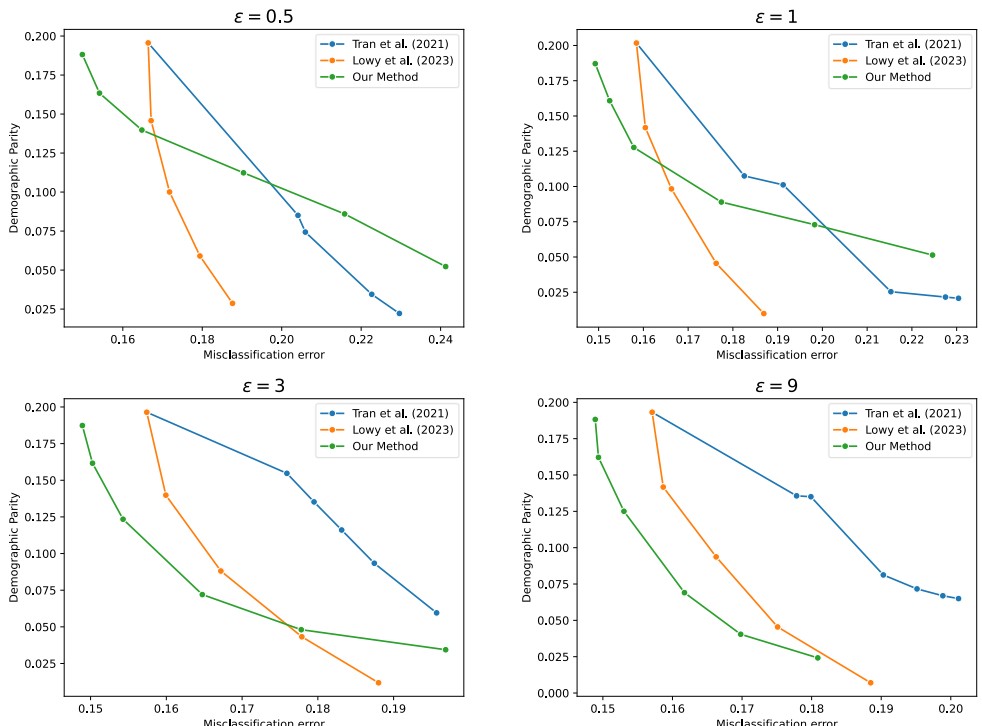

Figure 16: Privacy–utility tradeoff curves for different methods on the Adult dataset under the same experimental setting as Lowy et al. (2023), as detailed in Remark C.3. The curves for Lowy et al. (2023) and Tran et al. (2021) are averages over 15 runs obtained directly from the publicly available code of Lowy et al. (2023). Our curve represents the mean behavior over 5 runs.

possibly due to differences in implementation or hyperparameter tuning; and (ii) as $\alpha$ increases, our tradeoff becomes less favorable than that of Lowy et al. (2023), which is expected since our method enforces fairness through geometry-based constraints. Overall, the results suggest that our method is competitive in terms of the privacy–utility tradeoff.

## C.4 Regression

In our generating mechanism, the label $Y \in \{0, 1\}$ is defined as a set indicator function of a continuous response $Y^C \in [0, 1] \times [0, 1]$. From the data-generating process, it is easy to derive the distribution of $Y^C$ conditioned on the sensitive attribute. If $T_0$ denotes the triangle with vertices $(0, 0), (0, 1), (1, 0)$ and $T_1$ the triangles with vertices $(0, 1), (1, 1), (1, 0)$, then we know that $Y^C|A = j$ follows a mixture of the uniform distributions on $T_0$ and $T_1$, with weight $p$ in $T_0$ and $(1 - p)$ in $T_1$ if $A = 0$, and vice versa if $A = 1$. The aim of this experiment is to perform private and bi-dimensional fair regression over $Y^C$, which has never been considered before in the literature. To simplify our clipping bounds, we have centered our data to obtain a distribution in $[-1/2, 1/2] \times [-1/2, 1/2]$, and we have trained a two-layer neural network with hidden dimension 64, sigmoid activation function in the last layer, with the output centered by subtracting $(1/2, 1/2)$, and minimizing the loss (9), with mean square loss $\ell(g_\theta(x), y^C) = \|g_\theta(x) - y^C\|_2^2$ and applying the sliced Wasserstein penalization to the output layer $\varphi_\theta = g_\theta$.

Figure 17 shows the results of this experiment for different values of $\alpha$ and the privacy budget $\varepsilon$, with fixed $\delta = 0.1/n$, number of iterations $T = 1000$, clipping constants $C = 10$, $M = 1/\sqrt{2}$, $L = \sqrt{2}$, learning rate $= 0.05$ and number of projections in the Monte Carlo approximation $= 50$. From the visual inspection of the plots, we can appreciate that our statistical parity penalization helps to reduce the differences between the distributions of the predicted values. To aid visual inspection, we provide the values of the number of points over the diagonal for each class $A = 0$ and $A = 1$. If $g_\theta(x) = (g_\theta^1(x), g_\theta^2(x))$

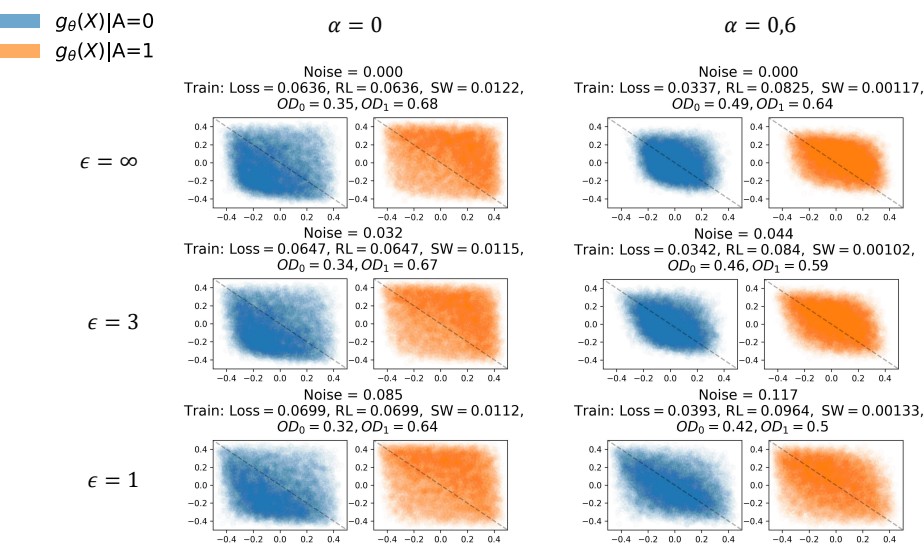

Figure 17: Predicted values $g_\theta(X)$ conditioned on the values of $A$ in the private regression model with SP regularization, for the different values of $\alpha$ and $\varepsilon$, and fixed $\delta = 0.1/n$. Above each graph we indicate the noise added at each step of DP-SGD to obtain the desired privacy level, the value of the loss (9) in the training procedure, together with the individual value of the regression loss (RL) and the sliced Wasserstein loss (SW). Last line includes the indexes $OD_0$ and $OD_1$.

$$OD_0 = \frac{\#\{X : g_\theta^2(X) > -g_\theta^1(X), A = 0\}}{n_0}$$

$$OD_1 = \frac{\#\{X : g_\theta^2(X) > -g_\theta^1(X), A = 1\}}{n_1}$$

Finally, Figure 18 shows the convergence of the loss curve for the different values of $\alpha$ and $\varepsilon$. As in the previous examples, the private loss curves are simply noisy versions of the non-private ones.

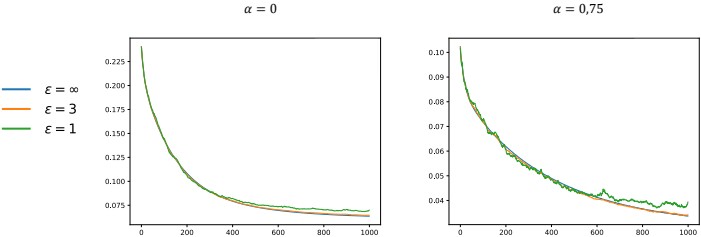

Figure 18: Training loss curve for the experiment of Figure 17. Each graph represents the training loss for a fixed value of $\alpha$ in (9) along the iterations of DP-SGD, for the different values of $\varepsilon$.

*Remark* C.4 (**Fair and private 1-D regression: comparison with the baseline**). As discussed in the related work section, the only existing method that allows for fair and private regression is the post-processing approach of Xian et al. (2024), which is only valid in the one-dimensional case. This method applies a post-processing step to an arbitrary private regressor $g$, consisting of three main stages: 1) approximate the conditional distributions $g_a = (g \mid A = a)$ using private histograms $h_a$; 2) transform each $h_a$ into $\tilde{h}_a$ within a Kolmogorov-Smirnov (KS) ball of radius $\beta$ around a "generalized barycenter" (see Xian et al. (2024) for precise definitions); and 3) use the optimal transport plan between $h_a$ and $\tilde{h}_a$ to define the random

post-processing maps. As noted in Section 7, the strength of our method lies in its versatility, enabling its application to a variety of problems, including novel ones. Our goal is not to outperform every existing method in each domain. For instance, in this case, we do not provide the fairness and utility guarantees of Xian et al. (2024). Nevertheless, the comparison helps contextualize our approach, and we show that our method remains empirically competitive with the procedure of Xian et al. (2024).

For the comparison, we use the same experimental setup as in Section 7. With the data described in Appendix C.2, let $Y^C = (Y_1^C, Y_2^C)$ and consider the one-dimensional regression problem of predicting $\tilde{Y} = Y_1^C + Y_2^C - 1$. From the discussion of the bidimensional fairness problem, it follows easily that the conditional densities of $\tilde{Y}$ given $A$ are

$$f_{\tilde{Y}|A=0} = \begin{cases} 2p(1+x) & \text{if } x \in [-1,0] \\ 2(1-p)(1-x) & \text{if } x \in [0,1] \end{cases}, \quad f_{\tilde{Y}|A=1} = \begin{cases} 2(1-p)(1+x) & \text{if } x \in [-1,0] \\ 2p(1-x) & \text{if } x \in [0,1] \end{cases}.$$

Note that if $p = 1/2$, both distributions coincide with the symmetric triangular distribution on $[-1, 1]$. In our example, with $p = 0.7$, the distribution conditioned on $A = 0$ is biased toward negative values, whereas the distribution conditioned on $A = 1$ is biased toward positive values. To address this regression problem, we use a two-layer fully connected neural network with hidden dimension 128 and compare the following approaches, where DP-SGD is trained in all cases with $T = 1000$ iterations, clipping constants $C = 10$, $M = 1/\sqrt{2}$, $L = 1$, and learning rate 0.02.

- **Baseline:** Train the NN with DP-SGD without fairness under $(2, 1/n)$-DP.

- **Xian et al. (2024):** Train the NN with DP-SGD under $(1, 1/n)$-DP, then apply the fair and private post-processing under $(1, 0)$-DP for various values of the fairness regularization parameter $\beta$ (radius of the KS ball around the barycenter, see Xian et al. (2024)). By composition, the mechanism satisfies $(2, 1/n)$-DP.

- **Our approach:** Train the model with DP-SGD under $(2, 1/n)$-DP, penalizing the squared Wasserstein distance between conditional distributions for different values of the regularization parameter $\alpha$.

Figure 5 shows the tradeoff between fairness and accuracy for all methods, averaged over 5 random seeds. Since our method enforces fairness via minimizing the Wasserstein distance, whereas Xian et al. (2024) uses the Kolmogorov-Smirnov distance, we report fairness both in terms of $W_2^2$ and KS distance, and accuracy using mean squared error (MSE). As shown, our method achieves a competitive tradeoff between utility and fairness under both metrics.

## C.5 Representation Learning

Finally, we present another completely novel application of our procedure: fair representation learning. Using the same data as before, the objective is to privately learn an encoder $\varphi_{\theta_a}$ and a decoder $\psi_{\theta_b}$ minimizing the mean squared error of the reconstructed values, penalized with the sliced Wasserstein distance to alleviate statistical parity unfairness present in the data. As in Section 6, we denote $\theta = (\theta_a, \theta_b)$, $\varphi_\theta = \varphi_{\theta_a}$ and $\psi_\theta = \psi_{\theta_b}$. Now, the encoder and decoder are defined as fully connected neural networks with two layers, hidden dimension 62 and bi-dimensional latent space. Defining $g_\theta = \psi_\theta \circ \varphi_\theta$, we consider the reconstruction error $\ell(g_\theta(x), x) = \|g_\theta(x) - x\|_2^2$ and the fairness penalty is now imposed over the encoded representations given by $\varphi_\theta$.

As usual, Figure 19 presents the result of training this model for different values of $\alpha$ and $\varepsilon$, with fixed $\delta = 0.1/n$, number of iterations $T = 500$ iterations, clipping values $C = 10, M = 2, L = \sqrt{2}$, learning rate $= 0.01$ and number of projections in the Monte Carlo approximation $= 50$. To assess the quality of the models, we have computed different comparative measures on an independent test sample. First, $RL_c$ denotes the reconstruction loss in the core part $X_{core}$, i.e. the first eight variables of $X$. The rest of the variables $X_{sp}$ are just a noisy version of $A$. Thus, $RL_c$ provides a measure of the error in the reconstruction

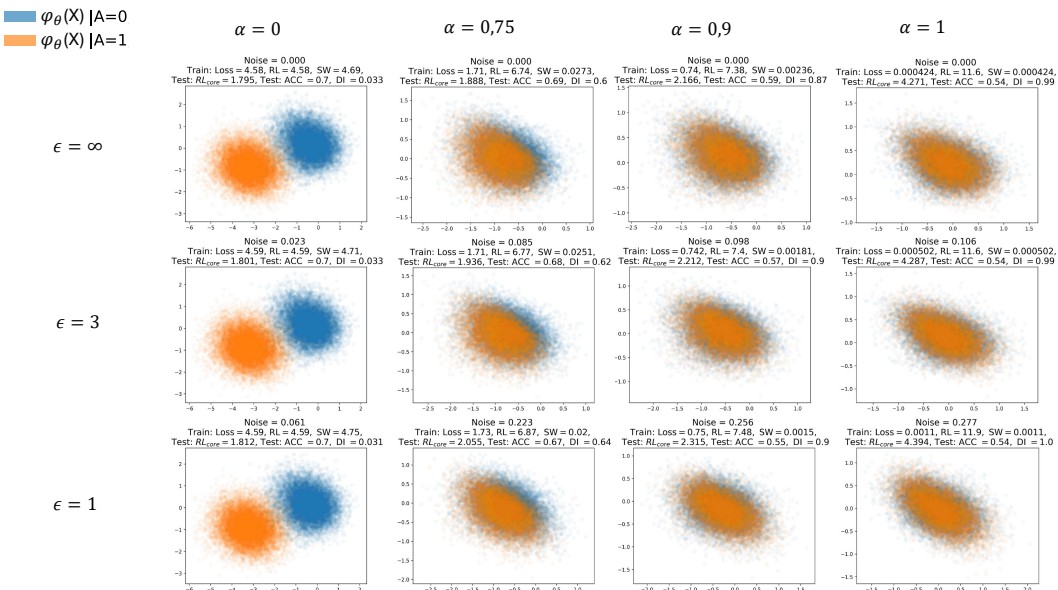

Figure 19: Plot of the latent space representations $\varphi_\theta(X)$ conditioned on the values sensitive attribute $A$ in the private representation learning model with SP regularization, for the different values of $\alpha$ and $\varepsilon$, and fixed $\delta = 0.1/n$. Above each plot we indicate the noise added at each step of DP-SGD to obtain the desired privacy level, the value of the loss (9) in the training procedure, together with the individual value of the reconstruction loss (RL) and the sliced Wasserstein loss (SW). Last line includes the reconstruction loss on the core variables ($RL_1$), accuracy and disparate impact on test data.

loss for the relevant part of the data, and Figure 19 shows that for increasing values of $\alpha$, even though the reconstruction loss increases significantly, the reconstruction associated with the core part is not affected much. The other measures computed on the test data are the accuracy and disparate impact of a simple logistic regression model trained on the encoded representation of a portion (60%) of the test data and evaluated on the remaining (40%). We observe that increasing values of $\alpha$ lead to values of the disparate impact index closer to 1, at the expense of a decrease in accuracy. Finally, we can infer from Figures 19 and 20 that privacy doesn't affect much to the results of the optimization.

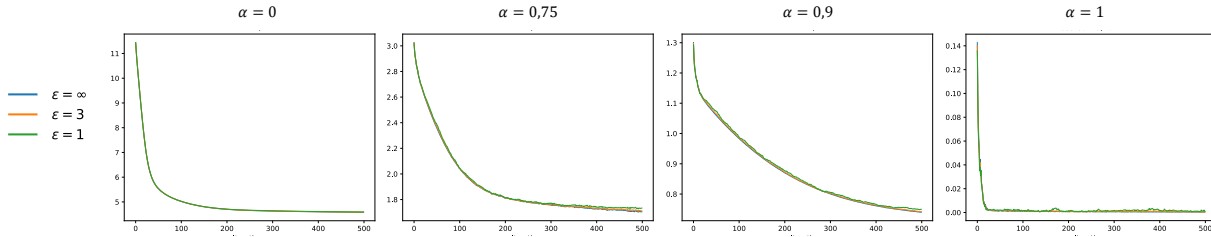

Figure 20: Training loss curve for the experiment of Figure 19. Each graph represents the training loss (9) for a fixed value of $\alpha$ along the iterations of DP-SGD, for the different values of $\varepsilon$.

### C.6 Randomness Strategy

To provide a deeper understanding of the previous results, it is important to detail the role of randomness across the experiments. For each of the four bias mitigation experiments, the training process was repeated for different values of the weight $\alpha$ and privacy budgets $\varepsilon \in \{\infty, 3, 1\}$. To ensure a fair comparison across $(\alpha, \varepsilon)$ pairs, we used the same random seed for all configurations. As a result, neural networks share the same weight initialization, the sampled batches are identical at each iteration, and the standard Gaussian noise used in private updates is also the same. With this setup, our experiments isolate the influence of $\alpha$ and $\varepsilon$ on the training process. The only differences across configurations stem from:

- **Clipping:** Applied only when $\varepsilon < \infty$.

- **Gradient composition:** The gradient is a linear combination of the classification gradient and the sliced Wasserstein gradient, weighted by $\alpha$.

- **Noise multiplier:** The standard Gaussian noise is scaled by a factor that depends on both $\alpha$ (via sensitivity) and the required privacy budget $\varepsilon$.

To illustrate the effectiveness of this isolation strategy, we repeated the Statistical Parity classification experiment—chosen for its simplicity and clarity—for 10 different random seeds. For each seed, the model was trained across the 12 combinations of $(\alpha, \varepsilon)$ used in Figure 12.

The first row of Figure 21 shows the average training loss for each $(\alpha, \varepsilon)$ pair, with $\pm 2\sigma$ confidence bands. The variability shows that if the randomness setup in Figures 12 and 13 had not been fixed, significant fluctuations in the training loss across different values of $\varepsilon$ would have appeared, contrary to what is shown in Figure 13. The isolation effect of our randomness setup is even more evident in the second row of Figure 21, which shows the differences between the private ($\varepsilon = 3, 1$) and non-private ($\varepsilon = \infty$) losses computed using the same random seed, with the same $\pm 2\sigma$ confidence bands computed over the 10 seeds. These differences stay within a narrow range around zero, indicating that, in this classification task, adding privacy mainly introduces a small amount of noise around the non-private trajectory, supporting the conclusion obtained from Figure 13.

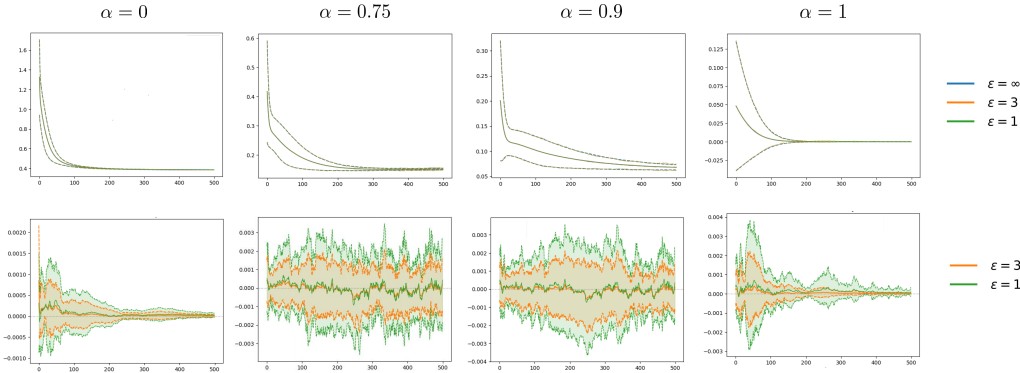

Figure 21: **Top**: The solid line represents the average training loss (9) over 10 random seeds for different values of $\varepsilon$ and $\alpha$. Dotted lines indicate the average $\pm 2$ standard deviations. For a given $\alpha$, the curves corresponding to different $\varepsilon$ values nearly overlap. **Bottom**: Differences between private ($\varepsilon = 3, 1$) and non-private ($\varepsilon = \infty$) losses, computed using the same random seed for each pair, and averaged over 10 different seeds. Shaded bands represent $\pm 2$ standard deviations.

# D  Detailed comparison with Rakotomamonjy & Ralaivola (2021)

In this section, we compare our method to the approach of Rakotomamonjy & Ralaivola (2021) in two ways: first, through a simple conceptual example highlighting the broader applicability of our method, and second, via a numerical comparison on an example where both methods can be applied.

## D.1  Conceptual Comparison

To simplify notations, we restrict ourselves to the simple case $h_\theta = I_d$. That is, we assume we have samples $\mathbf{X} = (x_1, \ldots, x_n)^T \in \mathcal{X}^n$ and $\mathbf{Z} = (z_1, \ldots, z_m)^T \in \mathbb{R}^{m \times d}$, and we want to learn a map $g_\theta : \mathcal{X} \to \mathbb{R}^d$, parametrized by $\theta \in \Theta$, minimizing $W_2^2(g_\theta(\mathbf{X}), \mathbf{Z})$ using a first-order algorithm, where with a slight abuse of notation we denote $g_\theta(\mathbf{X}) = (g_\theta(x_1), \ldots, g_\theta(x_n))^T \in \mathbb{R}^{n \times d}$.

The work of Rakotomamonjy & Ralaivola (2021) relies on the following simple idea. $SW_2^2(g_\theta(\mathbf{X}), \mathbf{Z})$ can be approximated via Monte Carlo sampling over $k$ i.i.d. directions $\vartheta = (\vartheta_1, \ldots, \vartheta_k) \in \mathbb{R}^{d \times k}$ drawn uniformly from $\mathbb{S}^{d-1}$ by

$$\frac{1}{k} \sum_{i=1}^{k} W_2^2(g_\theta(\mathbf{X}) \cdot \vartheta_i, \mathbf{Z} \cdot \vartheta_i).$$

For each $i$, the only information used about the data is $g_\theta(\mathbf{X}) \cdot \vartheta_i$ and $\mathbf{Z} \cdot \vartheta_i$. Their proposal releases $g_\theta(\mathbf{X}) \cdot \vartheta + \varepsilon$ and $\mathbf{Z} \cdot \vartheta + \xi$, where $\epsilon = (\epsilon_1, \ldots, \epsilon_n) \in \mathbb{R}^{n \times d}$ and $\xi = (\xi_1, \ldots, \xi_m) \in \mathbb{R}^{m \times d}$ have i.i.d. $N(0, \sigma^2)$ entries. As usual, the noise required to achieve a given privacy budget (w.r.t. both $\mathbf{X}$ and $\mathbf{Z}$) is proportional to the sensitivity of the projection, which does not decrease with the sample size and only vanishes as $k/d \to 0$. By post-processing,

$$DP_\sigma SW_2^2(g_\theta(\mathbf{X}), \mathbf{Z}) := \frac{1}{k} \sum_{i=1}^{k} W_2^2\big(g_\theta(\mathbf{X}) \cdot \vartheta_i + \epsilon_i, \mathbf{Z} \cdot \vartheta_i + \xi_i\big)$$

inherits the privacy guarantees (w.r.t. both $\mathbf{X}$ and $\mathbf{Z}$). In our optimization problem, however, we are interested in privacy guarantees for its gradient, namely

$$\mathcal{M}_R(\mathbf{X}, \mathbf{Z}; \theta) := \nabla_\theta DP_\sigma SW_2^2(g_\theta(\mathbf{X}), \mathbf{Z}).$$

Applying the chain rule to (1), $\mathcal{M}_R(\mathbf{X}, \mathbf{Z}; \theta)$ can be obtained by post-processing $g_\theta(\mathbf{X}) \cdot \vartheta + \epsilon$, $\mathbf{Z} \cdot \vartheta + \xi$, $\vartheta$, together with $\mathcal{J}_\theta g_\theta(\mathbf{X})$. If privacy is required only w.r.t. $\mathbf{Z}$, such as in a simple data generation problem where $g_\theta$ maps noise $X$ into samples $\mathbf{Z}$, then $\mathcal{J}_\theta g_\theta(\mathbf{X})$ is public, and, using the fix of Remark 1 of Greenewald et al. (2024), $\mathcal{M}_R(\mathbf{X}, \mathbf{Z}; \theta)$ inherits the privacy guarantees. However, no privacy guarantees can be derived w.r.t. $\mathbf{X}$ due to the presence of $\mathcal{J}_\theta g_\theta(\mathbf{X})$.

This can be illustrated with the following example. Assume $d = 1$ (then $k = 1$), $\mathbf{X} \in [-L/2, L/2]^n$ for some $L > 2$, $\mathbf{Z} \in [0, 1]^m$, $\theta = (\theta_1, \theta_2) \in \mathbb{R}^2$, and consider the hard sigmoid function $g_\theta(x) = \max(0, \min(1, \theta_1 x + \theta_2)) \in [0, 1]$, whose gradient is

$$\nabla_\theta g_\theta(x) = \begin{cases} (x, 1) & \text{if } -\frac{\theta_2}{\theta_1} < x < \frac{1 - \theta_2}{\theta_1} \\ (0, 0) & \text{otherwise} \end{cases}$$

We can interpret this as fitting a (hard) logistic regression model. We focus on the simple case $m = 1$, where denoting $a_i = z_1 + \xi_1 - \epsilon_i$,

$$\mathcal{M}_R(\mathbf{X}, \mathbf{Z}; \theta) = \nabla_\theta\Big(\frac{1}{n} \sum_{i=1}^{n} (g_\theta(x_i) - a_i)^2\Big) = \frac{2}{n} \sum_{i=1}^{n} (g_\theta(x_i) - a_i) \nabla_\theta g_\theta(x_i).$$

For instance, if $\theta = (1, 0)$ and $\mathbf{X} \in ([-L/2, L/2] \setminus [0, 1])^n$, then $\mathcal{M}_R(\mathbf{X}, \mathbf{Z}; \theta) = \delta_0$. However, if $\tilde{\mathbf{X}} \sim \mathbf{X}$ is obtained by replacing $x_1$ with $x_1' \in (0, 1)$, then $\mathcal{M}_R(\tilde{\mathbf{X}}, \mathbf{Z}; \theta) \neq \delta_0$, and in fact $P(\mathcal{M}_R(\tilde{\mathbf{X}}, \mathbf{Z}; \theta) = 0) = 0$, making it impossible to provide any non-trivial privacy guarantees with respect to $\mathbf{X}$. Our method solves

this problem as it provides guarantees for both $\mathbf{X}$ and $\mathbf{Z}$. Indeed, Theorem 4.1 directly applies with $M = 1$, $L_1 = L$, and $L_2 = 0$, providing private guarantees for

$$\mathcal{M}(\mathbf{X}, \mathbf{Z}; \theta) := \nabla_\theta W_2^2(g_\theta(\mathbf{X}), \mathbf{Z}) + N(0, \sigma^2)$$

w.r.t. both $\mathbf{X}$ and $\mathbf{Z}$, and the required amount of noise to achieve a fixed privacy budget scales as the inverse of the sample size.

## D.2 Numerical Comparison

In this second part, we consider the sliced version of the same problem, in the setting where privacy guarantees are required only with respect to $\mathbf{Z}$. That is, given a private dataset $\mathbf{Z} \in \mathcal{Z}^n$ with target distribution $Q$, the goal is to learn, in a privacy-preserving manner, a map $g_\theta$ that minimizes

$$\mathcal{L}(\theta) = SW_2^2\big(g_\theta \# P_\mathbf{X}, P_\mathbf{Z}\big) \ ,$$

where $\mathbf{X} \in \mathcal{X}^n$ is a dataset of noise drawn from a source distribution $P$ from which sampling is straightforward. As discussed above, this provides a simple setting in which both our method and the baseline of Rakotomamonjy & Ralaivola (2021) can be applied.

**Baseline of Rakotomamonjy & Ralaivola (2021) with the fix of Greenewald et al. (2024).** As mentioned before, the original privacy analysis of Rakotomamonjy & Ralaivola (2021) is contested. In our experiments, we therefore use the fix to the privacy analysis provided in Greenewald et al. (2024) (Theorem 1). For this baseline, we also incorporate standard privacy amplification by sub-sampling for fixed batch size without replacement for the replacement neighboring relation, as stated in Steinke (2022). A dataset is sub-sampled at first, and we then use this sub-sampled dataset until the end of the training procedure. As described in Algorithm 1 in Greenewald et al. (2024), the same amount of dummy privacy noise is added to the points in $g_\theta \# P_\mathbf{X}$ in order to learn the de-convolution map and not the map to the convoluted distribution.

**Data and results.** We define the dataset $\mathbf{Z}$ as $n = 100000$ samples drawn uniformly from the circumference of the circle with radius $3/4$, and the dataset $\mathbf{X}$ as an equal number of samples drawn from the standard Gaussian distribution in $\mathbb{R}^2$. The function $g_\theta$ is defined as a fully connected neural network with an input dimension 2, three hidden layers with dimensions $(128, 64, 64)$, and an output dimension 2. Figure 22 shows the evolution of the matching problem at different training steps. Thanks to Theorem A.1, our methodology provides privacy guarantees for both the fixed variable $\mathbf{Z}$ and the *trained* variable $\mathbf{X}$, in the sense that $g_\theta$ is applied to $\mathbf{X}$. Above each plot, we can see the iteration number, the value of the loss, and the privacy budget $\varepsilon$ for both $\mathbf{X}$ and $\mathbf{Z}$, at each training step. The optimization parameters are $\delta = 0.1/n$, batch size $= 10000$, learning rate $= 0.0075$, number of projections in the Monte Carlo approximation $= 50$, clipping values $M = 1$ and $L = 2\sqrt{2}$ (imposed using the suboptimal approach described in Remark A.2). To ensure more stable results, once we have privatized the gradient by adding noise, we clip the gradient again to improve the method's stability with the same clipping constants. Note that this step preserves privacy due to the post-processing property.

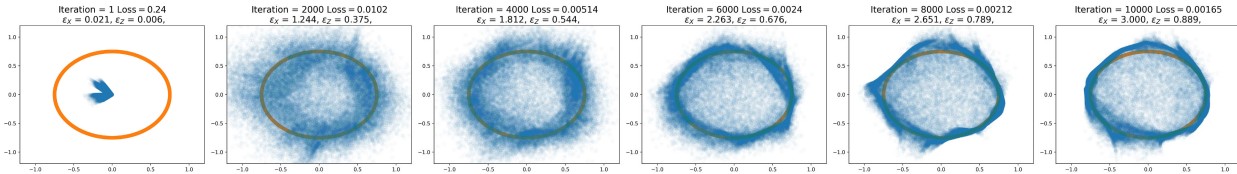

Figure 22: Data generation experiment. Samples from $X$ are represented in blue, samples from $Z$ in orange. Above each graph, we can see the iteration, the value of the loss, and privacy budgets w.r.t. the variables $X$ and $Z$.

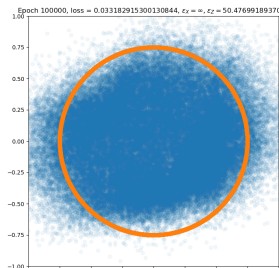 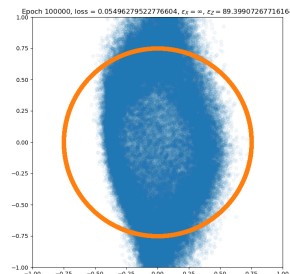 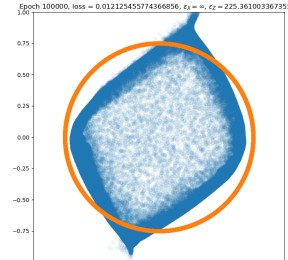

Figure 23: Baseline for the data generation experiment. Samples from $X$ are represented in blue, samples from $Z$ in orange. Above each graph, we can see the iteration, the value of the loss, and privacy budgets w.r.t. the variables $X$ and $Z$.

The results for the method proposed by Rakotomamonjy & Ralaivola (2021) are shown in Figure 23. For a fair comparison, we retained the same models and hyperparameters as in our approach, with the exception of reducing the number of projection directions to 15, as this adjustment improved the baseline's performance (trading increased bias for reduced privacy-related variance). Despite this tuning, the baseline performs poorly on this task, as shown by the substantially higher privacy budgets required to achieve acceptable results. Since the original paper (Rakotomamonjy & Ralaivola, 2021) did not include visual experiments of this kind, it remains unclear whether the poor performance is intrinsic to the method, a consequence of the correction proposed by Greenewald et al. (2024), or a limitation of our implementation.

## E  Computational Details

All experiments were conducted on a local server equipped with an NVIDIA RTX A6000 GPU with 48 GB of memory. The core of our implementation was developed in Python using the JAX library (Bradbury et al., 2018). The code represents a preliminary version aimed at illustrating the theoretical capabilities of our approach. We believe that a more refined implementation could lead to significantly faster execution. Notably, the current code does not leverage `jit`, the main just-in-time compilation tool in JAX, which could substantially reduce training time.

Nevertheless, the methodology presents inherent computational bottlenecks that limit the efficiency of our methodology. These bottlenecks are closely related to standard computational challenges in private learning. For illustration, assume the setting described in Remark 4.5 with output dimension $d = 1$, and let $r$ denote the number of parameters in $\theta$. Consider batches $\mathbf{X}_b$ and $\mathbf{Z}_b$ of size $m$. Using the same notation as in Remark 4.5, we aim to compute a private version of:

$$\underbrace{\nabla_\theta W_2^2(g_\theta(\mathbf{X}_b), \mathbf{Z}_b)}_{1 \times r} = \underbrace{\nabla_{g_\theta} W_2^2(g_\theta(\mathbf{X}_b), \mathbf{Z}_b)}_{1 \times m} \times \underbrace{\mathcal{J}_\theta g_\theta(\mathbf{X}_b)}_{m \times r} \tag{15}$$

If privacy were not a concern, standard automatic differentiation would allow for efficient computation of the left-hand side, leveraging common operations and shared computational graphs. However, to enforce differential privacy, we need to adopt the inner clipping approximation described in Eq. (5). The expression (5) can be viewed as a product between a clipped version of $\nabla_{g_\theta} W_2^2(g_\theta(\mathbf{X}_b), \mathbf{Z}_b)$ and a clipped version of $\mathcal{J}_\theta g_\theta(\mathbf{X}_b)$. The main computational bottleneck arises from the need to compute the matrix $\mathcal{J}_\theta g_\theta(\mathbf{X}_b) \in \mathbb{R}^{m \times r}$. A similar challenge is usually encountered in differentially private optimization, where, if $g_\theta(x)$ represents the individual loss, we need to compute $\frac{1}{m} \sum_{x \in \mathbf{X}_b} \text{clipp}_C\big(\nabla_\theta g_\theta(x)\big)$. Two key issues arise:

(i) Memory cost: If $m$ and $r$ are large, storing the full Jacobian $\mathcal{J}_\theta g_\theta(\mathbf{X}_b)$ can become infeasible. In standard differential privacy settings, this is typically addressed by reducing the batch size or by

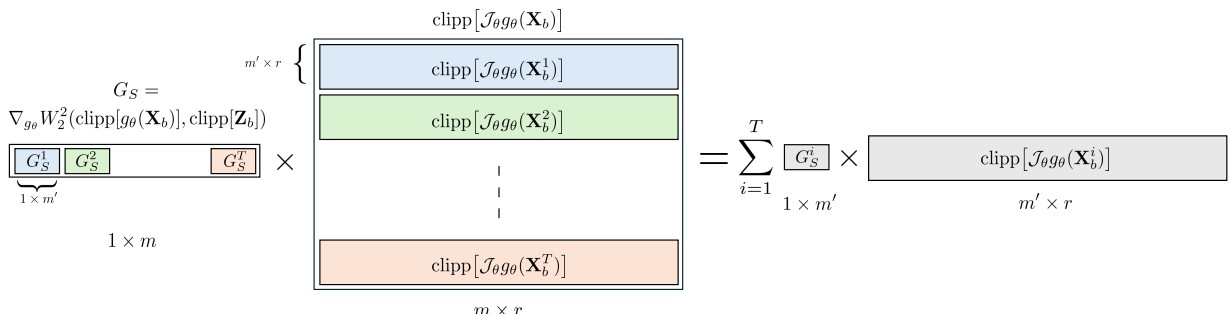

Figure 24: Naive approach used for computing the clipped gradient $\nabla_\theta^{M,L} W_2^2(g_\theta(\mathbf{X}_b), \mathbf{Z}_b)$. To avoid computing the full Jacobian matrix in the batch $\mathcal{J}_\theta g_\theta(\mathbf{X}_b)$, we: (i) split the batch $\mathbf{X}_b = (\mathbf{X}_b^1, \ldots, \mathbf{X}_b^T)$, (ii) Compute $G_S$, the gradient of the Wasserstein distance with clipping applied over the full batches $\mathbf{X}_b$ and $\mathbf{Z}_b$, as specified in Eq. (5). (iii) split $G_S = (G_S^1, \ldots, G_S^T)$, and (iv) iteratively compute $\mathcal{J}_\theta g_\theta(\mathbf{X}_b^i)$, premultiply by $G_S^i$, and aggregate.

computing gradients across individual data points (or across smaller subbatches) sequentially and then aggregating them.

(ii) Loss of GPU efficiency: These naive strategies undermine the advantages of batch computation on GPUs, leading to a significant increase in training time, as discussed in Goodfellow (2015) or Lee & Kifer (2021). In particular, within the JAX framework, computing the full Jacobian via `jax.jacobian` is substantially more efficient than computing all the per-sample gradients with `jax.grad`.

In our setting, using small batch sizes $m$ is not suitable, as this would introduce substantial bias in the estimation of the Wasserstein distance. However, splitting the batch into smaller subbatches remains a viable strategy—if done carefully—as explained in Figure 24. The same discussion extends naturally to the general case of the sliced Wasserstein distance. If we flatten $g_\theta$—i.e., we see $g_\theta(\mathbf{X}_b) \in (\mathbb{R}^d)^m$ as an element of $\mathbb{R}^{dm}$—then the chain rule yields the same decomposition as in Equation (15), replacing $W_2$ with $SW_2$ (or its Monte Carlo approximation). In this setting, the dimension of the hidden dimension in the vector–matrix product becomes $dm$ instead of $m$, further increasing the computational burden.

To quantify the runtime impact in a simple example, we fix $k = 30, n = 2^{12}, d = 2$, and vary the width $h$ of a one-hidden-layer network $g_\theta$ (with $4h$ parameters). Table 6 compares the average time (over 100 runs) to compute the sliced Wasserstein gradient using standard autodiff (jax.grad) versus our clipping strategy, which splits the computation into $T = \max(h/32, 1)$ smaller Jacobians using the formula in Figure 24. With clipping, each block computes a Jacobian of size at most $2^{20}$, which fits in our GPU memory. We find that clipping introduces no overhead when memory allows full Jacobians, and can even be slightly faster, thanks to our use of an explicit gradient formula rather than relying on generic autodiff through complex graphs.

Finally, as a practical reference, the training time for a single DP-SWAE run in the same experimental setting as in Section 6 (without sweeping over $\varepsilon$) is approximately 50 minutes with our implementation. As an SWAE baseline, we run the same number of iterations without clipping or noise addition, computing the gradient of the sliced Wasserstein distance directly via automatic differentiation. This baseline avoids the Jacobian computation required by our inner-clipping strategy, which is the main computational bottleneck discussed above. In this setup, SWAE takes approximately 13 minutes.

## F Counterexample for General Cost Functions

In this section, we demonstrate that we cannot bound in general the sensitivity of the gradient if we use the Wasserstein loss function $W_p$, for general $p \geq 1$. Following the notation of Theorem 4.1, we denote by

Table 6: Comparison of the required time to compute $\nabla_\theta W_2^2(g_\theta(\mathbf{X}_b))$.

| h | T | AutoGrad | Loop | Loop/Autograd |
|---|---|---|---|---|
| 8 | 1 | 0.0311s | 0.0303s | 0.97 |
| 16 | 1 | 0.0372s | 0.0294s | 0.79 |
| 32 | 1 | 0.0608s | 0.0379s | 0.62 |
| 128 | 4 | 0.0256s | 0.0682 s | 2.66 |
| 512 | 16 | 0.0239s | 0.1961 s | 8.20 |
| 2048 | 64 | 0.0213s | 0.7890s | 36.96 |
| 8192 | 256 | 0.0320s | 3.9563s | 123.81 |

$\mathbf{X} = \{x_1, \ldots, x_n\} \subset \mathcal{X}^n$ the private dataset, $\mathbf{Z} = \{z_1, \ldots, z_n\} \subset \mathbb{R}$ the non-private dataset, and $P_\mathbf{X}, P_\mathbf{Z}$ the associated empirical distributions. Given $g_\theta : \mathcal{X} \to \mathbb{R}$ depending on the parameter $\theta$, and considering $h_\theta = I_d$, we study the sensitivity of $\Phi(\mathbf{X}) = \nabla_\theta W_p(g_\theta \# P_\mathbf{X}, P_\mathbf{Z})$, for the particular values of

- $\mathbf{X} = \{x_1, \ldots, x_n\}$ with $x_i = \frac{i}{n}$, $i \in [n]$

- $\tilde{\mathbf{X}} = \{\tilde{x}_1, \ldots, \tilde{x}_n\}$ with $\tilde{x}_i = \frac{i-1}{n}$, $i \in [n]$

- $\mathbf{Z} = \{z_1, \ldots, z_n\}$, with $z_i = \frac{2i-1}{2n}$, $i \in [n]$.

- $g_\theta(x) = x + \theta$

For this particular choice, $\mathbf{X} \sim_1 \tilde{\mathbf{X}}$, and the assumptions of Theorem 4.1 are verified for certain constants (once we restrict the domain of $\theta$). From the quantile representation, it is easy to compute

$$W_p(g_\theta \# P_\mathbf{X}, P_\mathbf{Z}) = \left(\frac{1}{n} \sum_{i=1}^n |x_i + \theta - z_i|^p\right)^{1/p} = \left(\frac{1}{n} \sum_{i=1}^n \left|\frac{1}{2n} + \theta\right|^p\right)^{1/p}$$

$$= \begin{cases} \frac{1}{2n} + \theta & \text{if } \theta > -\frac{1}{2n} \\ -\theta - \frac{1}{2n} & \text{if } \theta \leq -\frac{1}{2n} \end{cases}$$

$$W_p(g_\theta \# P_{\tilde{\mathbf{X}}}, P_\mathbf{Z}) = \left(\frac{1}{n} \sum_{i=1}^n |\tilde{x}_i + \theta - z_i|^p\right)^{1/p} = \left(\frac{1}{n} \sum_{i=1}^n \left|-\frac{1}{2n} + \theta\right|^p\right)^{1/p}$$

$$= \begin{cases} -\frac{1}{2n} + \theta & \text{if } \theta > \frac{1}{2n} \\ \theta - \frac{1}{2n} & \text{if } \theta \leq \frac{1}{2n} \end{cases}$$

Therefore, by setting $\theta = 0$, we observe that the derivatives are $\Phi(\mathbf{X}) = 1$ and $\Phi(\tilde{\mathbf{X}}) = -1$. Consequently, $\Delta_2(\Phi) \geq 2$, indicating that the sensitivity does not decrease with the sample size $n$.

# G Proofs

## G.1 Proofs of Section 3

*Proof of Proposition 3.1.* If we denote by $F$ and $G$ the distribution functions of $P_\mathbf{U}$ and $P_\mathbf{V}$, we know that

$$W_2^2(P_\mathbf{U}, P_\mathbf{V}) = \int_0^1 (F^{-1}(t) - G^{-1}(t))^2 dt$$

$$= \int_0^1 \left(\sum_{i=1}^n U_{(i)} I\left(\frac{i-1}{n} < t \leq \frac{i}{n}\right) - \sum_{j=1}^m V_{(j)} I\left(\frac{j-1}{m} < t \leq \frac{j}{n}\right)\right)^2 dt$$

$$= \sum_{i=1}^n \sum_{j=1}^m (U_{(i)} - V_{(j)})^2 \int_0^1 I\left(\frac{i-1}{n} < t \leq \frac{i}{n}, \frac{j-1}{m} < t \leq \frac{j}{n}\right) dt$$

$$= \sum_{i=1}^{n} \sum_{j=1}^{m} (U_{(i)} - V_{(j)})^2 R_{i,j}$$

$$= \sum_{i=1}^{n} \sum_{j=1}^{m} (U_{(\sigma(i))} - V_{(\tau(j))})^2 R_{\sigma(i),\tau(j)}$$

$$= \sum_{i=1}^{n} \sum_{j=1}^{m} (U_i - V_j)^2 R_{\sigma(i),\tau(j)} \, ,$$

where the third equality follows from the fact that exactly one element in each sum is different from 0, and the fifth equality follows from reindexing the sum. $\qquad\square$

## G.2  Proofs of Section 4

*Proof of Theorem 4.1.* The second part of the theorem follows directly from the first and from the definition of the neighboring relation $\sim_2$ in $\mathcal{X}^n \times \mathcal{Z}^m$. To prove the first, consider two neighboring datasets under the substitution relation $\mathbf{X} \sim_1 \tilde{\mathbf{X}}$ in $\mathcal{X}^n$. We can assume without loss of generality that the datasets differ on the first observation $\tilde{x}_1 \neq x_1$. For ease of notation, denote $\tilde{\mathbf{X}} = (\tilde{x}_1, \ldots, \tilde{x}_n)$, even though $\tilde{x}_i = x_i$ for $i \neq 1$. Along this proof, we will define $U_i := g_\theta(x_i)$ and $\tilde{U}_i := g_\theta(\tilde{x}_i)$ for each $i \in [n]$, and $V_j := h_\theta(z_j)$ for $j \in [m]$. Again, $U_i = \tilde{U}_i$ for every $i \neq 1$. Define now the rank permutations $\sigma, \tilde{\sigma}$ and $\tau$ such that

$$\begin{aligned} U_i &= U_{(\sigma(i))} \,, & i \in [n] \,, \\ \tilde{U}_i &= \tilde{U}_{(\tilde{\sigma}(i))} \,, & i \in [n] \,, \\ V_j &= V_{(\tau(j))} \,, & j \in [m] \,. \end{aligned}$$

Denote $\mathbf{U} = (U_1, \ldots, U_n)$ and $\mathbf{V} = (V_1, \ldots, V_m)$. Proposition 3.2 ensures if we define $P_{\mathbf{U}} = g_\theta \# P_{\mathbf{X}} = \frac{1}{n} \sum_{i=1}^{n} \delta_{U_i}$ and $P_{\mathbf{V}} = h_\theta \# P_{\mathbf{Z}} = \frac{1}{m} \sum_{i=1}^{m} \delta_{V_j}$ , then

$$\nabla_{U,V} W_2^2(P_{\mathbf{U}}, P_{\mathbf{V}}) = \left( \left( 2 \sum_{j=1}^{m} R_{\sigma(i),\tau(j)} (U_i - V_j) \right)_{i \in [n]}, \left( 2 \sum_{i=1}^{n} R_{\sigma(i),\tau(j)} (V_j - U_i) \right)_{j \in [m]} \right) .$$

Applying the chain rule, we obtain that

$$\nabla_\theta W_2^2(g_\theta \# P_{\mathbf{X}}, h_\theta \# P_{\mathbf{Z}}) = 2 \sum_{i=1}^{n} \sum_{j=1}^{m} R_{\sigma(i),\tau(j)} (U_i - V_j) \nabla_\theta g_\theta(x_i) + 2 \sum_{j=1}^{m} \sum_{i=1}^{n} R_{\sigma(i),\tau(j)} (V_j - U_i) \nabla_\theta h_\theta(z_j) .$$

Similarly, for the dataset $\tilde{\mathbf{X}}$ we get

$$\nabla_\theta W_2^2(g_\theta \# P_{\tilde{\mathbf{X}}}, h_\theta \# P_{\mathbf{Z}}) = 2 \sum_{i=1}^{n} \sum_{j=1}^{m} R_{\tilde{\sigma}(i),\tau(j)} (\tilde{U}_i - V_j) \nabla_\theta g_\theta(\tilde{x}_i) + 2 \sum_{j=1}^{m} \sum_{i=1}^{n} R_{\tilde{\sigma}(i),\tau(j)} (V_j - \tilde{U}_i) \nabla_\theta h_\theta(z_j) .$$

Therefore,

$$\|\nabla_\theta W_2^2(g_\theta \# P_{\mathbf{X}}, h_\theta \# P_{\mathbf{Z}}) - \nabla_\theta W_2^2(g_\theta \# P_{\tilde{\mathbf{X}}}, h_\theta \# P_{\mathbf{Z}})\|_2 \leq$$

$$\leq 2 \left\| \sum_{i=1}^{n} \sum_{j=1}^{m} R_{\sigma(i),\tau(j)} (U_i - V_j) \nabla_\theta g_\theta(x_i) - \sum_{i=1}^{n} \sum_{j=1}^{m} R_{\tilde{\sigma}(i),\tau(j)} (\tilde{U}_i - V_j) \nabla_\theta g_\theta(\tilde{x}_i) \right\|_2 \qquad (16)$$

$$+ 2 \left\| \sum_{j=1}^{m} \sum_{i=1}^{n} R_{\sigma(i),\tau(j)} (V_j - U_i) \nabla_\theta h_\theta(z_j) - \sum_{j=1}^{m} \sum_{i=1}^{n} R_{\tilde{\sigma}(i),\tau(j)} (V_j - \tilde{U}_i) \nabla_\theta h_\theta(z_j) \right\|_2 . \qquad (17)$$

The term (17) is easier to bound, since the values inside $\nabla_\theta h_\theta(\cdot)$ coincide. First, note that

$$\sum_{j=1}^{m} R_{i,j} = \frac{1}{n} \,, \; \forall \, i \in [n] \quad \text{and} \quad \sum_{i=1}^{n} R_{i,j} = \frac{1}{m} \,, \; \forall \, j \in [m] \,. \qquad (18)$$

The triangular inequality, the assumption $\|\nabla_\theta h_\theta(z)\| \leq L_2$ for every $z, \theta$ and the previous property allow us to derive the following inequalities

$$(17) = 2 \left\| \sum_{j=1}^{m} V_j \nabla_\theta h_\theta(z_j) \Big( \sum_{i=1}^{n} R_{\sigma(i),\tau(j)} - \sum_{i=1}^{n} R_{\tilde{\sigma}(i),\tau(j)} \Big) - \sum_{j=1}^{m} \nabla_\theta h_\theta(z_j) \Big( \sum_{i=1}^{n} U_i R_{\sigma(i),\tau(j)} - \sum_{i=1}^{n} \tilde{U}_i R_{\tilde{\sigma}(i),\tau(j)} \Big) \right\|_2$$

$$\leq 2 \sum_{j=1}^{m} \left\| \nabla_\theta h_\theta(z_j) \Big( \sum_{i=1}^{n} U_i R_{\sigma(i),\tau(j)} - \sum_{i=1}^{n} \tilde{U}_i R_{\tilde{\sigma}(i),\tau(j)} \Big) \right\|_2$$

$$\leq 2L_2 \sum_{j=1}^{m} \left| \sum_{i=1}^{n} U_i R_{\sigma(i),\tau(j)} - \sum_{i=1}^{n} \tilde{U}_i R_{\tilde{\sigma}(i),\tau(j)} \right|$$

$$= 2L_2 \sum_{j=1}^{m} \left| \sum_{i=1}^{n} U_{(i)} R_{i,\tau(j)} - \sum_{i=1}^{n} \tilde{U}_{(i)} R_{i,\tau(j)} \right|$$

$$= 2L_2 \sum_{j=1}^{m} \left| \sum_{i=1}^{n} R_{i,\tau(j)} (U_{(i)} - \tilde{U}_{(i)}) \right| \tag{19}$$

where the last lines follows from $U_i = U_{(\sigma(i))}$, $\tilde{U}_i = \tilde{U}_{(\tilde{\sigma}(i))}$ and reindexing the sum. Since $U_i = \tilde{U}_i$ for every $i \neq 1$, we know that

- If $U_1 \geq \tilde{U}_1$, then $U_{(i)} \geq \tilde{U}_{(i)}$ for every $i \in [n]$.
- If $U_1 < \tilde{U}_1$, then $U_{(i)} \leq \tilde{U}_{(i)}$ for every $i \in [n]$.

This monotonicity property and the fact that $R_{i,j} \geq 0$ for every $i, j$ ensures that

$$(19) = 2L_2 \left| \sum_{j=1}^{m} \sum_{i=1}^{n} R_{i,\tau(j)} (U_{(i)} - \tilde{U}_{(i)}) \right|$$

$$= 2L_2 \left| \sum_{i=1}^{n} (U_{(i)} - \tilde{U}_{(i)}) \sum_{j=1}^{m} R_{i,\tau(j)} \right|$$

$$= \frac{2L_2}{n} \left| \sum_{i=1}^{n} (U_{(i)} - \tilde{U}_{(i)}) \right|$$

$$= \frac{2L_2}{n} \left| \sum_{i=1}^{n} (U_i - \tilde{U}_i) \right|$$

$$= \frac{2L_2}{n} \left| U_1 - \tilde{U}_1 \right| \leq \frac{4L_2 M}{n}$$

By the triangular inequality, the term (16) can be bounded as follows

$$(16) \leq 2 \left\| \sum_{i=1}^{n} \nabla_\theta g_\theta(x_i) U_i \sum_{j=1}^{m} R_{\sigma(i),\tau(j)} - \sum_{i=1}^{n} \nabla_\theta g_\theta(\tilde{x}_i) \tilde{U}_i \sum_{j=1}^{m} R_{\tilde{\sigma}(i),\tau(j)} \right\|_2 \tag{20}$$

$$+ 2 \left\| \nabla_\theta g_\theta(x_1) \sum_{j=1}^{m} R_{\sigma(1),\tau(j)} V_j - \nabla_\theta g_\theta(\tilde{x}_1) \sum_{j=1}^{m} R_{\tilde{\sigma}(1),\tau(j)} V_j \right\|_2 \tag{21}$$

$$+ 2 \left\| \sum_{i=2}^{n} \nabla_\theta g_\theta(x_i) \sum_{j=1}^{m} V_j (R_{\sigma(i),\tau(j)} - R_{\tilde{\sigma}(i),\tau(j)}) \right\|_2 \tag{22}$$

We can bound independently each term in the decomposition,

$$(20) = \frac{2}{n} \left\| \sum_{i=1}^{n} \nabla_\theta g_\theta(x_i) U_i - \nabla_\theta g_\theta(\tilde{x}_i) \tilde{U}_i \right\|_2$$

$$= \frac{2}{n} \left\| \nabla_\theta g_\theta(x_1) U_1 - \nabla_\theta g_\theta(\tilde{x}_1) \tilde{U}_1 \right\|_2$$

$$\leq \frac{2}{n} \Big( |U_1| \|\nabla_\theta g_\theta(x_1)\|_2 + |\tilde{U}_1| \|\nabla_\theta g_\theta(\tilde{x}_1)\|_2 \Big)$$

$$\leq \frac{4 L_1 M}{n}$$

$$(21) \leq 2 L_1 M \Big( \sum_{j=1}^m R_{\sigma(1),\tau(j)} + \sum_{j=1}^m R_{\tilde{\sigma}(1),\tau(j)} \Big)$$

$$= \frac{4 L_1 M}{n}$$

$$(22) \leq 2 L_1 \sum_{i=2}^n \Big| \sum_{j=1}^m V_j (R_{\sigma(i),\tau(j)} - R_{\tilde{\sigma}(i),\tau(j)}) \Big|$$

$$= 2 L_1 \sum_{i=2}^n \Big| \sum_{j=1}^m V_{(j)} (R_{\sigma(i),j} - R_{\tilde{\sigma}(i),j}) \Big| \qquad (23)$$

The last equality is a simple consequence of $V_j = V_{(\tau(j))}$ and reindexing the sum. To bound the last expression, it is useful to see that all the terms $\sum_{j=1}^m V_{(j)}(R_{\sigma(i),j} - R_{\tilde{\sigma}(i),j})$ have the same sign, for $i = 2, \ldots, n$. This will follow from the relationship between the permutations $\sigma$ and $\tilde{\sigma}$. For instance, if $\tilde{\sigma}(1) < \sigma(1)$, it follows that

a) $\tilde{\sigma}(i) \geq \sigma(i)$ *for every* $i \geq 2$. Remember that $\sigma(i)$ denotes the position of $U_i$ in the ordered statistic $(U_{(1)}, \ldots, U_{(n)})$, and $\tilde{\sigma}(i)$ denotes the position of $\tilde{U}_i$ in the ordered statistic $(\tilde{U}_{(1)}, \ldots, \tilde{U}_{(n)})$. Recall also that $\tilde{U}_i = U_i$ for every $i \geq 2$. Therefore, $\tilde{\sigma}(1) < \sigma(1)$ implies that $\tilde{U}_1 < U_1$, and for every $i \geq 2$,

- If $\sigma(i) < \tilde{\sigma}(1)$, then $\tilde{\sigma}(i) = \sigma(i)$.
- If $\tilde{\sigma}(1) \leq \sigma(i) < \sigma(1)$, then $\tilde{\sigma}(i) = \sigma(i) + 1$.
- If $\sigma(i) > \sigma(1)$, then $\tilde{\sigma}(i) = \sigma(i)$.

b) $\sum_{j=1}^m V_{(j)}(R_{\sigma(i),j} - R_{\tilde{\sigma}(i),j}) \leq 0$ *for every* $i = 2, \ldots, n$. If we denote by $G$ the empirical distribution function of $V_1, \ldots, V_m$, then by definition of $R_{i,j}$,

$$\sum_{j=1}^m V_{(j)}(R_{\sigma(i),j} - R_{\tilde{\sigma}(i),j}) = \sum_{j=1}^m V_{(j)} \left( \int_{\frac{\sigma(i)-1}{n}}^{\frac{\sigma(i)}{n}} I\Big(\frac{j-1}{m} < t \leq \frac{j}{m}\Big) dt - \int_{\frac{\tilde{\sigma}(i)-1}{n}}^{\frac{\tilde{\sigma}(i)}{n}} I\Big(\frac{j-1}{m} < t \leq \frac{j}{m}\Big) dt \right)$$

$$= \int_{\frac{\sigma(i)-1}{n}}^{\frac{\sigma(i)}{n}} \sum_{j=1}^m V_{(j)} I\Big(\frac{j-1}{m} < t \leq \frac{j}{m}\Big) dt - \int_{\frac{\tilde{\sigma}(i)-1}{n}}^{\frac{\tilde{\sigma}(i)}{n}} \sum_{j=1}^m V_{(j)} I\Big(\frac{j-1}{m} < t \leq \frac{j}{m}\Big) dt$$

$$= \int_{\frac{\sigma(i)-1}{n}}^{\frac{\sigma(i)}{n}} G^{-1}(t) dt - \int_{\frac{\tilde{\sigma}(i)-1}{n}}^{\frac{\tilde{\sigma}(i)}{n}} G^{-1}(t) dt$$

$$= \int_{\frac{\sigma(i)-1}{n}}^{\frac{\sigma(i)}{n}} G^{-1}(t) - G^{-1}\Big(t + \frac{\tilde{\sigma}(i) - \sigma(i)}{n}\Big) dt \leq 0$$

for every $i = 2, \ldots, n$, where the last bound is consequence of $(a)$ and the monotonicity of $G^{-1}$.

Similarly, if $\tilde{\sigma}(1) > \sigma(1)$, then $\tilde{\sigma}(i) \leq \sigma(i)$ for every $i \geq 2$, which implies $\sum_{j=1}^m V_{(j)}(R_{\sigma(i),j} - R_{\tilde{\sigma}(i),j}) \geq 0$. Finally, the case $\tilde{\sigma}(1) = \sigma(1)$ is trivial, since this implies $\tilde{\sigma} = \sigma$. In any case, the sign property implies that

$$(23) = 2 L_1 \Big| \sum_{i=2}^n \sum_{j=1}^m V_{(j)}(R_{\sigma(i),j} - R_{\tilde{\sigma}(i),j}) \Big|$$

$$= 2 L_1 \Big| \sum_{j=1}^m V_{(j)} \sum_{i=2}^n (R_{\sigma(i),j} - R_{\tilde{\sigma}(i),j}) \Big|$$

$$= 2L_1 \Big| \sum_{j=1}^m V_{(j)} \sum_{i=1}^n (R_{\sigma(i),j} - R_{\tilde{\sigma}(i),j}) - \sum_{j=1}^m V_{(j)}(R_{\sigma(1),j} - R_{\tilde{\sigma}(1),j}) \Big|$$

$$= 2L_1 \Big| \sum_{j=1}^m V_{(j)}(R_{\sigma(1),j} - R_{\tilde{\sigma}(1),j}) \Big|$$

$$\leq 2L_1 M \Big( \sum_{j=1}^m R_{\sigma(1),\tau(j)} + \sum_{j=1}^m R_{\tilde{\sigma}(1),\tau(j)} \Big) = \frac{4L_1 M}{n}$$

Putting everything together, we can conclude that,

$$\|\nabla_\theta W_2^2(g_\theta \# P_{\mathbf{X}}, h_\theta \# P_{\mathbf{Z}}) - \nabla_\theta W_2^2(g_\theta \# P_{\tilde{\mathbf{X}}}, h_\theta \# P_{\mathbf{Z}})\|_2 \leq \frac{12 L_1 M}{n} + \frac{4 L_2 M}{n} \ .$$

$\square$

## G.3 Other Proofs

*Proof of Lemma 5.2.* We adapt the proof of Theorem 29 in Steinke (2022) to the neighboring relation used in this article.

Throughout, datasets are identified up to within-category permutations, so that $M_{\text{batch}}$ is a function of the multiset of points in each category and the subsample below may be regarded as an (unordered) multiset.

For $D = (D_1, \ldots, D_k) \in \mathcal{D}_1^{n_1} \times \cdots \times \mathcal{D}_k^{n_k}$, we write $D_i = (d_1^i, \ldots, d_{n_i}^i)$. The mechanism $M$ draws, independently across categories, a uniformly random index set $S_i \subseteq [n_i]$ with $|S_i| = n_i'$, sets $S = (S_1, \ldots, S_k)$, and applies $M_{\text{batch}}$ to the subsampled dataset

$$D_S := \big( (d_l^1)_{l \in S_1}, \ldots, (d_l^k)_{l \in S_k} \big), \qquad \text{so that} \qquad M(D) = M_{\text{batch}}(D_S).$$

For any fixed index in category $i$, the inclusion probability is $p_i := \mathbb{P}[\, l \in S_i \,] = n_i'/n_i$, and we set $p = \max_i p_i$.

Let $D \sim_k \tilde{D}$. There is a category $j \in [k]$ such that $D_i$ and $\tilde{D}_i$ coincide up to a permutation for $i \neq j$, while $D_j$ and $\tilde{D}_j$ coincide up to a permutation and the replacement of a single element. Since the per-category subsampling is exchangeable and $M_{\text{batch}}$ acts on multisets, we may assume without loss of generality that $D_i = \tilde{D}_i$ for $i \neq j$ and that $D_j, \tilde{D}_j$ differ only in the first point, i.e. $d_1^j \neq \tilde{d}_1^j$ and $d_l^j = \tilde{d}_l^j$ for $l \geq 2$. In particular, on the event $\{1 \notin S_j\}$ we have $D_S = \tilde{D}_S$.

Conditionally on $1 \in S_j$, we define $S_j^\star$ by removing the index 1 from $S_j$ and inserting an index drawn uniformly at random from $[n_j] \setminus S_j$, leaving the index sets of all other categories unchanged, and we write $S^\star$ for the resulting subsampling. Two properties hold:

(P1)  $D_S$ and $D_{S^\star}$ differ by the replacement of the single point $d_1^j$. Hence $D_S \sim_k D_{S^\star}$.

(P2)  As $S_j$ is uniform over the size-$n_j'$ subsets of $[n_j]$, the law of $S_j^\star$ given $1 \in S_j$ coincides with the law of $S_j$ given $1 \notin S_j$. Combined with independence across categories, the law of $S^\star$ given $1 \in S_j$ equals the law of $S$ given $1 \notin S_j$.

Fix a measurable set $E \subseteq \mathcal{O}$. Conditioning on whether the differing index is sampled,

$$\mathbb{P}[M(D) \in E] = (1 - p_j)\, a + p_j\, b, \qquad \mathbb{P}[M(\tilde{D}) \in E] = (1 - p_j)\, a + p_j\, b',$$

where, using that $D_S = \tilde{D}_S$ on $\{1 \notin S_j\}$,

$$a := \mathbb{E}[\mathbb{P}[M_{\text{batch}}(D_S) \in E] \,|\, 1 \notin S_j], \quad b := \mathbb{E}[\mathbb{P}[M_{\text{batch}}(D_S) \in E] \,|\, 1 \in S_j],$$

$$b' := \mathbb{E}\big[\mathbb{P}\big[M_{\text{batch}}(\tilde{D}_S) \in E\big] \,\big|\, 1 \in S_j\big].$$

We establish two inequalities. First, $D_S$ and $\tilde{D}_S$ always differ by at most one replacement in category $j$, so $D_S \sim_k \tilde{D}_S$ and the $(\varepsilon, \delta)$-DP of $M_{\text{batch}}$ yields

$$b \ \leq \ e^\varepsilon b' + \delta. \tag{i}$$

Second, by (P1) and the DP of $M_{\text{batch}}$, for every realisation $\mathbb{P}[M_{\text{batch}}(D_S) \in E] \leq e^\varepsilon\, \mathbb{P}[M_{\text{batch}}(D_{S^\star}) \in E] + \delta$; taking the conditional expectation given $1 \in S_j$ and applying (P2),

$$b \ \leq \ e^\varepsilon\, \mathbb{E}[\mathbb{P}[M_{\text{batch}}(D_{S^\star}) \in E] \,|\, 1 \in S_j] + \delta \ = \ e^\varepsilon a + \delta. \tag{ii}$$

Since (i) and (ii) both hold, for any $\lambda \in [0, 1]$ we have $b \leq (1 - \lambda)(e^\varepsilon a + \delta) + \lambda(e^\varepsilon b' + \delta)$, and thus

$$\mathbb{P}[M(D) \in E] = (1 - p_j)a + p_j b \leq \big[1 - p_j + e^\varepsilon(1 - \lambda)p_j\big]a + e^\varepsilon \lambda p_j\, b' + p_j \delta.$$

Choosing $\lambda = p_j + (1 - p_j)e^{-\varepsilon} \in [0, 1]$ gives $e^\varepsilon(1 - \lambda) = (1 - p_j)(e^\varepsilon - 1)$ and $e^\varepsilon \lambda = 1 + p_j(e^\varepsilon - 1)$, so

$$\mathbb{P}[M(D) \in E] \leq \big[1 + p_j(e^\varepsilon - 1)\big]\big((1 - p_j)a + p_j b'\big) + p_j \delta = \big[1 + p_j(e^\varepsilon - 1)\big]\, \mathbb{P}[M(\tilde{D}) \in E] + p_j \delta.$$

For $\varepsilon \geq 0$ the map $q \mapsto 1 + q(e^\varepsilon - 1)$ is nondecreasing, and $p_j \leq p$. Hence $1 + p_j(e^\varepsilon - 1) \leq 1 + p(e^\varepsilon - 1) = e^{\varepsilon'}$ with $\varepsilon' = \ln\big(1 + p(e^\varepsilon - 1)\big)$, and $p_j \delta \leq p\delta = \delta'$. Therefore

$$\mathbb{P}[M(D) \in E] \leq e^{\varepsilon'}\, \mathbb{P}[M(\tilde{D}) \in E] + \delta'.$$

As $D \sim_k \tilde{D}$ and $E$ were arbitrary, $M$ is $(\varepsilon', \delta')$-DP, which is the claim. The worst case $p = \max_i n_i'/n_i$ corresponds to placing the differing point in the category with the largest sampling ratio. $\square$

*Proof of Corollary C.1.* Formally, with the notation of Definition 2.1, define for the first part $\mathcal{D} = \mathcal{D}_0^{n_0} \times \mathcal{D}_1^{n_1}$, where $\mathcal{D}_j = \mathcal{X} \times \mathcal{Y} \times \{j\}$ in the supervised case, and $\mathcal{D}_j = \mathcal{X} \times \{j\}$ in the unsupervised case, for $j = 0, 1$. Applying Theorem A.1 with $g_\theta = h_\theta$, we can bound the sensitivity of $\nabla_\theta \mathscr{L}_\alpha^{SP}(g_\theta)$ by (11).

For the second part, consider $\mathcal{D} = \prod_{j,k} \mathcal{D}_{j,k}^{n_{j,k}}$, where $\mathcal{D}_{j,k} = \mathcal{X} \times \{j\} \times \{k\}$, for $j \in \{0, 1\}, k \in \{0, \dots, R-1\}$. Under the relation $\sim_{2R}$, given two neighboring datasets, all the terms except one are the same in the sum in (9), and similarly for the gradient expression. More precisely, under the assumptions of the theorem, with the notation adopted in Section 7,

$$\sup_{\mathbf{D} \sim_{2R} \tilde{\mathbf{D}}} \left\| \frac{1}{R} \sum_{k=1}^K \nabla_\theta SW_2^2\Big(\varphi_\theta \# P_{\mathbf{X}_{0,k}}, \varphi_\theta \# P_{\mathbf{X}_{1,k}}\Big) - \frac{1}{R} \sum_{k=1}^K \nabla_\theta SW_2^2\Big(\varphi_\theta \# P_{\tilde{\mathbf{X}}_{0,k}}, \varphi_\theta \# P_{\tilde{\mathbf{X}}_{1,k}}\Big) \right\|_2$$

$$\leq \frac{1}{R} \sup_{\mathbf{D} \sim_{2R} \tilde{\mathbf{D}}} \sum_{k=1}^K \left\| \nabla_\theta SW_2^2\Big(\varphi_\theta \# P_{\mathbf{X}_{0,k}}, \varphi_\theta \# P_{\mathbf{X}_{1,k}}\Big) - \nabla_\theta SW_2^2\Big(\varphi_\theta \# P_{\tilde{\mathbf{X}}_{0,k}}, \varphi_\theta \# P_{\tilde{\mathbf{X}}_{1,k}}\Big) \right\|_2$$

$$\leq \frac{1}{R} \sup_{\substack{k=0,\dots,R-1 \\ (\mathbf{X}_{0,k}, \mathbf{X}_{1,k}) \sim_2 (\tilde{\mathbf{X}}_{0,k}, \tilde{\mathbf{X}}_{1,k})}} \left\| \nabla_\theta SW_2^2\Big(\varphi_\theta \# P_{\mathbf{X}_{0,k}}, \varphi_\theta \# P_{\mathbf{X}_{1,k}}\Big) - \nabla_\theta SW_2^2\Big(\varphi_\theta \# P_{\tilde{\mathbf{X}}_{0,k}}, \varphi_\theta \# P_{\tilde{\mathbf{X}}_{1,k}}\Big) \right\|_2$$

$$\leq \frac{1}{R} \max_{k=0\dots,R-1} \frac{16ML}{\min\{n_{0,k}, n_{1,k}\}} = \frac{1}{R} \frac{16ML}{\min_{j,k}\{n_{j,k}\}}$$

which implies (12). $\square$

