# OpenReview forum: "Learning with Differentially Private Sliced Wasserstein Gradients"
_TMLR — Accepted by TMLR_

### Review · Reviewer_5CUs · 2026-04-06

**Summary Of Contributions:**

This paper introduces a framework for privately optimizing objectives that depend on sliced Wasserstein distances between empirical measures. The main theoretical contribution is a sensitivity analysis of the Wasserstein gradient (Theorem 4.1), showing that the ℓ2-sensitivity of the gradient of W₂² between push-forward empirical measures decays as O(1/n), where n is the number of data points. This result enables the application of the Gaussian mechanism to obtain differential privacy at a vanishing noise cost as the dataset grows.
Building on this theoretical foundation, the authors propose a practical deep learning framework that combines inner-clipping of activations and Jacobians, subsampling-based privacy amplification, and GDP-based privacy accounting. Two applications are demonstrated: (i) DP-SWAE, the first differentially private training procedure for sliced Wasserstein autoencoders, validated on MNIST and Fashion-MNIST; and (ii) a fairness-via-private-in-processing method that adds Wasserstein penalties between conditional distributions of different demographic groups, demonstrated on synthetic data for classification, multidimensional regression, and representation learning.


Key Strengths

•	The core sensitivity bound (Theorem 4.1) is non-trivial and technically clean. The proof carefully tracks how rank permutation changes propagate through the gradient when a single data point is replaced, showing that indirect effects telescope due to sign consistency.

•	The framework is general enough to cover both source-only and two-ended privacy, and the comparison with [1] clearly demonstrates broader applicability.

•	The randomness control strategy in the fairness experiments (Section C.6) is commendable.


Key Weaknesses

•	All experiments are limited to MNIST/Fashion-MNIST or simple synthetic data, raising concerns about scalability.

•	The DP-SWAE generation quality is substantially below competing methods (FID 123.6 vs. DPDM's 5.0 at ε = 10).

•	Fairness experiments rely solely on synthetic data with no real-world benchmarks.

•	The bias introduced by inner-clipping is neither analyzed theoretically nor evaluated empirically.

[1] Alain Rakotomamonjy and Liva Ralaivola. Differentially private sliced Wasserstein distance. In Proceedings of the 38th International Conference on Machine Learning (ICML), volume 139, pp. 8810–8820. PMLR, 2021.

**Audience:**

Yes

**Audience Explanation:**

This paper sits at the intersection of differential privacy, optimal transport, and fairness—three areas with active communities within the ML audience. The theoretical result (O(1/n) sensitivity for Wasserstein gradients) is of independent interest to researchers working on privacy-preserving optimization with non-standard loss structures. The fairness application introduces novel problem settings (private and fair multidimensional regression, private and fair representation learning) that have not been addressed in prior work, which should attract attention from the fairness community. Furthermore, the general framework for privatizing sliced Wasserstein losses could inspire follow-up work on other distributional learning tasks (e.g., domain adaptation, generative modeling with more modern architectures).

**Broader Impact Concerns:**

There are no specific ethical implications or broader impact concerns associated with this work that require further addressing.

**Claims And Evidence:**

Yes

**Claims Explanation:**

The central theoretical claim—that the ℓ2-sensitivity of the Wasserstein gradient is O(1/n)—is rigorously proven in Appendix G.2. The key technical insight, that rank-permutation-induced changes in the weight matrix R have consistent sign across all non-replaced data points and thus telescope, is clearly articulated.

The practical claims are well supported by the presented experiments. The fairness experiments convincingly show that (i) increasing α mitigates bias via the Wasserstein penalty, and (ii) adding privacy has minimal impact on fairness outcomes. The randomness isolation strategy also strengthens the credibility of these observations.

However, a few claims deserve more nuanced support. First, the authors state that DP-SWAE provides “strong downstream utility” (Section 6), but the FID of 123.6 and accuracy of 0.755 at ε=10 are in the lower tier of Table 1. The claim is more convincing for the reconstruction-based accuracy (0.848), but this is not a standard metric and lacks comparison with other DP representation learning methods. Second, the positioning of DP-SWAE as a “complementary tool for representation learning” would be more convincing with direct representation quality evaluations (e.g., linear probing on learned features). Despite these limitations, the core theoretical contributions are well-supported, and  the empirical results substantiate the claims made.

**Requested Changes:**

Critical (required for acceptance):

1.	The fairness experiments (Section 7) are conducted entirely on synthetic data. At minimum, the classification experiment should be validated on a standard real-world fairness benchmark (e.g., Adult or German Credit dataset), with comparison to at least one existing private-and-fair classification method such as [2] or [3]. The synthetic data generation mechanism (Appendix C.2) uses a relatively simple bias structure (Bernoulli mixing with p=0.7), and it is unclear whether the favorable results generalize to more complex, real-world distributional biases.

2.	The inner-clipping technique (Section 5.1) is central to the practical applicability of the framework, yet its bias is not analyzed. The authors should provide either (a) a theoretical bound on the bias introduced by clipping at levels M, L₁, L₂, or (b) an empirical ablation studying how varying these hyperparameters affects both convergence and final performance. At present, the choice of clipping values (e.g., M=1.5, L=√6, C=1 for MNIST) appears dataset-specific with no guidance for practitioners.

3.	Table 1 presents a comparison of generation quality (FID/Acc), but the text claims that DP-SWAE’s value lies in representation learning rather than generation. If so, the current framing is misaligned with the paper's stated goals. The authors should either (a) add representation learning evaluations and compare with other DP representation learning methods, or (b) reposition Table 1 to the appendix and clarify in the main text that the generative comparison is included only for completeness.


Recommended (would strengthen the paper):

1.	The computational overhead of per-sample Jacobian computation is acknowledged (Remark 5.1, Table 5), but no wall-clock training time comparison is provided. Including total training time for DP-SWAE versus non-private SWAE and comparing with the computational cost of competing DP generative methods, would help practitioners assess the practical feasibility of this approach.

2.	Remark A.2 acknowledges that the naive Jacobian clipping (per-coordinate clipping instead of SVD-based spectral norm clipping) is suboptimal. A small experiment comparing the two strategies on at least one task would quantify the gap and help justify the simpler approach.

3.	The sensitivity bound (Theorem 4.1) relies on the quadratic structure of W₂², and Appendix F shows it does not extend to general Wₚ. A brief discussion in the main text on which practical applications require p≠2 and whether the framework can be adapted (e.g., via smoothed approximations) would be valuable.

4.	The visual quality of generated samples at ε=5 (Figure 2, bottom) is noticeably degraded, with several digits being difficult to recognize. A discussion of the practical privacy regime where DP-SWAE remains useful (e.g., ε ≥ 8–10) would help set realistic expectations.


Minor issues:
1.	Page 1, Abstract: “without significant hampering the efficiency” should be “without significantly hampering the efficiency.”

2.	Section 7.4: “further datails” should be “further details.”

3.	The resolution of Figure 4 is low, making it difficult to read the details.

4.	The "-" entries in Table 1 should be clarified in the caption — it is unclear whether they indicate that the metric was not reported or that the method is inapplicable.

[2] Andrew Lowy, Devansh Gupta, and Meisam Razaviyayn. Stochastic differentially private and fair learning. CoRR, abs/2210.08781, 2022.

[3] Hrad Ghoukasian and Shahab Asoodeh. Differentially private fair binary classifications. CoRR, abs/2402.15603, 2024.

---

> ### Author Response · Authors · 2026-05-07
>
> Dear Reviewer 5CUs,
>
> We thank the reviewer for the very detailed and technical review. We are pleased that the reviewer found the core sensitivity bound (Theorem 4.1) to be "non-trivial and technically clean."
>
> We now address the different points raised in the review.
>
> 1. Following the reviewer’s suggestion, we include in Remark C.3 an evaluation of the privacy-utility tradeoff of our method on the Adult dataset and compare it with the results reported in Figure 2 of Lowy et al. (2023), including the baseline of Tran et al. (2021).
>
> 2. We thank the reviewer for the suggestion. We have added an ablation study in Appendix B. Furthermore, we would like to respectfully disagree with the claim that the choice of hyperparameters is highly problem-specific. The clipping constant $C = 1$ is the standard hyperparameter choice in mainstream DP-SGD implementations such as Opacus. The choice $M = 1.5$ simply comes from the fact that, since we want the embeddings to be uniform on a unit ball, $M$ has to be greater than $1$ in order not to induce bias in a neighborhood of the solution, and we simply took a small safety margin. The choice $L = \sqrt{6}$ is detailed in Remark A.2. These hyperparameters were set using these pre-training heuristics and were not optimized, so as not to cheat on the overall privacy budget.
>
> 3. We acknowledge the reviewer’s point that SWAEs are often valued more for representation learning. In the revised version, we have moved Table 1 to the appendix.
>
> 4. We thank the reviewer for highlighting the importance of reporting training time. Training DP-SWAE on MNIST/F-MNIST takes approximately 50 minutes in our implementation. It is not straightforward to establish a fully fair comparison with non-private SWAE. A natural baseline is to run the same number of iterations without clipping or noise addition, while computing the gradient of the sliced Wasserstein distance directly using automatic differentiation. In this case, we avoid the Jacobian computation, which is the main computational bottleneck in our method (see Section E). In this setup, SWAE requires approximately 13 minutes, compared to 50 minutes for DP-SWAE. If the reviewer believes that further benchmarking or additional breakdowns would strengthen the paper, we would be happy to include them in the final version.
>
>  5. We agree that comparing naive per-coordinate Jacobian clipping with SVD-based spectral norm clipping would be valuable. We use the naive variant because it naturally adapts to the flattening approach discussed in Section E, which is also used in practice to simplify the implementation and reduce computational cost by avoiding repeated conversions between flattened and structured tensor representations, in addition to the additional cost incurred by SVD decomposition. Providing an efficient implementation incorporating the SVD-based clipping approach is left for future.
>
>
> 6. We thank the reviewer for the suggestion. We added a remark in Section 4 to explain that the same scaling does not hold for general orders. We also added the important example of Wasserstein GANs to motivate other orders.
>
> 7. We thank the reviewer for the suggestion. We added a sentence in Section 6 discussing that, for generative tasks, utility remains most robust for $\epsilon \geq 10$, while reconstruction remains viable at lower $\epsilon$.
>
> 8. We thank the reviewer for the careful proofreading. We corrected the typos, clarified the “-” entries in the tables, and improved the resolution of Figure 4.
>
>
>
>
> **REFERENCES**
>
> Lowy et al. (2023) Stochastic differentially private and fair learning.
>
> Tran et al. (2021) Differentially private empirical risk minimization under the fairness lens.

---

### Review · Reviewer_hY14 · 2026-04-06

**Summary Of Contributions:**

The paper introduces a general framework for differentially private optimization of sliced Wasserstein objectives, addressing a key gap in the intersection of optimal transport and privacy-preserving machine learning. The core contribution is a novel sensitivity analysis of Wasserstein gradients, showing that their L2-sensitivity scales as O(1/n), despite the absence of a standard finite-sum structure. This result enables the use of classical Gaussian mechanisms to privatize gradients, with the noise cost vanishing as the dataset size increases. Building on this theoretical result, the paper proposes:

* A practical training framework integrating clipping, subsampling, and privacy accounting for Wasserstein-based objectives
* The first differentially private training procedure for sliced Wasserstein autoencoders (SWAEs)
* A general in-processing method for fair and private learning, leveraging Wasserstein penalties between conditional distributions.

Empirical evaluations demonstrate that the approach maintains a favorable privacy-utility trade-off, enables meaningful generative modeling under privacy, and achieves competitive or improved fairness-accuracy trade-offs compared to existing baselines

**Strengths**

* Strong and non-trivial theoretical contribution (sensitivity analysis without finite-sum structure)
* Clear connection between theory and practice (end-to-end DP training recipe)
* Broad applicability across generative modeling and fairness
* First work enabling end-to-end DP training with sliced Wasserstein losses

**Weaknesses**

* Computational overhead due to Jacobian / per-sample gradient requirements
* Limited comparison to state-of-the-art DP generative models (e.g., diffusion-based)
* Some assumptions (boundedness, Lipschitz constants) rely on clipping heuristics in practice

**Additional Comments:**

This is a technically strong and conceptually meaningful contribution. The sensitivity analysis for Wasserstein gradients is particularly novel and addresses a non-trivial challenge in private optimization. The paper would benefit primarily from:

* Stronger empirical positioning,
* Clearer discussion of computational trade-offs,
* Slight improvements in exposition.

Overall, the work represents a significant step toward bridging optimal transport and differential privacy, and has the potential to inspire follow-up research in both theory and applications.

**Audience:**

Yes

**Audience Explanation:**

This work is likely to be of interest to multiple segments of the TMLR audience, including researchers in differential privacy, optimal transport, and Wasserstein methods, practitioners working on fairness in machine learning, and the broader community interested in privacy-preserving deep learning. The paper addresses a fundamental methodological gap: enabling differentially private optimization for distributional objectives that lack a finite-sum structure. This is a recurring challenge in modern ML (e.g., generative modeling, domain adaptation). Moreover, the introduction of:

* DP training for SWAEs, and
* A unified framework for private + fair learning

extends the applicability of Wasserstein-based methods to sensitive-data settings, which is of clear practical and scientific interest.

**Broader Impact Concerns:**

The paper addresses an important societal need by enabling privacy-preserving learning and fairness-aware modeling, both of which are highly relevant in sensitive domains (e.g., healthcare, finance, public policy). However, potential concerns include:

* The method may be used to deploy models that appear fair under certain metrics while still encoding other forms of bias
* As with all DP methods, there is a risk of misinterpretation of privacy guarantees if parameters (\theta, \delta) are not carefully chosen

These concerns are relatively standard for DP/fairness work and do not raise major ethical red flags. A brief clarification on practical privacy levels and limitations would further strengthen the paper

**Claims And Evidence:**

Yes

**Claims Explanation:**

The claims are generally well supported by a combination of theoretical analysis and empirical validation. On the theoretical side, the paper provides:

* A rigorous derivation of Wasserstein gradient expressions,
* A clear sensitivity bound (Theorem 4.1),
* Proper linkage to differential privacy guarantees via the Gaussian mechanism

These results are mathematically grounded and appear internally consistent. On the empirical side:

* Experiments on MNIST and Fashion-MNIST demonstrate the feasibility of DP SWAEs, including reconstruction and generation quality
* Fairness experiments show consistent trends in the fairness-accuracy trade-off, supporting the claims regarding Wasserstein-based regularization under privacy

However, while the evidence is convincing overall, comparisons with strong DP baselines (e.g., diffusion models) are limited, and some experimental settings remain relatively small-scale. Despite these limitations, the evidence sufficiently supports the main claims.

**Requested Changes:**

**Critical (required for acceptance)**
1. **Stronger empirical comparison with state-of-the-art DP generative models**
   * Include comparisons with recent DP diffusion or score-based models
   * Clarify positioning: is the method competitive, complementary, or orthogonal?
2. **Clarify computational cost and scalability**
   * Provide runtime/memory benchmarks
   * Quantify overhead from per-sample gradients / Jacobians
   * Discuss feasibility for larger models/datasets
3. **Better justification and analysis of clipping choices**
   * Provide sensitivity of results to clipping hyperparameters (M, L, C)
   * Discuss bias introduced by clipping more explicitly

**Important (would strengthen the paper)**

4. **Expand ablations**
   * Effect of the number of projections in sliced Wasserstein
   * Effect of batch size on privacy-utility trade-off
5. **Clarify assumptions in theory vs practice**
   * Explicitly distinguish when assumptions hold vs when clipping enforces them.
6. **Improve clarity of presentation**
   * Some sections (e.g., sensitivity derivations) could benefit from more intuition
   * Add a short high-level summary before Section 4
7. **Broader experimental coverage**
   * Include at least one **non-image dataset** (e.g., tabular or text embeddings)

---

> ### Author Response · Authors · 2026-05-07
>
> Dear Reviewer hY14,
>
> We thank the reviewer for the thorough and insightful review. We are pleased that they recognized the "strong and non-trivial theoretical contribution" of our sensitivity analysis and the broad applicability of our framework across generative modeling and fairness.
>
> We now address the different points raised in the review.
>
> 1. We would like to point out that we already report the performance of DP-SWAEs against state-of-the-art DP generative models. For instance, Table 1 and Table 2 both include DP diffusion models (DPDMs). The conclusion is that DP-SWAEs are not competitive with diffusion models, primarily because even without privacy, diffusion models already significantly outperform SWAEs. DP-SWAEs are, however, quite competitive against more ad hoc methods.
>
> 2. As noted in Remark 5.1, the primary overhead comes from computing per-sample Jacobians. Note that we already report in Section E the runtimes of this step compared to classical autodiff, as well as how this step scales with the different hyperparameters of interest.
>
> 3. We thank the reviewer for the suggestion. We included in Appendix B of the revised version an ablation study for all the hyperparameters, which highlights the dependence of the privacy-utility tradeoff on each of them. As for a further theoretical discussion of the bias induced by clipping, we do not know exactly what to say. To the best of our knowledge, the bias of DP-SGD, even without the transport-related difficulty, is still not well understood, so we do not know what to add here for a structure that appears even more difficult to handle, other than to acknowledge its existence.
>
> 4. Please see our answer to Point 3, which includes what is asked here.
>
> 5. We thank the reviewer for the suggestion. We have explicitly added a correspondence table between what matters for the theory and how it is enforced in practice in Section 5.
>
> 6. We thank the reviewer for the suggestion. We added a high-level "Intuition" paragraph in Section 4. The core idea is that changing one data point only affects the rank permutation of the sorted samples locally, and the resulting changes in the gradient expression possess a telescoping property that prevents the sensitivity from exploding.
>
> 7. We agree with the reviewer that non-image datasets would strengthen the paper; we leave this as a future work direction for generative modeling. For the fairness experiments, on the other hand, as requested by Reviewer 5CUs, we added numerical comparisons to the existing literature for fair classification, which is performed on tabular data.

---

### Review · Reviewer_TTJ2 · 2026-04-24

**Summary Of Contributions:**

The paper studies minimizing the Wasserstein distance between the pushforward distributions of two empirical distributions under parameterized functions. The primary technical contribution is a sensitivity analysis of Wasserstein gradient, which allows the authors to apply the Gaussian mechanism. They show how to use this technique to privately perform first order minimization of the Wasserstein loss (possibly with clipping). Finally, they show how their technique can be used to privately train sliced Wasserstein autoencoders.

**Audience:**

Yes

**Audience Explanation:**

I am not overly familiar with this literature, but the sensitivity analysis is new as far as I can tell. As such, I would think this paper can find individuals interested in its results.

**Claims And Evidence:**

Yes

**Claims Explanation:**

Proofs are provided for each statement, and the proofs are direct and presented well.

**Requested Changes:**

The authors claim this is the first DP training procedure for sliced Wasserstein autoencoders. Presumably, this is not the first DP procedure for training of any kind of autoencoder. Could the authors comment on how their procedure compares to previous work on private training of autoencoders?

---

> ### Author Response · Authors · 2026-05-07
>
> Dear Reviewer TTJ2,
>
> We thank the reviewer for the positive assessment of our work, particularly for noting that our proofs are direct and well presented, and that our sensitivity analysis constitutes a novel contribution to the literature. We also appreciate the constructive feedback regarding the positioning of our work.
>
> We now address the different points raised in the review.
>
> 1. While standard differentially private autoencoders (DP-AEs) typically apply DP-SGD to a reconstruction loss (such as MSE), this naive method fails for Sliced Wasserstein Autoencoders (SWAEs) since one needs a bound on the sensitivity of the gradient of the regularization term. The key difference lies in the objective function: SWAEs minimize the Wasserstein distance between the encoded latent distribution and a prior, which does not admit a simple finite-sum structure. Unlike standard DP-AEs, which rely on the sensitivity of finite-sum gradients, our method addresses the sensitivity of the Wasserstein gradient itself. In Equation (6) of our paper, the classical autoencoder loss appears as the first term, which can be trained using standard DP-SGD, while our method enables training of the second term. We have added a comment on this point in Section 6.1.

---

### Decision · Action_Editor_P5cF · 2026-06-11

**Recommendation:** Accept with minor revision

**Additional Comments:**

**Required changes (conditions for acceptance)**

These clarity issues impede a reader's ability to follow or check the paper's central arguments and must be resolved in the minor revision. The most concrete is a definition that does not typecheck:

1. **Lemma 2.3 ($\mu$ is ill-defined).** The GDP parameter $\mu$ is written as $\Delta_2 h / \sigma$, but $h$ is a function, so the expression as written is not a well-defined scalar and $\mu$ cannot be computed from it. I read this as a notation slip: $\mu$ should be the $\ell_2$-sensitivity of $h$ (a scalar, $\Delta_2(h) = \sup_{D \sim \tilde{D}} \lVert h(D) - h(\tilde{D}) \rVert_2$) divided by $\sigma$, not the function $h$ itself divided by $\sigma$. Please correct the statement so $\mu$ is defined through the sensitivity, and check the same notation is used consistently wherever the Gaussian mechanism is invoked downstream.

2. **The experimental section (Section 7) does not support its conclusions on its own.** The fairness experiments are reported too briefly for a reader to judge how the conclusions follow from the results. The main text says too little about the data (how it is generated or sourced, the groups and outcomes, the metrics) and the analysis, and deferring all of this to the appendix leaves the central empirical claims unsupported in the body of the paper. The essential setup and the reasoning from results to conclusions must appear in Section 7 itself, not only in the appendix.

3. **Undefined symbols in Section 3.** $U$ and $V$ appear in Proposition 3.1 and the surrounding text without ever being defined, so the proposition and its proof cannot be read as stated. These must be defined before their first use.

4. **Undefined spaces in the main theorem.** Theorem 4.1 is stated over $\Theta$, $\mathcal{X}$, and $\mathcal{Z}$, but these spaces are never defined (the parameter space, the source and target data spaces, their dimensions, and the assumptions placed on them). Without them the hypotheses and the scope of the main result are unclear and the bound cannot be checked. They must be specified before or within the theorem.

5. **Ambiguous quantity in the main theorem.** The sensitivity in Theorem 4.1 is written $\Delta_2 \Psi_\theta$, but $\Psi$ is defined in the same theorem as a function of the data, $\Psi(X, Z) = \nabla_\theta W_2^2(g_\theta$ # $P_X, h_\theta$ # $P_Z)$, with no $\theta$ subscript. The two forms do not match, so it is unclear exactly which quantity the central result bounds. This must be reconciled, either by defining $\Psi_\theta$ explicitly or by using $\Psi$ consistently throughout.

6. **The proof of Lemma 5.2 is not self-contained.** It is derived from (Steinke, 2022) but the paper does not restate the referred theorem for the reader's convenience, and the proof sketch does not give enough detail to verify the result without reading Steinke's paper.

**Other requested changes**

These are smaller, but should also be addressed in the revision:

1. **Add the runtime numbers.** In the rebuttal the authors gave concrete training times (about 50 min for DP-SWAE versus about 13 min for non-private SWAE on MNIST) and offered to put them in the final version. They are not in the PDF yet; Appendix E only describes the per-sample Jacobian cost in words. Please add the numbers.

2. **Fix a stale reference.** Table 1 moved to Appendix B at Reviewer 5CUs's request, but the main text on p. 10 still calls it "presented in Table 1 in the main text." Please update the pointer.

3. **Fix a garbled sentence in the Introduction.** The sentence "The formal definition of differential privacy, as well as the basic results, and tools surrounding it, was moved to Section 2." reads oddly; "was moved to" describes an editing action rather than the paper's structure. Please rephrase, e.g., "The formal definition of differential privacy and the basic results and tools around it are given in Section 2."

4. **Make the references consistent.** The bibliography mixes formats. A few examples: "Differential privacy has bounded impact on fairness in classification" lists editors while similar entries do not; "Stochastic differentially private and fair learning" has been accepted to ICLR but is cited as an arXiv preprint; "On the Private Estimation of Smooth Transport Maps" is marked "working paper or preprint, January 2025" but has an arXiv version with no ID given; and "Wasserstein GAN" (Arjovsky et al., 2017b) gives only a URL without the arXiv ID. Please standardize: cite the published venue where one exists, and give arXiv IDs consistently for preprints.

5. **Enlarge the text in Figure 3.** The labels in Figure 3 are too small to read at normal size. Please increase the font so the figure is legible.

6. **Clarify "such property" after Lemma 2.3.** In "In particular, Section 4 proves that the Wasserstein gradients enjoy such property, motivating the methods presented in this article.", the referent of "such property" is unclear in a context about the Gaussian DP mechanism. Please name the property explicitly (e.g., bounded $\ell_2$-sensitivity).

7. **Remove "the use" in Section 1.** "leveraging the use the Wasserstein distance between one-dimensional projections" should read "leveraging the Wasserstein distance between one-dimensional projections."

Thanks to the authors for a rebuttal that genuinely improved the paper.

**Audience:**

Yes

**Audience Explanation:**

The paper speaks to three communities that are all active in TMLR: differential privacy, optimal transport, and fairness. Its core problem, DP optimization for objectives with no finite-sum structure, shows up well beyond this paper (generative modeling, domain adaptation, fair learning).

**Claims And Evidence:**

Yes

**Claims Explanation:**

The reviewers agree on soundness. The central result (Theorem 4.1: the $\ell_2$-sensitivity of the $W_2^2$ gradient between push-forward empirical measures decays as $O(1/n)$ without a finite-sum structure) was judged correct by both substantive reviews: Reviewer 5CUs worked through the telescoping argument in detail (rank-permutation changes keep a consistent sign, so they cancel rather than accumulate), and hY14 checked the derivation and its link to the Gaussian mechanism. (I give little weight to Reviewer TTJ2's review, which was too brief to assess soundness; see the Recommendation.) The practical recipe (inner-clipping, subsampling amplification, GDP accounting) is consistent, and the MNIST/Fashion-MNIST and fairness experiments back the claims the paper actually makes.

My reservation is clarity, not correctness. The formal development is harder to follow than it should be: symbols are used before they are defined or are defined inconsistently (the spaces $\Theta$, $\mathcal{X}$, $\mathcal{Z}$ and the $\Psi_\theta$ versus $\Psi(X,Z)$ mismatch in Theorem 4.1; $U$ and $V$ in Section 3; the ill-defined $\mu$ in Lemma 2.3), some proof steps are omitted, and Section 7 reports the fairness experiments too briefly to stand on its own. These must be fixed in the revision, but they are presentation problems rather than flaws in the results, and they do not overturn the reviewers' agreement on soundness.

To the authors' credit, the revision already addressed the earlier evidence-related requests: a real-world fairness comparison on Adult against Lowy et al. (2022) and Tran et al. (2021) (Remark C.3, Figure 16), a hyperparameter ablation over $C$, $L$, $M$, the number of projections, and batch size (Appendix B, Figure 6), a theory-to-practice correspondence table (Section 5), and intuition for the main proof (Section 4). The remaining limitations (no theoretical bound on inner-clipping bias; generation quality behind specialized DP generators, which the paper honestly does not claim to match) are honestly scoped and do not undermine the claims.

---

> ### Author Response · Authors · 2026-06-21
>
> We thank the Action Editor for the careful assessment of the revised manuscript and for the clear list of remaining issues. We are grateful that the Action Editor and the reviewers agree on the soundness of the main results and on the relevance of the paper to the TMLR audience.
>
> In this minor revision, we have focused on improving the clarity, consistency, and self-containedness of the presentation. For ease of reading, we summarize below the changes made in response to the required revisions, followed by the additional requested changes.
>
>
> ### Main changes
>
> 1. **Lemma 2.3 ($\mu$ is ill-defined).**
>    We have replaced the notation $\Delta_2 h$ by $\Delta_2(h)$, where
> $$
> \Delta_2(h) := \sup_{D\sim \widetilde D}||h(D)-h(\widetilde D)||_2
> $$
> denotes the $\ell_2$-sensitivity of $h$. Thus, the GDP parameter is now written as $\mu=\Delta_2(h)/\sigma$. We have made the same notation consistent throughout the paper.
>
>
>
> 2. **The experimental section (Section 7) does not support its conclusions on its own.**
>    We have revised Section 7 so that the fairness experiments can be understood from the main text. Although we still refer to the appendix for the complete experimental details, the revised Section 7 now describes the synthetic dataset and the models trained in each scenario. We have also added selected evaluation metrics to Figure 4 and explain their meaning in the main text, so that the conclusions can be followed directly from the main paper.
>
>
>
> 3. **Undefined symbols in Section 3.**
>     We have revised the beginning of Section 3 so that the notation used in Proposition 3.1 is introduced before the statement. In particular, we now define $U$, $V$, the empirical measures $P_U$ and $P_V$, and the corresponding order statistics before stating the weighted quadratic formula.
>
> 4. **Undefined spaces in the main theorem.**
>    We have revised the beginning of Section 4 to make the notation explicit. In particular, we now specify that $\Theta\subseteq\mathbb R^r$ is the parameter space and that $r$ is the number of parameters. We also clarify that we make no assumptions on $\mathcal X$ and $\mathcal Z$, apart from requiring differentiability with respect to $\theta$. The same clarification has been added at the beginning of Appendix A for the sliced Wasserstein extension.
>
>
>
> 5. **Ambiguous quantity in the main theorem.**
>    We apologize for the typo. We have corrected the mismatch and now use $\Psi$ consistently.
>
>
>
> 6. **The proof of Lemma 5.2 is not self-contained.**
>  In this version, we give the full adaptation of the proof of Theorem 29 in Steinke 2022 to make the proof of Lemma 5.2 self-contained.
>
>
>
> ### Additional changes
>
> 1. **Add the runtime numbers.**
>    We added the runtime numbers mentioned in the rebuttal at the end of Appendix E.
>
> 2. **Fix a stale reference.**
>    We corrected the reference to Table 1 after its relocation to Appendix B.
>
> 3. **Fix a garbled sentence in the Introduction.**
>    We have rephrased the sentence accordingly.
>
> 4. **Make the references consistent.**
>    We have updated and standardized the bibliography. In particular, we now cite published venues whenever available and provide arXiv identifiers consistently for preprints.
>
> 5. **Enlarge the text in Figure 3.**
>    We enlarged the text in Figure 3 so that the labels are readable at normal size.
>
>
> 6. **Clarify "such property" after Lemma 2.3.**
>    We replaced the ambiguous phrase with an explicit reference to the $O(1/n)$ decay of the $\ell_2$-sensitivity.
>
> 7. **Remove "the use" in Section 1.**
>    We have corrected the phrase. Thank you for pointing this out.
>
> Additionally, we have added a link to the code repository needed to reproduce the experiments and included an acknowledgments section.
>
> We thank the Action Editor again for the constructive feedback. We believe these changes address the remaining concerns and make the manuscript clearer and easier to verify.